# Re-evaluation of the Bahariya Formation carcharodontosaurid (Dinosauria: Theropoda) and its implications for allosauroid phylogeny

**Maximilian Kellermann**[1,2]*, **Elena Cuesta**[2,3], **Oliver W. M. Rauhut**[1,2,3]

**1** Staatliche Naturwissenschaftliche Sammlungen Bayerns–Bayerische Staatssammlung für Paläontologie und Geologie, Munich, Germany, **2** Department für Geo- und Umweltwissenschaften, Sektion Paläontologie und Geobiologie, Ludwig-Maximilians-Universität, Munich, Germany, **3** GeoBioCenter, Ludwig-Maximilians-Universität, Munich, Germany

* maximilian.kellermann.paleo@gmail.com

**Data Availability Statement:** All relevant data are within the paper and its Supporting Information files.

## Abstract

The first partial skeleton of a carcharodontosaurid theropod was described from the Egyptian Bahariya Oasis by Ernst Stromer in 1931. Stromer referred the specimen to the species *Megalosaurus saharicus*, originally described on the basis of isolated teeth from slightly older rocks in Algeria, under the new genus name *Carcharodontosaurus saharicus*. Unfortunately, almost all of the material from the Bahariya Oasis, including the specimen of *Carcharodontosaurus* was destroyed during World War II. In 1996, a relatively complete carcharodontosaurid cranium was described from similar aged rocks in Morocco and designated the neotype of the species *Carcharodontosaurus saharicus* in 2007. However, due to the destruction of the original material, comparisons of the neotype to the Egyptian fossils have so far only been done cursorily. A detailed reexamination of the available information on the Egyptian carcharodontosaurid, including a previously undescribed photograph of the exhibited specimen, reveals that it differs from the Moroccan neotype in numerous characters, such as the development of the emargination of the antorbital fossa on the nasals, the presence of a horn-like rugosity on the nasal, the lack of a dorsoventral expansion of the lacrimal contact on the frontals, and the relative enlargement of the cerebrum. The referability of the Egyptian specimen to the Algerian *M. saharicus* is found to be questionable, and the neotype designation of the Moroccan material for *C. saharicus* is accepted here under consideration of ICZN Atricle 75, as it both compares more favorably to *M. saharicus* and originates from a locality closer to the type locality. A new genus and species, *Tameryraptor markgrafi* gen. et sp. nov, is proposed for the Egyptian taxon. The theropods of the Bahariya Oasis and the Moroccan Kem Kem Group are thus not as closely related as previously thought, and the proposed faunal similarities between these two strata need further examination.

**Funding:** The author(s) received no specific funding for this work.

**Competing interests:** The authors have declared that no competing interests exist.

## Introduction

Carcharodontosaurids are a group of medium sized to very large allosauroid theropod dinosaurs with a near cosmopolitan distribution, with members having been described from North America, South America, Africa, Europe and Asia [1–6]. However, the Albian and Cenomanian of Gondwana appears to have been a hotspot for carcharodontosaurid diversity, with multiple taxa being known from northern Africa [1, 7–10] and South America [3, 11–13].

The first carcharodontosaurid remains reported were two teeth found some 5 km apart in the vicinity of Timimoun, Algeria, which were described as a new species of *Megalosaurus*, *M. saharicus* [14, 15]. The specimens came from the "Continental Intercalaire", a series of terrestrial sediments underlying the limestones of the late Cenomanian transgression that have a wide distribution in north-western Africa and are often of problematic stratigraphic assignment [16–18]. The "Continental Intercalaire" in the area of Timimoun was long considered to be Albian in age [14, 15, 19, 20], but Le Loeuff et al. [21] argued for a lower Cenomanian age, principally based on comparisons of the vertebrate fauna with that of the Bahariya Formation of Egypt.

In 1931, Ernst Stromer described the first partial carcharodontosaurid skeleton from the Cretaceous of Northern Africa [1]. The specimen SNSB-BSPG 1922 X 46 came from the Bahariya Formation, from a locality in the northern part of the Bahariya Oasis in Egypt (Fig 1)

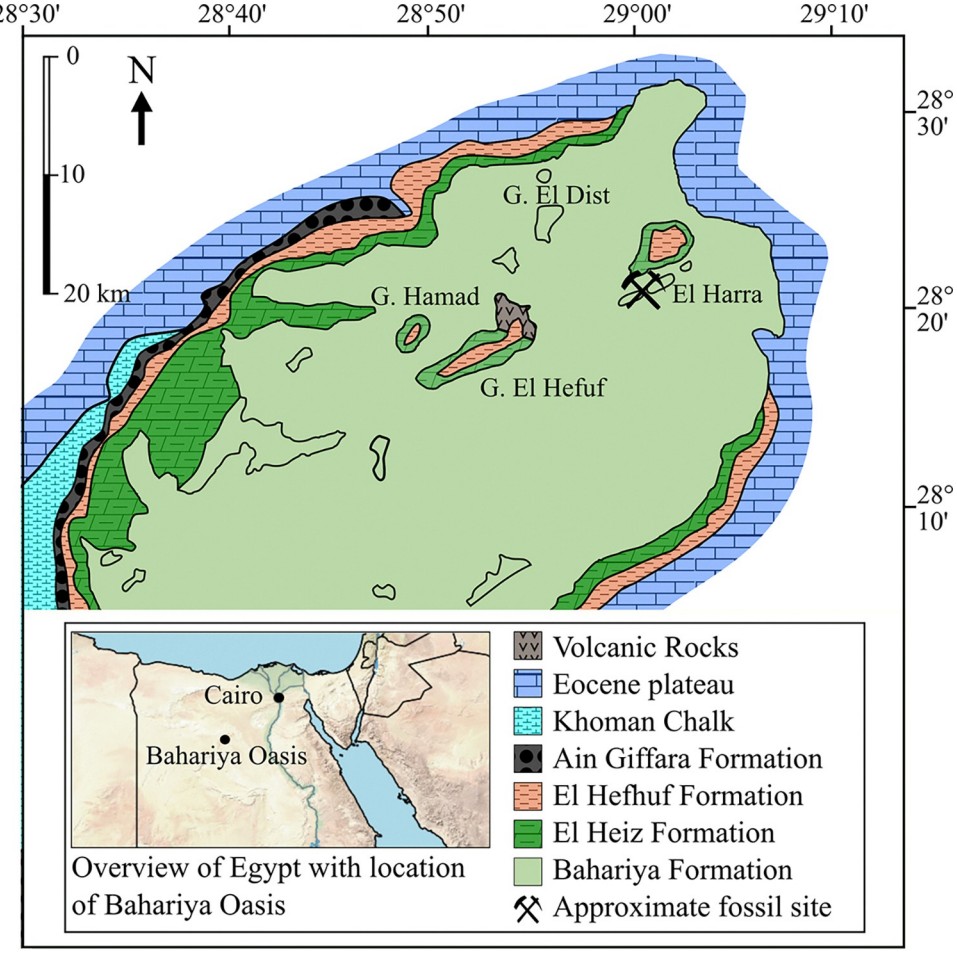

**Fig 1. Geological map of the Bahariya Oasis showing approximate fossil sites of SNSB-BSPG 1922 X 46.** (Modified from [23]).

and comprised skull fragments (maxillae, nasals, partial braincase), vertebrae, partial pubis and ischium, femora, and a fibula. Realizing common features of an associated tooth, Stromer referred SNSB-BSPG 1922 X 46 to *Dryptosaurus saharicus* [15], but proposed a new genus name, *Carcharodontosaurus*, for this species. In a subsequent publication, Stromer furthermore tentatively referred an isolated ilium to *Carcharodontosaurus* [22]. All of the material was brought to Munich, where it was destroyed during the second World War. The only surviving data on the specimen consists of Stromer's descriptions and depictions of the specimen, along with an endocast of the braincase, currently housed in Berlin, and two previously undescribed pictures of the specimen, one of which shows the left maxilla in medial view, while a newly rediscovered photograph shows the specimen as it was mounted in the palaeontological museum in Munich prior to its destruction.

A relatively complete carcharodontosaurid skull (SGM-Din1) from the Kem Kem Group of Morocco was described in 1996 [7]. Based on shared dental features with the Algerian teeth, as well as similarities to the Egyptian carcharodontosaurid described by Stromer, the Moroccan skull was referred to *Carcharodontosaurus saharicus* and eventually proposed as neotype for this species, as the holotype teeth of this species seem to be lost [8].

However, comparisons between the Egyptian and Moroccan carcharodontosaurid have so far only been done cursorily, as the Egyptian carcharodontosaurid has not been reexamined in detail in almost 30 years [24]. With new data available from a previously unpublished photograph of the specimen and the increased knowledge on the anatomy of carcharodontosaurids and the distribution of carcharodontosaurid traits, a detailed reevaluation of the Egyptian carcharodontosaurid is presented here.

## Institutional abbreviations

BYU, Brigham Young University Museum of Paleontology, Provo, Utah, USA; FMNH, Field Museum of Natural History; DINO, Dinosaur National Monument, Vernal, Utah, USA; MCF-PVPH, Museo Carmen Funes, Paleontología de Vertebrados, Plaza Huincul, Neuquén, Argentina; IVPP, Institute of Vertebrate Paleontology and Paleoanthropology, Peking, China; MFN, Museum für Naturkunde, Berlin, Germany; MMCH, Museo Municipal Ernesto Bachmann, Villa El Chocón Neuquén, Argentina; MNBH, Musée national Boubou Hama, Niamey, Niger; MNHN, National Museum of Natural History, Paris, France; MPEF-PV, Museo Paleontológico Egidio Feruglio, Trelew, Argentina; MUCPv-CH, Museo de la Universidad Nacional del Comahue, El Chocón collection, Neuquén, Argentina; NCSM, North Carolina Museum of Natural Science, Raleigh, North Carolina, USA; OMNH, Oklahoma Museum of Natural History, Norman, Oklahoma, SGM, Service Géologique du Maroc, Rabat, Morocco; SMA, Saueriermuseum Aathal, Aathal, Switzerland; SMU, Southern Methodist University, Dallas, Texas, USA; SNSB-BSPG, Staatliche Naturwissenschaftliche Sammlungen Bayerns, Bayerische Staatssammlung für Paläontologie und Geologie, Munich, Germany; USA; UAT, Universitätsarchiv Tübingen, Tübingen, Germany; YPM, Yale Peabody Museum, New Haven, Connecticut, USA

## Materials and methods

Due to the destruction of the original material, the following descriptions and comparisons are mainly based on Stromer's descriptions and figures [1, 22, 25, 26]. A previously undescribed photograph housed in the Huene Archive at the University of Tübingen (UAT 678/20/SNSB-BSPG 1922 X 46) and a photograph of the maxilla in medial view (SNSB-BSPG Archive unnumbered) are used to complement the description of the specimen, as well as to verify Stromer's observations (Figs 2 and 3). The latter photograph has only recently been

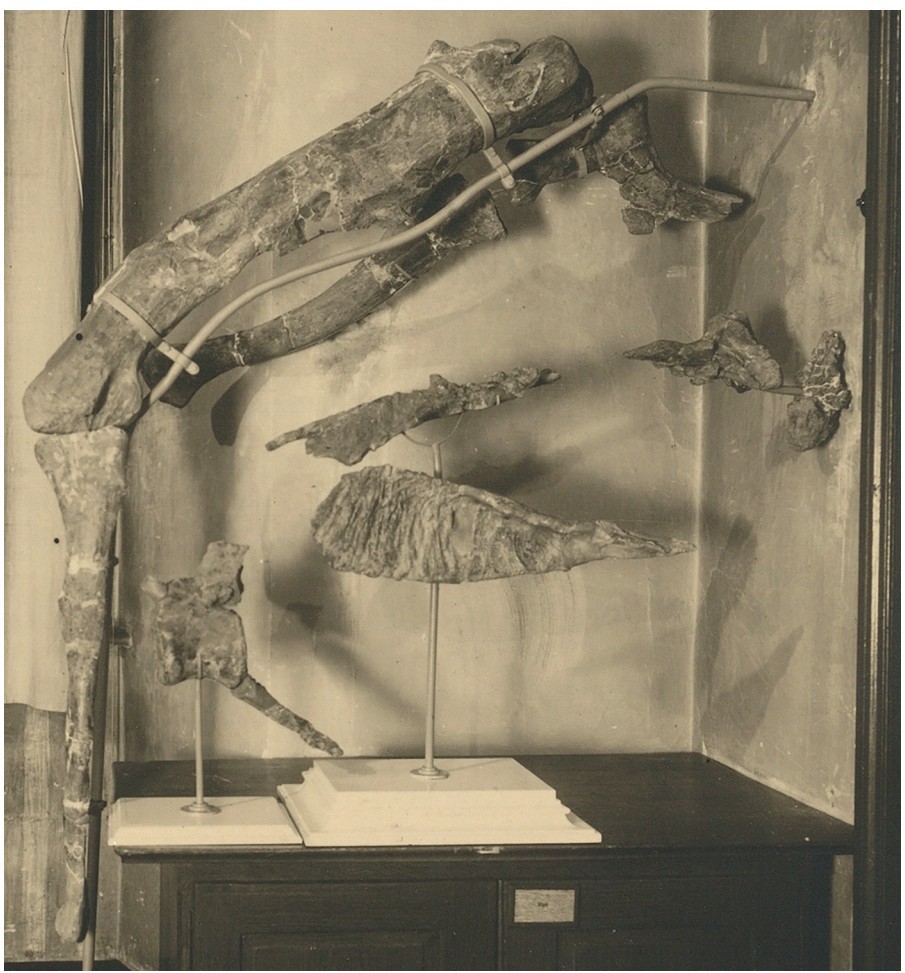

**Fig 2. The holotype of *Tameryraptor markgrafi* (SNSB-BSPG 1922 X46).** Photograph of the mounted specimen at a point prior to April 1944.

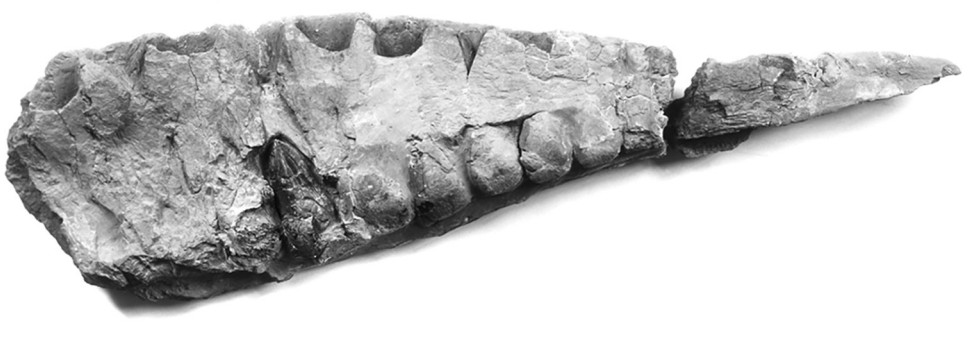

**Fig 3. The left maxilla of *Tameryraptor markgrafi* (SNSB-BSPG 1922 X46).** Photograph in medial view, taken at a point prior to April 1944.

rediscovered and shows the specimen in left lateral view as it was mounted and exhibited in the "Alte Akademie" in the center of Munich, which housed the paleontological museum before its destruction in 1944.

The ilium referred to *Carcharodontosaurus* by Stromer [25] is not taken into consideration here, as there is no overlap with the rest of the material. The nomenclature used for vertebral laminae and fossae in this paper follows Wilson [27] and Wilson et al. [28].

To establish its phylogenetic position, the Egyptian specimen was coded into a large dataset of theropod dinosaurs that is continuously updated and improved on by the Mesozoic Tetrapod work group at the BSPG. This matrix constitutes a modified version of the matrix used by Rauhut and Pol [29], which, in turn, was modified from Wang et al. [30]. For this analysis, the codings for *Carcharodontosaurus saharicus* were revised to include only the material of the Moroccan skull (SGM-Din1), and the Egyptian material was added as a separate OTU, as were several other taxa, such as *Meraxes gigas* and *Lusovenator santosi*. On the other hand, we removed a number of taxa that were less relevant for the question at hand (i.e. more distantly related to carcharodontosaurs and very incomplete). Some taxa were modified, as they included referred material that was not found close to the holotype and did not overlap with the latter. Thus, in *Aerosteon*, we separated the codings for the tibia, fibula, astragalus and calcaneum, as they were found in isolation some km distant from the holotype [31], and in *Carcharodontosaurus iguidensis*, the holotypic maxilla was found some 3 km away from the referred braincase, dentary fragment and cervical vertebra [8], and the latter most probably represents a spinosaurid [32, 33]. The more informative referred braincase (MNBH IGU 3) and dentary (MNBH IGU 5) were coded as a separate OTU in the dataset. Both elements exhibit carcharodontosaurid features, show a similar preservation and were found "in situ embedded in . . . the Echkar Formation" [8: 904], thus we combined the codings of the two elements for our analysis. In *Eocarcharia*, we accepted the referral of an isolated frontal to the holotypic postorbital, as the articular surfaces of the bones match. The referred maxilla (MNBH GAD 7) was again found isolated from the holotype in the very fossiliferous Elrhaz Formation [9] and was thus coded separately. Recently, a partial posterior skull was figured by Sereno et al. [34]: Appendix 5 –Fig 2] and referred to *Suchomimus*. As already mentioned by Schade et al. [35], the postorbital and frontal closely resemble the coeval *Eocarcharia* and are thus likely referable to this genus. Consequently, we added further codings to *Eocarcharia* based on this specimen. Furthermore, we separated the codings of the juvenile specimen of *Megaraptor* (MUCPv 595; [36]) from the larger specimens (MUCPv 79 and 341; [37, 38]), as the main reason for the referral to the same taxon seems to be that these specimens come from the same formation, all represent megaraptorids and are generally similar. However, the juvenile specimen differs in several respect from the larger material (e.g. the more elongate cervical vertebral centra with single, rather small pleurocoels, more gradually expanded acromion process of the scapula), and so a separate coding of this specimen can be used to test the referral of this specimen to the genus. The resulting "Split" matrix thus includes scorings of 831 characters for 179 taxa. To examine the influence of previously lumped codings, all analyses were also performed in a "Merged" dataset. In the "Merged" dataset, codings were not split for *Aerosteon*, *Megaraptor*, *Carcharodontosaurus iguidensis*, *Eocarcharia* and the respective referred specimens, resulting in a dataset that codes 831 characters for 175 taxa. Both datasets can be found in the the supplementary material (S1 Dataset). For a summary of the differences between the two datasets see also Table 1.

Both datasets were analyzed using the software TNT, version 1.5 [39]. An initial new technology search using tree fusing and sectorial search with a driven search to find the minimum length trees 30 times was followed up by a traditional search, using the tree bisection and reconnection (TBR) swapping algorithm with trees from RAM generated by the previous new

**Table 1. Overview of differences between the split and merged datasets.**

| Split Dataset | Merged Dataset |
|---|---|
| *Aerosteon*<br>= holotype (MCNA-PV-3137)<br><br>**MCNA-PV-313**<br>= partial hindlimb referred to *Aerosteon* [31] | *Aerosteon* **merged**<br>= MCNA-PV-3137 + MCNA-PV-313 |
| *Eocarcharia*<br>= holotype (MNBH GAD 2) + referred postorbitals (MNBH GAD 3–6) + referred frontals & prefrontals (MNBH GAD 10 & 11) [9] + partial posterior skull (field no. GAD 302) [34]<br><br>**MNBH GAD 7**<br>= Maxilla referred to *Eocarcharia* [9] | *Eocarcharia* **merged**<br>= MNBH GAD 2–7 + MNBH GAD 10 & 11 |
| *Carcharodontosaurus iguidensis*<br>= holotype (MNBH IGU 2)<br><br>**MNBH IGU3/5**<br>= partial braincase (MNBH IGU 3) and dentary (MNBH IGU 5) referred to *C. iguidensis* [8] | *Carcharodontosaurus iguidensis* **merged**<br>= MNBH IGU 2 + MNBH IGU 3 + MNBH IGU 5 |
| *Megaraptor*<br>= holotype (MUCPv 79) + referred specimen (MUCPv 341) [38]<br><br>**Megaraptorid MUCPv 595**<br>= referred juvenile specimen of *Megaraptor* [36] | *Megaraptor* **merged**<br>= MUCPv 79 + MUCPv 341 +MUCPv 595 |

technology search. The resulting most parsimonious trees (maximum set to 100,000) were summarized using a reduced strict consensus tree [40] obtained by IterPCR [41]. Both "Merged" and "Split" datasets were each run with equal weights and in three different weighted analyses ($K = 9$, $K = 12$, $K = 15$). Constrained analyses were run on both equal weight "Split" and "Merged" datasets to see the change in the number of steps required to move SNSB-BSPG 1922 X 46 in a sister group relation with the neotype of *Carcharodontosaurus saharicus*, SGM-Din1 ("saharicus constraint") and to create a monophyletic *Carcharodontosaurus*, with the specimens referred to *C. saharicus* in a sister group relation with all specimens referred to *C. iguidensis* ("*Carcharodontosaurus* constraint"). Further constrained analyses were done, constraining *Shaochilong* into the Carcharodontosauria ("*Shaochilong* constraint"), *Datanglong* into the Megaraptora ("*Datanglong* Megaraptora") and *Taurovenator* into a sister group relation with *Mapusaurus* ("*Mapusaurus-Taurovenator*"). In the Split dataset yet another constrained analysis was run ("merged constraint") constraining the previously split specimens back together. This was done to see the additional steps needed and to highlight any differences to the Merged dataset.

No permits were required for the described study, which complied with all relevant regulations.

## Nomenclatural acts

The electronic edition of this article conforms to the requirements of the amended International Code of Zoological Nomenclature, and hence the new names contained herein are available under that Code from the electronic edition of this article. This published work and the nomenclatural acts it contains have been registered in ZooBank, the online registration system for the ICZN. The ZooBank LSIDs (Life Science Identifiers) can be resolved and the associated information viewed through any standard web browser by appending the LSID to the prefix "http://zoobank.org/". The LSID for this publication is: urn:lsid:zoobank.org:pub:58FDAD9B-5880-4E2F-98CA-7D42943EA1E0. The electronic edition of this work was published in a journal with an ISSN, and has been archived and is available from the following digital repositories: LOCKSS.

## Results

### The Bahariya Formation carcharodontosaurid

**Horizon and locality.** While the exact locality of the specimen cannot be reconstructed, an approximate locality (Fig 1) can be inferred from the descriptions in Stromer's publications. Stromer [1] reported that the specimen was found "two kilometers from Ain Gedid" on the Western foot of the Gebel Harra in hardened layers of gypsum-free marl corresponding to layer "p" of his 1914 profile [42].

**Description.** The partial skeleton SNSB-BSPG 1922 X 46 preserved parts of the skull, namely the almost complete left and right nasals, and the majority of the left and parts of the right maxilla. Additionally, parts of the braincase, including parietals, frontals, partial supraoccipital and partial otoccipitals were preserved, allowing a reconstruction of an endocast of the braincase (MB. R. 2056) that survived to this day. Several teeth were found associated with the skeleton. Associated parts of the postcranium included three cervical vertebrae, an anterior caudal vertebra, the proximal part of a dorsal rib, two chevrons, the shafts of the right and left pubis, the proximal part of the left ischium, both femora and a left fibula (Fig 4). Although all of this material has been destroyed, we here attempt to provide a description as detailed as possible, based on Stromer's [1, 22, 25, 26], descriptions and illustrations, as well as the photographs of the material (Figs 2 and 3). Measurements of the specimen are given in the supplementary material (S1 Table).

*Nasals.* The nasals of SNSB-BSPG 1922 X 46 (Fig 5A–5D) have not been completely preserved, and were missing the most anterior tips of the sub- and supranarial processes and the posterior frontal contacts. The nasals are subequal in width throughout their length, with a slight constriction at about their mid-length. Stromer noted that the bones were rather thin anteriorly and posteriorly, but thickened in their narrowest portion to a thickness of approximately 3 cm. Here, the medial articular surface was inclined ventromedially (Fig 5C), so that the left and right nasals met in the midline in a "roof-like fashion", thus forming a low midline crest ([1]: p. 6). This contrasts with the situation in the skull of *Carcharodontosaurus* and other carcharodontosaurids, where the internasal suture is flat and the articulated nasals rather slightly indented at their suture, especially in the posterior part (SGM Din 1).

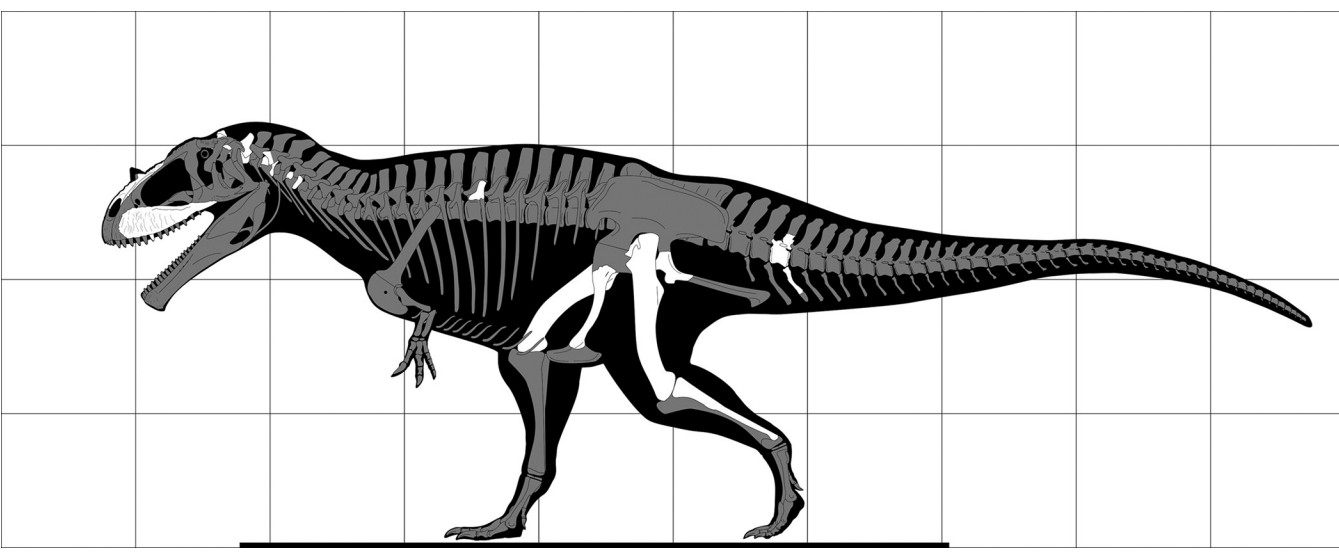

**Fig 4. Skeletal reconstruction of SNSB-BSPG 1922 X 46.** Preserved elements shown in white. Each square is 1m$^2$.

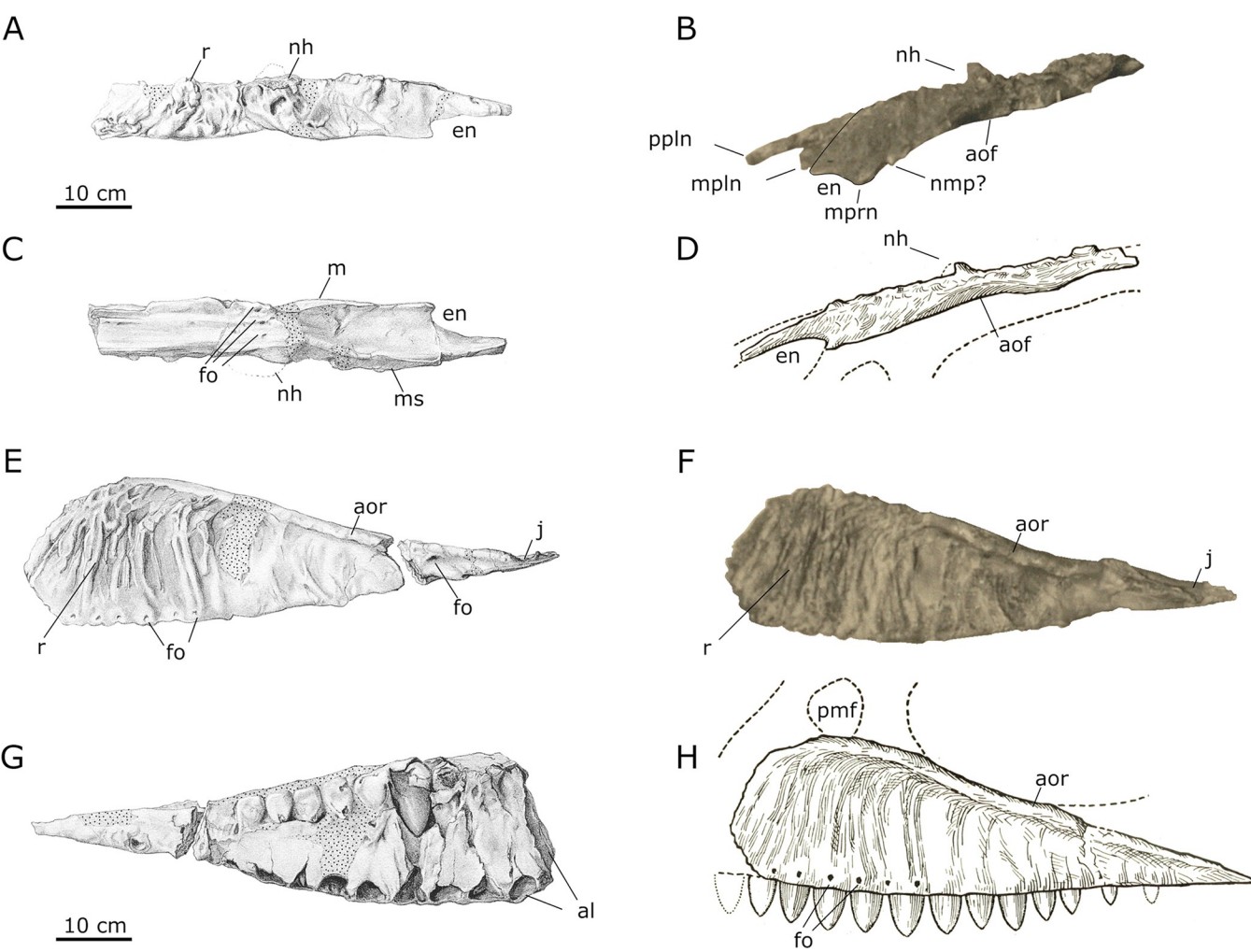

**Fig 5. Nasalia and maxilla of SNSB-BSPG 1922 X 46.** (**A–D**) Nasalia and (**E–H**) left Maxilla of SNSB-BSPG 1922 X 46 in (**A**) dorsal, (**B**, **D–F**, **H**) lateral, (**C**) ventral and (**G**) medial view. Dotted areas represent areas restored with gypsum; *Abbreviations*: **al** alveolus, **aor** antorbital fossa ridge, **en** external nares, **fo** foramina, **j** jugal contact, **m** maxillary contact, **ms** medial symphysis, **mpln** maxillary process of left nasal, **mprn** maxillaryprocess of right nasal, **nh** nasal horn, **nmp** naso-maxillary process, **pmf** premaxillary fenestra, **r** rugosity; *Sources* (**A**, **C**, **E**, **G**) modified after Stromer [1], (**B**, **F**) UAT 678/20/SNSB-BSPG 1922 X 46, (**D**, **H**) modified from Stromer [25].

Stromer [1] described the left nasal as deformed and more incomplete than the right nasal. This explains the unusual shape of the nasal in the exhibited specimen, as the visible premaxillary process appears to be the premaxillary process of the right nasal, rather than the left nasal. Directly below this, the maxillary process of the right nasal is visible, with a small incision marking the dorsal border of the broken premaxillary process of the left nasal (Fig 5B). The nasals exhibit an unusually rugose dorsal surface, with large bumps that are especially noticeable in the picture of the mount (Figs 2 and 5B). Anteriorly, this rugosity is described as restricted to the dorsomedial parts of the nasals. This is in line with the condition present in other carcharodontosaurids, such as *Carcharodontosaurus*, *Giganotosaurus*, *Mapusaurus* and *Meraxes*. A noteworthy feature of the nasals is a c. 3cm high nasal "horn", which is clearly observable in the photograph of the exhibited specimen. Stromer ([1]: p. 6) described this projection as being present on the medial rim of each nasal. While not as pronounced as the nasal horn of *Ceratosaurus*, such a crest or horn is not known for any other carcharodontosaurid.

Anterolateral to this nasal "horn", a marked depression is present, increasing the impression of a medial nasal crest, according to Stromer.

An expansion of the narial fossa to the nasal in the form of a supranarial fossa, as it is present in *Acrocanthosaurus*, *Concavenator*, *Carcharodontosaurus* and *Meraxes* [11, 18, 43, 44] is not clearly visible in the photo of SNSB-BSPG 1922 X 46, but Stromer [1] described the area of the anterior end of the nasal lateral to the median premaxillary process as slightly concave. However, a clearly defined rim of such a fossa, as it is present in *Carcharodontosaurus* ([18]: Fig 134A) is not at all indicated in Stromers figures, nor visible on the photograph. Stromer further described some small foramina on the ventral surface of the nasals, which are also visible in his sketch illustration (Fig 5C). These are most probably vascular foramina, as they are frequently present on the ventral surfaces of theropod nasals (e.g. [45, 46]). Stromer [1] did not mention any pneumatopores in the antorbital fossa, as they are usually present in other allosauroids, including *Giganotosaurus* (MUCPv-Ch1), *Mapusaurus* [12], *Meraxes* [11] or *Carcharodontosaurus* (SGM Din 1), but in contrast noted the presence of these openings in *Allosaurus* as a difference to the nasals described by him.

The antorbital fossa extends onto the lateral surface of the nasals, which represents an allosauroid synapomorphy [7, 47], and is clearly visible in lateral view on the nasals of SNSB-BSPG 1922 X 46. This is visible in the photograph of the specimen and also indicated in Stromer's reconstruction of the skull (Figs 2 and 5B, 5D), and clearly differentiates this taxon from most derived carcharodontosaurids, like *Carcharodontosaurus* (SGM-Din1), *Giganotosaurus* (MUCPv-Ch1), *Mapusaurus* [12], or *Meraxes* [11], where the emargination of the antorbital fossa on the nasal is covered by a laterally overhanging shelf. This feature was already mentioned and illustrated by Rauhut [24], but was not considered in the later synonymization of the specimen with the Moroccan carcharodontosaurid. Anterior to the antorbital fossa, a small process is visible in the exhibited specimen. This process is conform in size and position with the naso-maxillary process present in *Acrocanthosaurus*, *Carcharodontosaurus*, *Concavenator* and inferred for *Eocarcharia* based on an incision on the referred maxilla [43, 44].

*Maxilla*. Stromer [1] mentioned the right maxilla as having initially been preserved, but, apart from two fragments used to supplement his descriptions of the left maxilla, it was broken and lost during transport to Munich. The following descriptions are therefore mainly based on the left maxilla (Figs 3 and 5E–5H). It too was incompletely preserved and broken into two pieces, with minor parts of the jugal ramus and the entire ascending process being absent. The maxilla is also described as "anteriorly incomplete", but it is unclear, how much is missing anteriorly. The medial surface of the left maxilla was initially incomplete, as is visible in the picture of the fossil (Figs 3 and 5I). While some parts of the medial surface of the left maxilla were lost to erosion, others were deliberately removed during preparation, revealing one of the replacement teeth.

As noted by Stromer [1], due to the damage of the bone, it cannot be said whether the maxilla participated in the external nares or was separated from it by a subnarial premaxilla-nasal contact, as in most non-avialian theropods. In its anterior two thirds, the lateral surface of the maxilla is strongly rugose and covered with pronounced vertical ridges and furrows, similar to, but considerably more pronounced than in *Giganotosaurus* (MUCPv-Ch1), *Carcharodontosaurus* [18] and *Meraxes* [11] (Fig 5E and 5F). A strongly developed ridge demarcates the border to the antorbital fossa, which gradually rises from the posterior end anterodorsally, being c. 18 cm distant from the alveolar margin in the anterior part. Thus, as in other carcharodontosaurids more derived than *Acrocanthosaurus* [43] and *Concavenator* [44], the antorbital fossa was obviously largely restricted to the maxilla anterior to the antorbital fenestra, with a very limited or even absent exposure on the jugal ramus. According to Stromer, this antorbital fossa ridge extended up to 3 cm laterally from the lateral surface of the maxilla in its posterior

part, where it is more strongly developed. In its development this ridge is most similar to *Carcharodontosaurus saharicus*, where it rises abruptly from the surface of the maxilla (SGM-Din1). In *Mapusaurus*, *Meraxes* and *Giganotosaurus*, this ridge is less pronounced (MCF-PVPH-108.169; MMCh-PV 65; MUCPv-Ch1), while it is reduced to a small swelling in *Concavenator*, *Acrocanthosaurus* and *Carcharodontosaurus iguidensis* (MCCM-LH 6666; NCSM 14345; MNBH IGU2). Towards the posterior end of the maxilla, Stromer [1] mentions an enlarged, posterolaterally directed nutrient foramen below the ridge, and this is also clearly visible in his illustration of the bone (Fig 5E). Such an enlarged nutrient foramen, usually associated with a posteriorly directed lateral groove (indicated in Stromer's figure), has been described for *Majungasaurus* and other abelisaurids by Sampson & Witmer [46] p. 44), but is also present in some other theropods, including tyrannosaurids (e.g. [48, 49]), *Acrocanthosaurus* (NCSM 14345), and *Meraxes* (MMCh-PV 65). Medial to this foramen, Stromer [1] noted the presence of a dorsally facing groove for the reception of the anterior end of the jugal. Thus, the posterior end of the maxilla, where the jugal articulation is found, was aligned with the main body of the maxilla in SNSB-BSPG 1922 X 46, and not downturned, as it is in *Acrocanthosaurus* [43], *Carcharodontosaurus saharicus* (SGM Din 1), and *Meraxes* [11].

Even though the ascending process was completely absent in SNSB-BSPG 1922 X 46, Stromer (1931 p. 7) mentioned the presence of "spherical cavities in the base of the ascending process". Such large cavities are also present in other carcharodontosaurids [8]. Brusatte & Sereno ([8]: p. 906) interpreted two large pockets in the base of the ascending process of the maxilla of *Carcharodontosaurus iguidensis* as the promaxillary antrum (anteriorly) and the maxillary antrum (posteriorly) and consequently argued that the opening that leads into the posterior recess represents the maxillary fenestra, whereas the promaxillary foramen was said to be absent. However, in other tetanurans, the promaxillary antrum might be subdivided into distinct chambers (e.g. *Piatnitzkysaurus*; PVL 4073), but the maxillary antrum is usually placed more posteriorly, medial to the maxillary antorbital fossa. As the relatively large opening entering the recess in the base of the ascending process in *Carcharodontosaurus* and other carcharodontosaurids is furthermore in the same position and has the anteromedial direction as the promaxillary foramen, we concur with Canale et al. [50] that this opening represents the latter foramen, and a maxillary fenestra is absent in at least most adult carcharodontosaurids. Thus, the Egyptian carcharodontosaurid most probably also had a promaxillary foramen (as also reconstructed by Stromer [25]), but it remains unknown if this was the only additional antorbital opening, as in *Carcharodontosaurus* and *Giganotosaurus*, or if a maxillary fenestra might have been present, as in *Acrocanthosaurus* and at least juvenile specimens of *Mapusaurus* [51].

In both photographs of the maxilla, the alveolar margin is gently convex over its entire length, as it is in the drawing of the maxilla in medial view, whereas the drawing of the lateral view makes it appear rather straight. Stromer ([1] p. 7) describes multiple nutrient foramina being present along the alveolar surface. These are clearly indicated in Stromer's figures ([1] pl I, Fig 6B; [25]: p. 36). A total of 12 alveoli were preserved in the left maxilla, two in the posterior fragment and 10 in the anterior fragment, with the anteriormost alveolus being incompletely preserved. Due to this incompleteness anteriorly, it is unclear how many teeth were originally present in the maxilla, but Stromer [1] considered it unlikely that many tooth positions are missing. The length of the maxilla, and the number of teeth for the Egyptian carcharodontosaurid, can be estimated by examining the extent of the antorbital ridge. In most theropods this ridge is only strongly expressed on the jugal ramus (e.g. *Allosaurus* (USNM 4734), *Mapusaurus* [12]) while in others it reaches the anteroventral corner of the antorbital fossa (e.g. *Carcharodontosaurus saharicus* [18]; *Ceratosaurus* USNM 4735). When comparing SNSB-BSPG 1922 X 46 to these taxa it seems unlikely that much of the maxilla was missing anteriorly. Consequently, the maxilla likely only had 12 or 13 alveoli. This assumption is

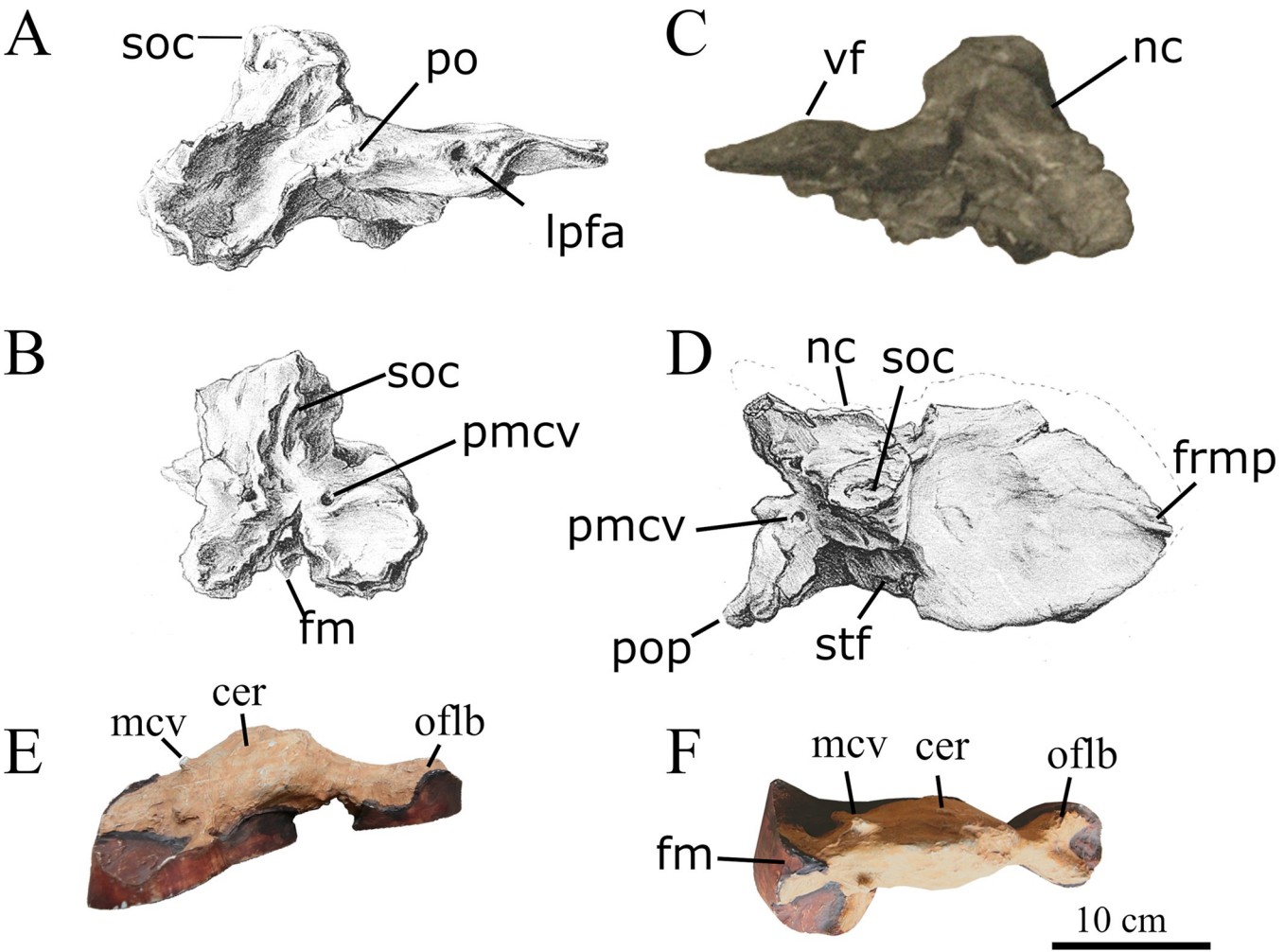

**Fig 6. Braincase and endocast of SNSB-BSPG 1922 X 46.** Elements in (**A**, **E**) right lateral, (**C**) left lateral, (**B**) posterior and (**D**, **F**) dorsal views; *Abbreviations*: **cer** cerebrum, **fm** foramen magnum, **frmp** frontal right median process, **mcv** mid cerebral vein, **lpfa** lacrimal/prefrontal articulation, **nc** nuchal crest, **olfb** olfactory bulb; **pmcv** posterior exit of middle cerebral vein, **po**, postorbital articulation, **pop** paroccipital process, **soc** supraoccipital, **stf** supratemporal fossa, **vf** vaulted frontal; *Sources*: (**A**, **B–D**) modified after [1], (**C**) UAT 678/20/SNSB-BSPG 1922 X 46, (**E**, **F**) MB. R. 2056.

consistent with Stromer's 1931 description (12 teeth) and his 1936 reconstruction (13 teeth). With each of the anterior alveoli being 4 cm long, and separated by 0.8 cm to 1 cm thick interdental bone (according to Stromer [1]) and the anteriormost alveoli being only partially preserved, the length of the maxilla was likely between 75 cm (12 teeth assumed) and 80 cm (13 teeth assumed). It is therefore notably smaller than the maxilla of the neotype of *Carcharodontosaurus saharicus* (90 cm measured from a photograph of the left maxilla of SGM-Din1), even if 14 teeth are assumed (85 cm).

*Braincase.* The braincase of SNSB-BSPG 1922 X 46 (Fig 6A–6D) was heavily damaged on the left lateral side and is missing the parts ventral to the foramen magnum. The absence of sutures in the braincase indicates that SNSB-BSPG 1922 X 46 probably represented a fully mature animal. Apart from the fusion of the actual braincase bones, Stromer [1] also noted that the frontalia and parietalia were fused in the midline and to each other. Apart from these bones, parts of the prootic, the supraoccipital and parts of the right otoccipital seem to have been present. Interestingly, an ossified sphenethmoid, as it is present in other

carcharodontosaurids (e.g. [43, 52] (labeled as orbitosphenoid in this paper)), appears to have been absent. According to Stromer [1] the preserved part of the skull roof was 28 cm long and 14 cm wide, with the original width being estimated to have been 16.2 cm.

As expected, the braincase is overall similar to other carcharodontosaurids, such as *Acrocanthosaurus*, *Giganotosaurus* or *Carcharodontosaurus saharicus* [18, 43, 52]. In lateral view, the occiput above the foramen magnum forms a slightly obtuse angle with the skull roof, as in *Giganotosaurus* and *Carcharodontosaurus* [52]. The shape of the frontals seems to have been very similar to that in *Giganotosaurus*, with a gently convexly curved lateral margin and no postorbital process that is offset from the orbital margin. Towards the anterior midline, the attachment of the right median anterior process is visible in Stromer's figure (Fig 6D), so the fused frontals most probably had paired median anterior processes, as in *Giganotosaurus* [52]. According to Stromer [1], the median part of the frontals was vaulted dorsally anteriorly, and this is clearly visible in the photograph of the specimen (Figs 2 and 5C). This contrasts with the medially flat frontal skull table of other carcharodontosaurids, such as *Giganotosaurus* [52], *Carcharodontosaurus* (SGM-Din 1), *Acrocanthosaurus* [43, 53], and *Meraxes* (MMCh-PV 65). The prefrontal facet is not strongly dorsoventrally expanded, in contrast to *Giganotosaurus* [52] or *Carcharodontosaurus saharicus* (SGM-Din1). Instead it is more similar to the unexpanded prefrontal contact of *Acrocanthosaurus* [43] or *C. iguidensis* [8]. This is clearly observable in both the exhibited specimen and Stromer's illustration of the braincase (Figs 2 and 6A, 6C). Details of the postorbital articulation have not been described by Stromer and cannot be discerned either in the drawing in his publication or the photograph. Thus, it also cannot be said if the postorbital contact extended anteriorly up to the prefrontal facet, and thus if a postorbital-lacrimal/prefrontal contact was present, as in other carcharodontosaurids, like *Carcharodontosaurus* (SGM-Din1).

As in other carcharodontosaurids, the supratemporal fenestrae seem to have been rather short anteroposteriorly and broad mediolaterally, being separated by a broad dorsal roof of the parietals medially. According to Stromer's illustration (Fig 6D), their medial and anterior margin seems to have been slightly raised, and it seems that the anteromedial corners of the fossae were overhang by a lamina, as is the case in other carcharodontosaurids [52]. Most notable on the posterior end of the skull roof there is a prominent dorsal projection formed by the parietals and supraoccipital, which stands at an angle of about 90° to the frontals, similar to *Carcharodontosaurus saharicus*. The nuchal crest is partially preserved on the left lateral side, running posteroventrally from the midline projection, slightly overhanging the supratemporal fenestrae in a way similar to *Giganotosaurus* [52].

Only the dorsal part of the occiput, with the dorsal and parts of the lateral margins of the foramen magnum were preserved. The paroccipital processes were obviously posterolaterally directed, as in most non-avian theropods, but are largely broken off. Two large, posteriorly opening foramina 3 cm above the foramen magnum and approximately at the mediolateral level of the lateral margins of the latter, represent the posterior openings of the mid-cerebral vein. Unlike the situation in *Allosaurus*, these foramina were not placed within a depression on the supraoccipital, but are flush with the posterior surface of the latter. A notable crest extending laterally towards the dorsal margin of the paroccipital process and becoming more marked laterally seems to have been present below the foramen for the vein, as it is also present in *Meraxes* (MMCh-PV 65). The robust supraoccipital crest starts some 5 cm dorsal to the foramen magnum and becomes anteroposteriorly higher and transversely wider towards its dorsal end, where it forms the posterior part of the dorsal projection at the end of the skull roof. It is generally similar to the same structure in other carcharodontosaurids, and wider than the width of the foramen magnum at its dorsal extremity.

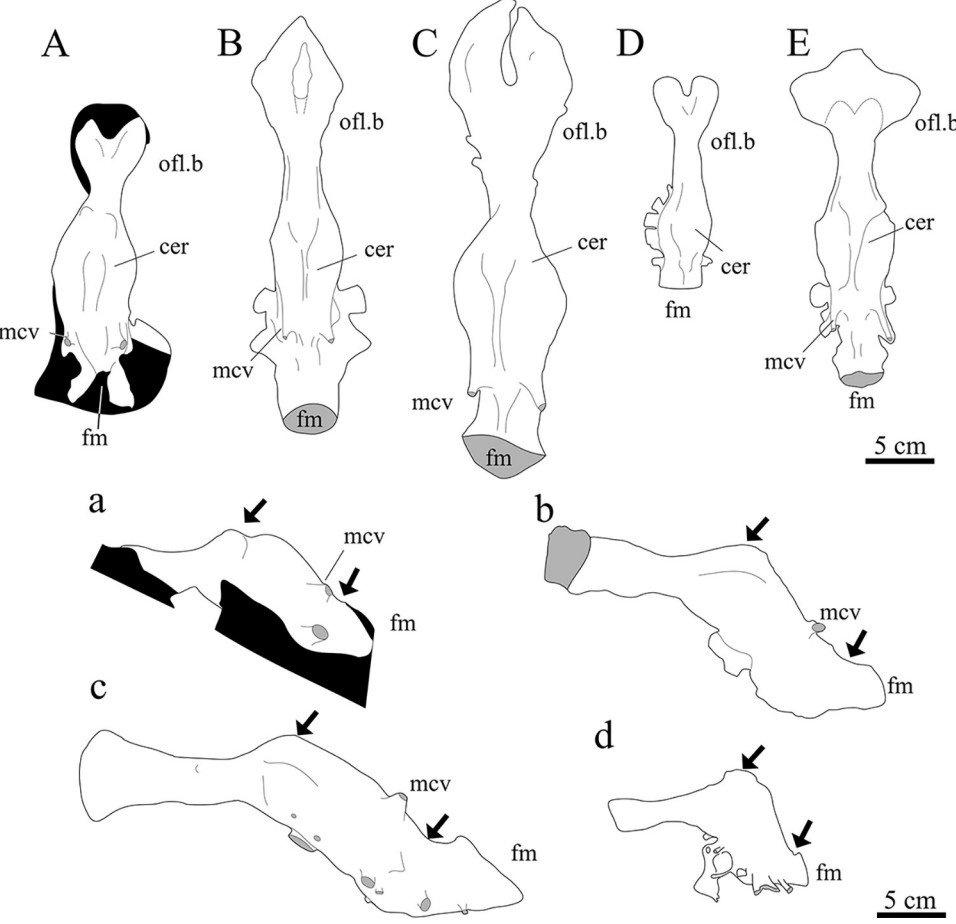

**Fig 7. Cranial endocasts of several carcharodontosaurids.** Elements in (**A**–**E**) dorsal and (**a**—**d**) lateral view: (**A**, **a**) SNSB-BSPG 1922 X 46, (**B**, **b**) *Carcharodontosaurus saharicus*, (**C**, **c**) *Giganotosaurus carolinii*, (**D**, **d**) *Acrocanthosaurus atokensis*, *Carcharodontosaurus iguidensis* (**E**); *Abbreviations*: **cer** cerebrum, **fm** foramen magnum, **mcv** mid cerebral vein, **olf.b** olfactory bulb; Arrows pointing to flexures between forebrain, midbrain and hindbrain; *Sources*: (**A**, **a**) MB.R. 2056, Black area represents pedestal of cast, (**b**) modified from [18], (**B, C, c, D, d**) modified from [54], (**E**) modified from [8].

*Endocast of the braincase.* While not the most detailed, the endocast for SNSB-BSPG 1922 X 46 (Fig 6E and 6F) compares well to the endocasts of other carcharodontosaurids like *Acrocanthosaurus*, *Giganotosaurus* and *Carcharodontosaurus* (Fig 7). A direct comparison between the endocast of the braincase of *Carcharodontosaurus saharicus* and the Egyptian carcharodontosaurid reveals that the cerebrum of the latter is comparatively enlarged, with a maximum transverse width at the cerebral hemispheres of 52 mm. In the considerably larger *C. saharicus*, the maximum transversal width is basically the same size with 53 mm. An enlargement of the cerebrum has also been observed for *C. iguidensis* and *Giganotosaurus* [8, 54]. The olfactory lobes of the Egyptian taxon are also comparatively shorter than in other carcharodontosaurids and more reminiscent of the condition in *Acrocanthosaurus* [43].

*Dentition.* The maxillae of SNSB-BSPG 1922 X 46 did not preserve any functional teeth. However, Stromer observed several replacement teeth in various stages of development in both the left and right maxilla. In one case, the bone covering the tooth was removed during preparation (Figs 3 and 5G). All the teeth associated with the maxilla were described by Stromer as being highly symmetrical and transversely flat, being more than twice as long

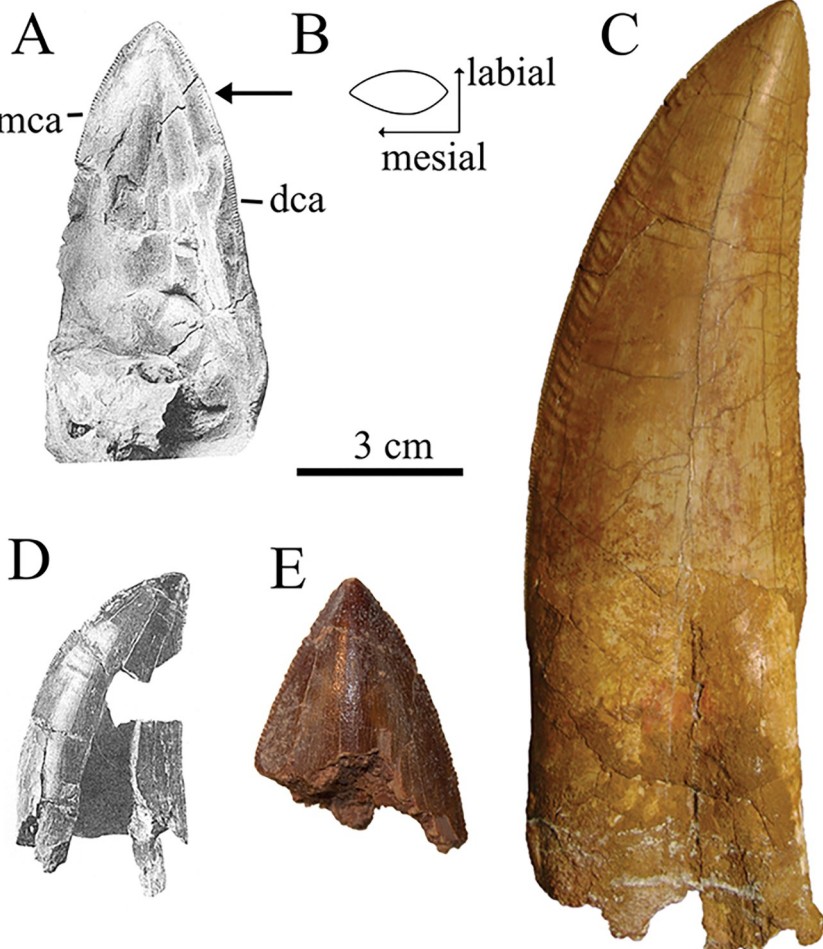

**Fig 8. Comparison of Egyptian and Moroccan carcharodontosaurid teeth.** (**A**) SNSB-BSPG 1922 X 46 maxillary replacement tooth in lingual view, (**B**) Crossection of A at point indicated with arrow; (**C**) SGM-Din1 maxillary tooth in labial view, (**D**) SNSB-BSPG 1922 X 46 isolated tooth in labial view; (**F**) FSAC-KK 914 in labial view; *Abbreviations*: **dca** distal carina, **mca** mesial carina; Sources: (**A**, **B**, **D**) [1]; (**C**, **E**) [18].

anteroposteriorly as thick transversely. Stromer reiterated this point in his 1934 description of the teeth of *Compsognathus* [26]. These very shark-like teeth are not found in any other known theropod, apart from an isolated tooth fragment from the Moroccan Kem Kem Group (Fig 8E), and an isolated tooth assigned to *Mapusaurus* (MCF-PVPH-108.103). All teeth preserved for SNSB-BSPG 1922 X 46 represented replacement teeth and the degree of symmetry can only be verified for one tooth (Figs 3 and 8A), which is still quite far from erupting.

It is unknown how strong the curvature would have been in a fully erupted tooth. The maxilla assigned to *Eocarcharia* (MNBH GAD7) shows an early replacement tooth in the 5th alveolus which is very symmetrical, while the more grown 7th replacement tooth exhibits an increased backwards curvature. An increase in backwards curvature during tooth development might be a pattern of tooth development unique to carcharodontosaurids, since a study of tooth ontogeny of *Tarbosaurus* by Hanai and Tsuihiji [55] shows teeth that already exhibit a strong backwards curvature early on in their development. For now, this cannot be validated, as no study of tooth development for carcharodontosaurids has been conducted yet. In contrast to SNSB-BSPG 1922 X 46, all preserved maxillary teeth of *Carcharodontosaurus saharicus*

show a more typical backwards curvature with the tip of the tooth positioned over the distal half of the tooth base. This degree of backwards curvature appears unlikely to be achievable during the tooth ontogeny of SNSB-BSPG 1922 X 45, as the tooth figured by Stromer is similar in size of a fully erupted tooth of SGM-Din 1, even without accounting for the larger size of SGM-Din 1 (Fig 8). All teeth described by Stromer exhibit both mesial and distal carinae. In the exposed replacement tooth of the maxilla (Fig 8A), the mesial carinae are described as not being "traceable" to the base of the tooth, unlike the distal carinae, which extend all the way to the base of the crown. It is not quite clear if this is an artifact of preservation, as the drawing seems to indicate a damaged mesial margin, while the image of the specimen (Fig 3) shows this tooth as more complete than in the detailed drawing. In his 1934 paper, Stromer describes the mesial carinae as extending "far downwards" [26]: p.78). The extent of the mesial carinae is variable for Allosauroids. In *Allosaurus* and *Acrocanthosaurus* the carinae end well above the cervix, while in *Sinraptor*, *Mapusaurus* and *Carcharodontosaurus saharicus* the mesial carinae extend all along the mesial side of the tooth crown [2, 12, 26, 56].

The maxillary teeth of SNSB-BSPG 1922 X 46 are finely serrated and show 9–10 denticles per 5 mm in the upper half of the tooth. The teeth also exhibit fine marginal horizontal enamel wrinkles, similar to the teeth of other carcharodontosaurids, such as *Giganotosaurus*, *Mapusaurus*, *Tyrannotitan* or *Carcharodontosaurus* [8].

An isolated tooth found in association with SNSB-BSPG 1922 X 46 (Fig 8D) exhibits a curvature more typical for theropods. Stromer interpreted this tooth as possibly belonging to a more mesial part of the toothrow, but it is also possible that this tooth did not belong to SNSB-BSPG 1922 X 46 at all, as there were also multiple other unrelated crocodilian teeth found in association with the specimen [1].

*Axis*. Three cervical vertebrae were preserved, including the axis. Stromer ([1]: p. 11) noted that no neurocentral sutures were visible in any of the preserved vertebrae, furthermore supporting the somatically adult stage of the animal. The axis of SNSB-BSPG 1922 X 46 (Fig 9A and 9B) was damaged posteriorly and was thus depicted only in anterior view. In the photograph of the exhibited specimen it is visible in lateral view, but the resolution does not allow for the recognition of many details. Stromer [1] describes the vertebral body as "evidently short" and anteriorly wider (11 cm) than high (8 cm), with the posterior end being lost to erosion. Stromer [1] further described the odontoid as a 3 cm high and 4.5 cm wide and anteriorly blunt bony peg. Below the odontoid, the crescent-shaped anterior articular surface for the atlantal intercentrum was concave, as can be seen in Stromer's illustration (Fig 9A). Parapophyses were present at about half height on the vertebral body.

With its posterodorsally rising neural arch and c. 3cm low neural spine, the axis of SNSB-BSPG 1922 X 46 bore close resemblance to the axis of *Giganotosaurus* (MUCPv-Ch1) and *Mapusaurus* [12] and clearly differed from the higher neural spines of earlier branching carcharodontosaurians like *Neovenator*, *Concavenator* and, especially, *Acrocanthosaurus* [2, 57, 58]. Stromer [1] described the preserved right prezygapophysis as small and located ventrally on the neural arch. As described by Stromer, and also discernible on his illustration, the articular surface of the prezygapophyses faced dorsally and very slightly laterally (Fig 9A), as is usual in the axes of theropods. The postzygapophyses were described as being much larger and located considerably higher on the neural arch. The ventral margin of the postzygapophyses was located 7.5 cm above the vertebral body. This dorsal displacement was almost as high as the anterior height of the vertebral body. In contrast, the dorsal offset of the postzygapophyses on the axis of *Giganotosaurus* (MUCPv-Ch1) is about half the anterior centrum height. The remains of the incomplete epipophyses are visible on Stromer's drawings (Fig 9A), but it is not apparent if they ended transversely narrow, as in *Mapusaurus*, or wide, as in *Giganotosaurus*.

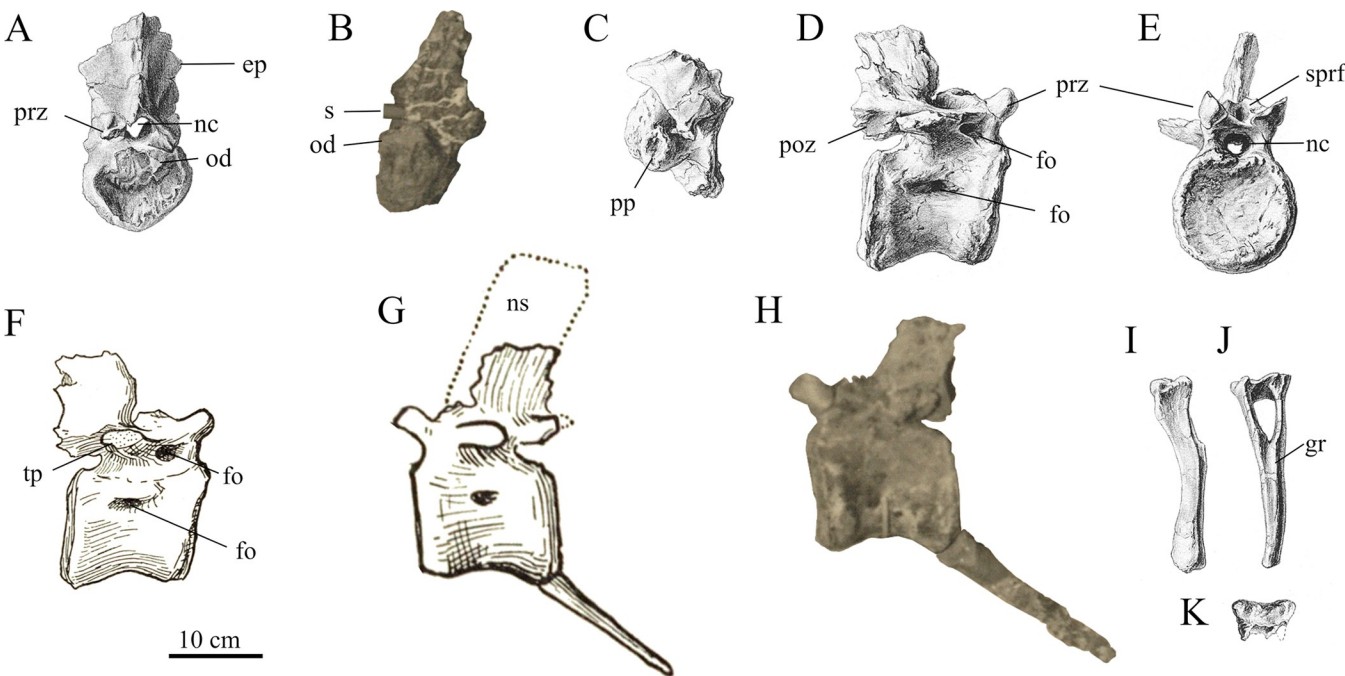

**Fig 9. Axial material of SNSB-BSPG 1922 X 46.** Axis in (**A**) anterior and (**B**) right lateral view, (**C**) 4th cervical in lateral view, (**D, F**) anterior caudal vertebra in right lateral and (**G, H**) left lateral view, Chevron "B" in (**I**) lateral, (**J**) anterior and (**K**) proximal view. *Abbreviations*: **ep** epiphysis, **fo** foramen, **gr** groove, **nc** neural canal, **ns** neural spine, **poz** postzygapophysis, **od** odontoid, **pp** parapophysis, **prz** prezygapophysis, **s** support strut of the mount, **sprf** spinoprezygapophyseal fossa; *Sources*: (**A, C–E, I—K**) [1], (**F, G**) Stromer [25]; (**H**) UAT 678/20/SNSB-BSPG 1922 X 46.

*Middle cervical vertebrae*. As in many derived carnosaurs, the middle cervical vertebra "b" (Fig 9C) was strongly opisthocoelous, but not subspherical, as it is the case in *Giganotosaurus*, or *Tyrannotitan* [50]. The posterior parts of the neural arch, along with the posterior parts of the transverse processes and the neural spine were not preserved. With a ratio of transversal width to anterior centrum height of ~1.3, the vertebral body was slightly wider than the vertebrae of *Giganotosaurus* (~1.1 [MUCPv-Ch])) and *Tyrannotitan* (~1.1 [50]) and more similar to *Allosaurus fragilis* (~1.4 [45]). A single undivided pneumatic foramen was present on each side of the vertebral body within a deep groove. This is unlike the situation in *Acrocanthosaurus*, *Tyrannotitan* [2, 50] and *Giganotosaurus* [MUCPv-Ch1] where the pneumatic foramina are subdivided by a thin lamina. There appears to be some variability in the development of cervical pneumaticity both for carcharodontosaurids. Russel [59], described a carcharodontosaurid cervical vertebra where the pneumatic foramen is only subdivided on one side of the vertebra and a similar situation is also present in the axis of *Giganotosaurus* [MUCPv-Ch1]. Stromer [1], described a posteriorly thickening ventral keel on the ventral side of the vertebral body. Such a keel is absent in the medial cervical vertebrae of many other allosauroids, such as *Tyrannotitan*, *Acrocanthosaurus*, *Allosaurus* or *Neovenator* [2, 45, 50, 58]. Another notable feature of SNSB-BSPG 1922 X 46 was the strong dorsal offset of the anterior articular surface compared to the posterior articular surface. In this respect, this vertebra is similar to the 4th cervical of *Sinraptor* and *Allosaurus fragilis* [45, 60]. The location of the parapophyses, which are located ventrally on the body and project roughly 1 cm laterally and the ventrally directed transverse processes lend additional support for an identification as the 4th cervical vertebra. The dorsal offset of the anterior articular surface is much more pronounced than in any of the preserved cervicals of *Giganotosaurus*, *Tyrannotitan*, *Neovenator* or *Acrocanthosaurus*

(MUCPv-Ch1; [2, 50, 58]), but this could be a result of a different position in the cervical series.

Stromer described the presence of a deep prezygapophyseal centrodiapophyseal fossa below the robustly developed prezygodiapophyseal lamina. This fossa is also developed as a deep pit in the cervical vertebrae of *Giganotosaurus* (MUCPv-Ch1). Stromer [1] further noted that the prezygapophyses were broad and short and directed slightly laterally and ventrally from the anterior part of the neural arch. Their articular surfaces were subcircular and oriented dorsomedially and slightly anteriorly.

A third, more posterior cervical vertebra "c" was mentioned by Stromer, but due to its strong deformation it was neither pictured nor described in detail. It was only described as being strongly opisthocoelous and possessing a pleurocentral cavity with a singular pneumatic foramen on each side.

*Anterior caudal vertebra*. The single preserved anterior to anterior mid-caudal vertebra of the Egyptian carcharodontosaurid (Fig 9D–9H) was slightly longer than high and amphiplatyan. Pneumatic foramina were present on either side of the vertebral body. While the exact development on the left lateral side is not clearly discernable from the photograph of the mounted specimen, Stromer's drawing of the right side depicts it as a roughly circular foramen in an anteroposteriorly elongated depression, while a small circular foramen is indicated in the drawing of the left lateral side (Fig 9D and 9G). While the anteroposteriorly elongated depressions are also present in *Giganotosaurus*, *Tyrannotitan*, *Acrocanthosaurus* and *Mapusaurus* (MUCPv-Ch1; MCF-PVPH-108.81; [2, 50]), large pneumatic foramina in the vertebral centrum are an unusual feature that is shared with megaraptorans and the enigmatic South American *Aoniraptor* [61, 62]. The ventral surface was not directly described by Stromer, aside from noting that it was strongly convex transversely. However, in the comparison with *Ceratosaurus*, *Dryptosaurus* and SNSB-BSPG 1912 VIII 62, Stromer notes the absence of a ventral groove for SNSB-BSPG 1922 X 46 [1, 22]. Furthermore, he mentioned the presence of the articular facet for the chevron at the posterior end.

The neural arch was low and stretched over the entire length of the vertebral centrum. The prezygapophyses were described as short and broad with circular articular surfaces that faced dorsomedially [1]. According to the illustration of the anterior view of the vertebra (Fig 9E), their inclination from the horizontal is approximately 45°. Deep spinopost- and spinoprezygapopyseal fossae, delimited by well-developed spinopre- and spinopostzygapophyseal laminae were described by Stromer [1]. This is a feature that is shared with other carcharodontosaurids, such as *Tyrannotitan*, *Giganotosaurus*, *Meraxes* and *Mapusaurus*. A foramen posterior and ventral to the prezygapophyses that was dorsally bordered by the prezygodiapophyseal lamina and opened medially into the spinoprezygapophyseal fossa was described by Stromer for SNSB-BSPG 1922 X 46. The foramen on the right side is clearly visible in the lateral view of the vertebra figured by Stromer (Fig 9D and 9F). Apparently equivalent pneumatic structures have recently been described in the megaraptoran *Maip* and the probable megaraptoran *Aoniraptor* [63, 64]. Other megaraptorans, such as *Aerosteon*, *Murusraptor* or *Megaraptor*, as well as the enigmatic *Siats* also exhibit pneumatic openings in the prezygapophyseal centrodiapophyseal fossa (FMNH PR 3059; MUCPv 341; [65, 66]), but it is unclear if these also communicated with the spinoprezygapophyseal fossa. Among carcharodontosaurids the only taxon exhibiting a pneumatization of the neural arch of caudal vertebrae is *Meraxes*. However, here the pneumatic foramina are located anteroventral to the postzygapophyses and open into the spinopostzygapophyseal fossa instead [11].

The transverse processes were located centrally on the neural arch and directed laterally and slightly posteriorly, but not dorsally. A seemingly short and robust anterior centrodiapophyseal lamina was present, and dorsally defined by the aforementioned pneumatic foramen.

A posterior centrodiapophyseal lamina was apparently absent, although a slight thickening seems to have been present centrally below the transverse process, according to Stromer's illustration. Although the vertebra is not figured in posterior view, it seems unlikely that a well-developed hyposphene was present, as the margin of the neural arch is deeply incised posteriorly below the postzygapophyses. Likewise, the illustration of the vertebra in anterior view by Stromer does not show any indication of a hypantrum between the prezygapophyses, which were relatively widely spaced.

*Dorsal ribs*. The proximal part of a dorsal rib was found in association with SNSB-BSPG 1922 X 46. This rib is neither depicted by Stromer, nor present in the photo of the exhibited specimen, and no information was provided other than that it was plate-like and double-headed.

*Chevrons*. Two anterior chevrons were found associated with SNSB-BSPG 1922 X 46. Here they will be referred to as chevron "A" and "B". The slightly larger chevron "A" was represented by a proximal articular end and a fragment of associated shaft below the symphysis. Chevron "B" (Fig 9I–9K) was better preserved and its shaft was transversely compressed and slightly anteroposteriorly expanded distally. Stromer [1] described a slightly anteriorly convex ridge that divided the proximal articular surface into an anterior and posterior part. A small anterior spur is present anterolaterally on either side of the proximal articular end of both chevrons, as is usually the case in theropods [47]. Stromer's illustration (Fig 9I and 9J) furthermore shows that the chevron was slightly posteroventrally bowed and had a notable longitudinal groove below the hemal canal on the anterior surface.

*Pubis*. The left and right pubes were incompletely preserved, with their proximal and distalmost parts missing; in addition, only the bases of the laminae of the pubic apron were preserved. In lateral view, the robust pubic shaft curved notably anteriorly (Fig 10A and 10B), resembling the condition in *Giganotosaurus*, *Meraxes*, *Allosaurus*, and some specimens of *Acrocanthosaurus* (MUCPv-Ch1; [2, 11, 45]), and differing from the straight and slender pubic shafts of *Tyrannotitan*, *Neovenator* or *Concavenator* [50, 58, 67]. While not depicted in anterior view, Stromer described the pubic shafts as showing little mediolateral curvature. Thus, the pubes seem to have lacked the marked medial flexure at the level of the pubic tubercle that is seen in some theropods (e.g. *Condorraptor*, *Allosaurus* [68, 69]). Above the apron, the pubic shafts expanded rapidly posteriorly. Stromer [1] contrasted this from the condition present in *Allosaurus*, where the pubic shafts remain narrow and circular in cross-section above the pubic symphysis. Instead he compared the pubis of SNSB-BSPG 1922 X 46 directly to *Ceratosaurus* and speculated on the presence of a pubic foramen in SNSB-BSPG 1922 X 46. However, although the left pubis figured by Stromer ([1] pl. 1, Fig 13) clearly flexes posteriorly towards the ischial peduncle at the proximal break, there is no indication of a medial posterior flange below this flexure, as it is usually present in theropods that have an enclosed obturator foramen. The presence of an enclosed obturator foramen is an unusual feature for allosauroids, but it seems to be present in *Meraxes* [11]. However, based on the illustration of the pubis of this taxon in Canale et al. ([11]: Fig 1C), the ventral border of this foramen seems to be formed by a flange that extends from the ischial contact posteroventrally and is not connected to the pubic shaft, rather than a medially positioned flange extending from the pubic shafts towards the ischial peduncle. Thus, it is unclear whether the enclosed pubic peduncle in *Meraxes* is real or might be an artifact of preservation, for example by a replacement of the proximal end of the lamina that forms the pubic apron.

More distally, the pubic shafts are round in cross-section up until the half-length of the shaft, from which point onward they were described as lateromedially compressed, as is the case in *Meraxes* [11]. The pubic symphysis is described by Stromer [1] as starting at the half-length of the shaft and extending for 40 cm distally, before opening into a pubic fenestra.

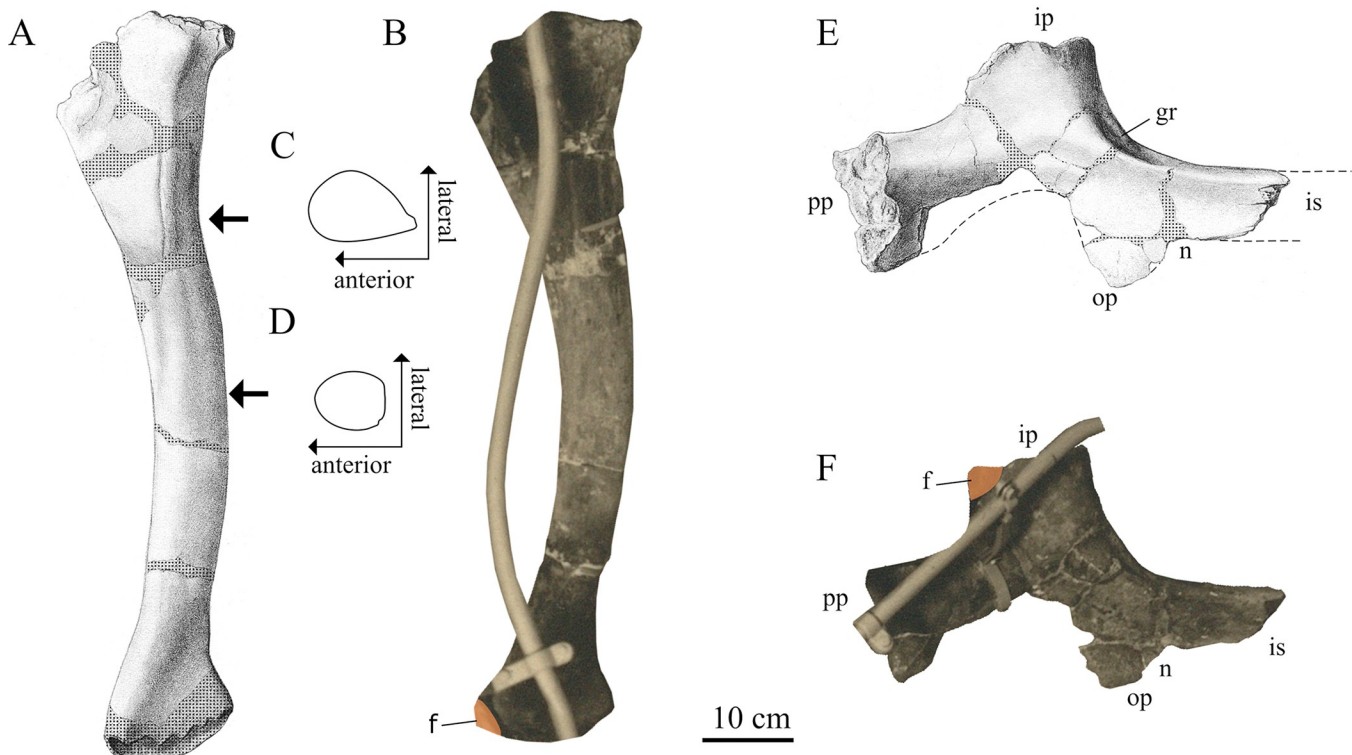

**Fig 10. Pelvic material of SNSB-BSPG 1922 X 46.** (**A**, **B**) Right pubis in lateral view, (**C**, **D**) crosssections of pubis at areas indicated with arrows, (**E**, **F**) right ischium in lateral view; *Abbreviations*: **f** area covered by femur in mounted specimen, **gr** groove, **ip** ischial peduncle, **is** ischial shaft, **n** notch, **op** obturator process, **pp** pubic peduncle; *Sources*: (**A**, **C–E**) modified from [1] (**B**, **F**) UAT 678/20/SNSB-BSPG 1922 X 46.

Distally, the pubic shafts expanded to form a pubic boot, but most of the boot was missing, so nothing can be said about its development.

*Ischium.* The left ischium was rather incompletely preserved. Most of the ischial shaft, parts of the obturator process, as well as the ventral margin of the pubic peduncle were missing. Proximally, the ischium was separated into a short ischial and a comparatively long pubic peduncle (Fig 10E and 10F). The pubic peduncle was expanded towards the pubic articulation. While the exact degree of this expansion is not verifiable, due to the damaged ventral margin of the pubic peduncle, it appears to have been less dorsoventrally expanded than the pubic peduncle of *Giganotosaurus* (MUCPv-Ch1), and instead more closely resembled the ischium of *Mapusaurus* or *Meraxes* [11, 12]. Similarly, the ischial peduncle closely resembled the condition in *Mapusaurus* or *Meraxes* and was less expanded than in *Giganotosaurus*. Unlike *Meraxes*, neither the pubic, nor iliac peduncle were fused to the ilium or pubis, respectively, in SNSB-BSPG 1922 X 46. Stromer [1] described a distinct notch on the iliac articulation for a prong of the ilium. This peg and socket articulation is shared with other carcharodontosaurids and *Siats* and differentiates it from the ischia of *Allosaurus fragilis* and sinraptorids [67, 70]. A distinct groove was located proximally and dorsolaterally on the ischial shaft at its base (Fig 10E). This groove was interpreted by Cuesta et al. [67] as a reduced ischial tuberosity.

A curious feature of SNSB-BSPG 1922 X 46 is the ischial shaft, which, judging from the angle between the main axis of the pubic peduncle and the ischial shaft was held in an almost horizontal position. This is similar to the ischium of *Siats* and contrasts with the ischia of other carcharodontosaurids like *Acrocanthosaurus*, *Concavenator*, *Mapusaurus*, *Meraxes*, *Giganotosaurus* [MUCPv-Ch1] and *Tyrannotitan* [2, 12, 50, 67, 70] (Fig 11).

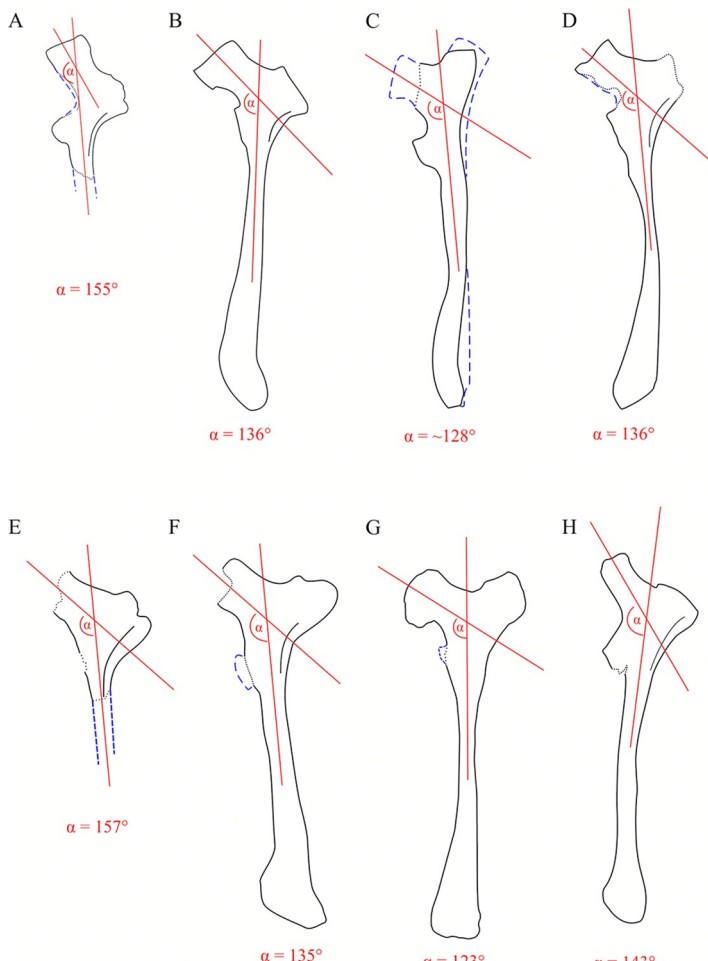

**Fig 11. Ischia of carchaodontosaurids in lateral view.** (**A**) SNSB-BSPG 1922 X 47, (**B**) *Meraxes*, (**C**) *Tyrannotitan*, (**D**) *Mapusaurus*, (**E**) *Siats*, (**F**) *Acrocanthosaurus*, (**G**) *Giganotosaurus*, (**H**) *Lusovenator*; (dashed blue lines) reconstruction; (dotted lines) damaged surface of bone; Not to scale; Sources: (**A**) [1], (**B**) [11], (**C**) mirrored from [50], (**D**) [12], (**E**) mirrored from [70], (**F**) mirrored from [2], (**G**) MUCPv-Ch1, (**H**) mirrored from [71].

The obturator process was present close to the proximal end of the ischium and was most likely separate from the pubic peduncle. Its exact shape is hard to determine, as it was incompletely preserved. However, it was almost certainly better developed than the small ridges present in *Giganotosaurus* (MUCPv-Ch1), *Mapusaurus* or *Meraxes* [11, 12], and likely more closely resembled the obturator process of *Tyrannotitan* [50]. Stromer [1] described a small notch below the obturator process, directly comparing it to *Allosaurus*. The exact development of this notch is unclear from the drawing, but is somewhat discernible in the photograph of the exhibited specimen.

*Femur.* Both femora were strongly compressed and broken during transport to Germany. Therefore, Stromer was unable to fully reconstruct the femoral shafts and an estimation of their diameter or curvature was therefore impossible. In the exhibited specimen the femoral shaft was reconstructed with a slight posterior curvature in its distal part (Figs 2 and 12), but it is not verifiable if this was the original shape. In contrast to the femoral shaft, both the distal and proximal ends of the femora were preserved with little deformation. The femoral head was oriented proximo medially, similar to other carcharodontosaurids, such as *Mapusaurus*,

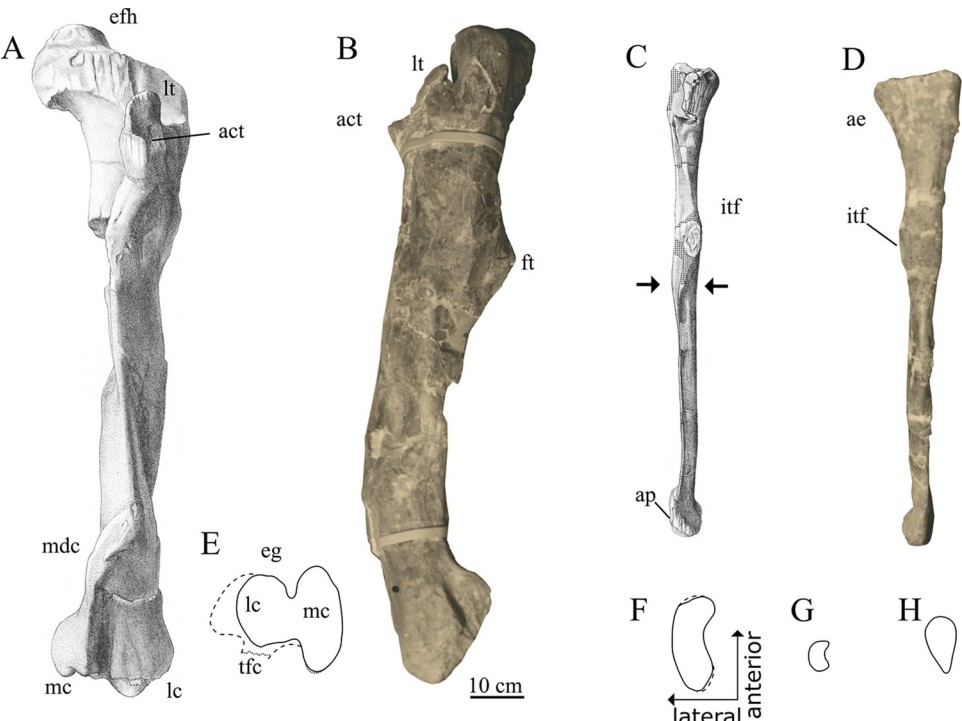

**Fig 12. Hind limb elements of SNSB-BSPG 1922 X 47.** Right femur in (**A**) anterior, (**B**) lateral and (**E**) distal view, right fibular in (**C**) anterior, (**D**) lateral, (**F**) proximal, (**H**) distal view and (**G**) crossection of fibular shaft at area indicated with arrows; *Abbreviations*: **act** accessory trochanter, **ae** anterior expansion, **ap** articular surface of ascending process of astragalus, **efh** dorsal expansion of femoral head, **eg** extensor groove, **ft** fourth trochanter, **itf** insertion of tibiofibularis muscle, **lc** lateral condyle, **lt** lesser trochanter, **mdc** mediodistal crest, **tfc** tibiofibularis crest; *Sources*: (**A, C, E, F–H**) [1], (**B, D**) UAT 678/20/SNSB-BSPG 1922 X 46.

*Meraxes*, *Giganotosaurus* and *Tyrannotitan* [11, 12, 50], but more so than in *Concavenator* [67] and *Acrocanthosaurus* (NCSM 14345). Furthermore, the femoral head of SNSB-BSPG 1922 X 46 exhibited a notable rounded dorsal expansion which is not present in any of the aforementioned carcharodontosaurids. Stromer described a distinct ligament groove on the posterior medial surface of the femoral head, directly comparing it to the condition in *Allosaurus*. A groove like this is common in allosauroids and for example present in *Concavenator* or *Mapusaurus* [12, 67].

The lesser trochanter of SNSB-BSPG 1922 X 46 is well developed and clearly separated from the greater trochanter by a distinct cleft. This is most similar to the condition of basal carcharodontosaurids like *Concavenator* and *Tyrannotitan* [50, 67] or allosauroids like *Allosaurus* or *Sinraptor* [45, 60] and differs from the reduced lesser trochanters of *Giganotosaurus* and *Mapusaurus* (MUCPv-Ch1; [12]). Stromer [1] described the accessory trochanter as an approximately rectangular process projecting anteriolaterally. This very prominent spike-like accessory trochanter is visible in all of Stromer's drawings of the femur ([1]: pl I, Fig 14A; [25]: Fig 9D), but especially conspicuous in the photograph of the exhibited specimen and Stromer's 1936 reconstruction. A similarly prominent, triangular accessory trochanter is only present in *Neovenator* [58] and *Acrocanthosaurus* (NCSM 14345), and absent in other allosauroids, such as *Allosaurus* and *Sinraptor* [45, 60], or other derived carcharodontosaurids like *Mapusaurus* [12] and *Giganotosaurus* [MUCPv-Ch1]. The fourth trochanter was well-developed as an elongate, triangular posteromedial flange, as is clearly visible in the exhibited specimen (Figs 2 and

12B). A well-developed fourth trochanter is present in most tetanurans but is reduced in most coelurosaurs [72] and some carcharodontosaurids like *Mapusaurus* and *Giganotosaurus* [50].

The distal femur was wider lateromedially than anteroposteriorly. Stromer [1] described the anteromedial surface of the distal femur as deeply concave and bordered by a well-developed mediodistal crest. This feature is also present in *Allosaurus* [69], as already mentioned by Stromer [1], and commonly present in other allosauroids, such as *Giganotosaurus* (MUCPV-Ch1), *Tyrannotitan*, or *Meraxes* [11, 50]. The anterior surface of the lateral distal condyle is described by Stromer [1] as vertically striated. Distally, the lateral condyle projected distinctly more distally than the medial one. The condition present in SNSB-BSPG 1922 X 46 was very similar to *Meraxes* [11], being not as pronounced as in *Giganotosaurus* (MUCPv-Ch1), but more developed than in *Tyrannotitan* or *Mapusaurus* [12, 50]. The anterior extensor groove on the femur of BSPG SNSB 1922 X 46 was deep and narrow, once again resembling the condition present in *Meraxes* [11] and differing from other carcharodontosaurids such as *Mapusaurus*, or *Acrocanthosaurus* where this groove is comparatively wider (MCF-PVPH-108.203; NCSM 14345).

*Fibula*. In comparisons with other carcharodontosaurids, the fibula of SNSB-BSPG 1922 X 46 was rather gracile. With a ratio of maximum length to anteroposterior shaft width of 16, it was similar in elongation to *Meraxes* (16.8) (MMCh-PV 65) and more gracile than the fibulae of *Mapusaurus* (13.2) or *Giganotosaurus* (10.4) [12], but more robust than the fibula of *Acrocanthosaurus* (20.0) [73]. The ratio of fibula length to femur length is 1.4 for SNSB-BSPG 1922 X 46, which is in line with similarly large carcharodontosaurids, with the exception of *Giganotosaurus*, which has a comparatively shorter fibula with a ratio of 1.7 (MUCPv-Ch1). Similar to *Giganotosaurus*, the fibula of SNSB-BSPG 1922 X 46 is strongly concave in proximal view and constant in thickness (Fig 12F), differing from the less concave and posteriorly tapering fibulae of *Tyrannotitan*, *Allosaurus* or *Neovenator* [45, 50, 58]. As noted by Stromer [1], and visible in the exhibited specimen, the proximal fibula lacks a strong posterior expansion in lateral view, being more notably anteriorly than posteriorly expanded (Figs 2 and 12D). In contrast, in most other allosauroids, such as *Giganotosaurus* (MUCPv-Ch1), *Allosaurus*, *Sinraptor* or *Neovenator*, as well as the vast majority of other theropods, the fibula is more strongly posteriorly than anteriorly expanded proximally [45, 58, 60]. *Mapusaurus* shows a similar lack of a pronounced posterior expansion [12]. Stromer [1] described a medially projecting "corner" on the anteroproximal margin of the fibula. While this "corner" is visible in his drawings (Fig 12C), it is not clear if this was developed as a distinct process, or as a flange, since the fibula was damaged directly proximal to it.

The attachment for the m. iliofibularis was developed as a pronounced, flange-like rugosity slightly projecting anterolaterally from the proximal half of the fibular shaft. Similar to *Concavenator* [67] and *Giganotosaurus* (MUCPv-Ch1), it differed from the strongly rugose flange present in *Meraxes* (MMCh-PV 65), yet was more strongly developed than the tubercles of *Mapusaurus* and *Tyrannotitan* [12, 50]. According to Stromer's [1] description, the medial surface of the fibular shaft was flat in its proximal most third. Distal to that it was medially concave up until its distal end, resulting in a crescent shaped cross section at its halfway point, as visible in Stromer's drawing (Fig 12G).

The distal end of the fibula was expanded for articulation with the astragalus and calcaneum. Stromer [1] described a groove on its anterior margin of the distal end for articulation with the ascending process of the astragalus.

## Phylogenetic results

**General results.** In general, both split and merged analyses return similar results, when the same character weight is used. The main difference between the two different datasets is in

regards to the position of the fused/split taxa within the tree, although other minor effects on ingroup relations within specific clades can also be observed. All reduced strict consensus trees are provided within the Supplement (S1 File). Scores between split and merged weight analyses are very similar, with the split datasets returning slightly better values (Table 2). Interestingly, constraining the taxa of the split dataset to resemble the taxa used in the merged dataset ("merged constraint") resulted in a tree with a score of 6493 steps, four more than the actual merged dataset. Merged and "merged constraint" analysis only differ slightly in the position of some of the pruned taxa. Results across analyses with different weights are also generally similar and mostly differ in ingroup relationships and the positions of individual taxa, with trees with a stronger concavity generally showing better resolved ingroup relations. The better resolution is usually in congruence with one of the positions a taxon would have that has been pruned from a run with less severe weighting. For description of synapomorphies and character evolution, as well as general tree morphology, we will be referring to the trees obtained by IterPcr using a weight of $K = 12$ (Fig 13), as it was shown that this value performs best, especially in larger datasets [74]. The phylogeny of the more fragmentary and unstable taxa, as well as the positions and effects of split taxa, are initially disregarded here, as their position will be highlighted in the discussion. Similarly, if there are strongly different results for different weights, they will be mentioned. For a detailed list of unambiguous synapomorphies within the groups relevant for this paper see S2, S3 Tables and S1 Fig.

Regardless of the dataset and weight used, SNSB-BSPG 1922 X 46 is recovered as a derived carcharodontosaurid outside of Carcharodontosaurinae. This is the case for all datasets with the exception of one (Split equal weight), where the interrelations of Carcharodontosaurinae could not be resolved. In the split $K = 12$ set the Egyptian specimen is recovered within the Caracharodontosauria based on the following unambiguous synapomorphies: Maxilla with fused interdental plates (also present in ceratosaurs e.g. *Ceratosaurus*, *Majungasaurus* [46, 75] and some other Tyrannosauroid taxa *Fukuiraptor*, *Australovenator*, *Shaochilong* [5, 76, 77]); frontal firmly fused to parietal (outside the clade only present in abelisauroids e.g. *Majungasaurus*, *Carnotaurus* [46, 78] and a few other taxa like *Zuolong* and *Shaochilong* [5, 79], as well as in birds); femur with dorsomedially oriented femoral head (Also independently present in some Tyrannosauroids like *Australovenator*, *Tyrannosaurus* [80, 81] and in therizinosaurs e.g. *Nothronychus*, *Jianchangosaurus* [82, 83]). The following characters also support this position, but are not unambiguous synapomorphies for the clade, as they are unknown in some of the basal taxa: nasals of subequal width throughout their length (In non carcharodontosauroid Allosauroids and *Acrocanthosaurus* (NCSM 14345) the nasals widen posteriorly e.g. *Sinraptor*, *Allosaurus* [45, 60] and *Asfaltovenator* (MPEF-PV 3440), while it is also present in some tyrannosauroids like *Tyrannosaurus*, *Tarbosaurus*, or *Megaraptor* [36, 84] and in some ceratosarus like *Ceratosaurus*, *Majungasaurus* or *Carnotaurus* [46, 75, 78]); interfrontal suture completely fused (Outside this clade only present in ceratosaurs e.g. *Ceratosaurus*, *Majungasaurus* [46, 75] *Tyrannosaurus*, *Shaochilong* [5, 80] and birds); anteromedial corner of supratemporal fossa roofed by shelf of frontal and parietal (outside of Carcharodontosauria only present in *Pivetausaurus* (MNHN 1920–7)); horizontal ridge extending from the dorsal edge of the paroccipital process medially above the foramen magnum, developed as a sharp crest (This ridge is usually absent, or only developed as a small swelling e.g. *Sinraptor*, *Allosaurus*, *Tyrannosaurus* [45, 80, 85] but also sharply developed in *Ekrixinatosaurus* (MUCPv-294)); lateral distal condyle of femur projects further distally than medial distal condyle (outside of carcharodontosauria this is only present in a hand full of other basal Tetanurans like *Australovenator* and *Leshansaurus* [81, 86]). With Carcharodontosauridae it shares the following unambiguous synapomorphy in the Split dataset: Ilium and ischium with peg and socket articulation (also present in tyrannosauroids like *Tyrannosaurus* and *Alioramus* [49, 80] and ceratosaurs, like *Ceratosaurus* (BYU

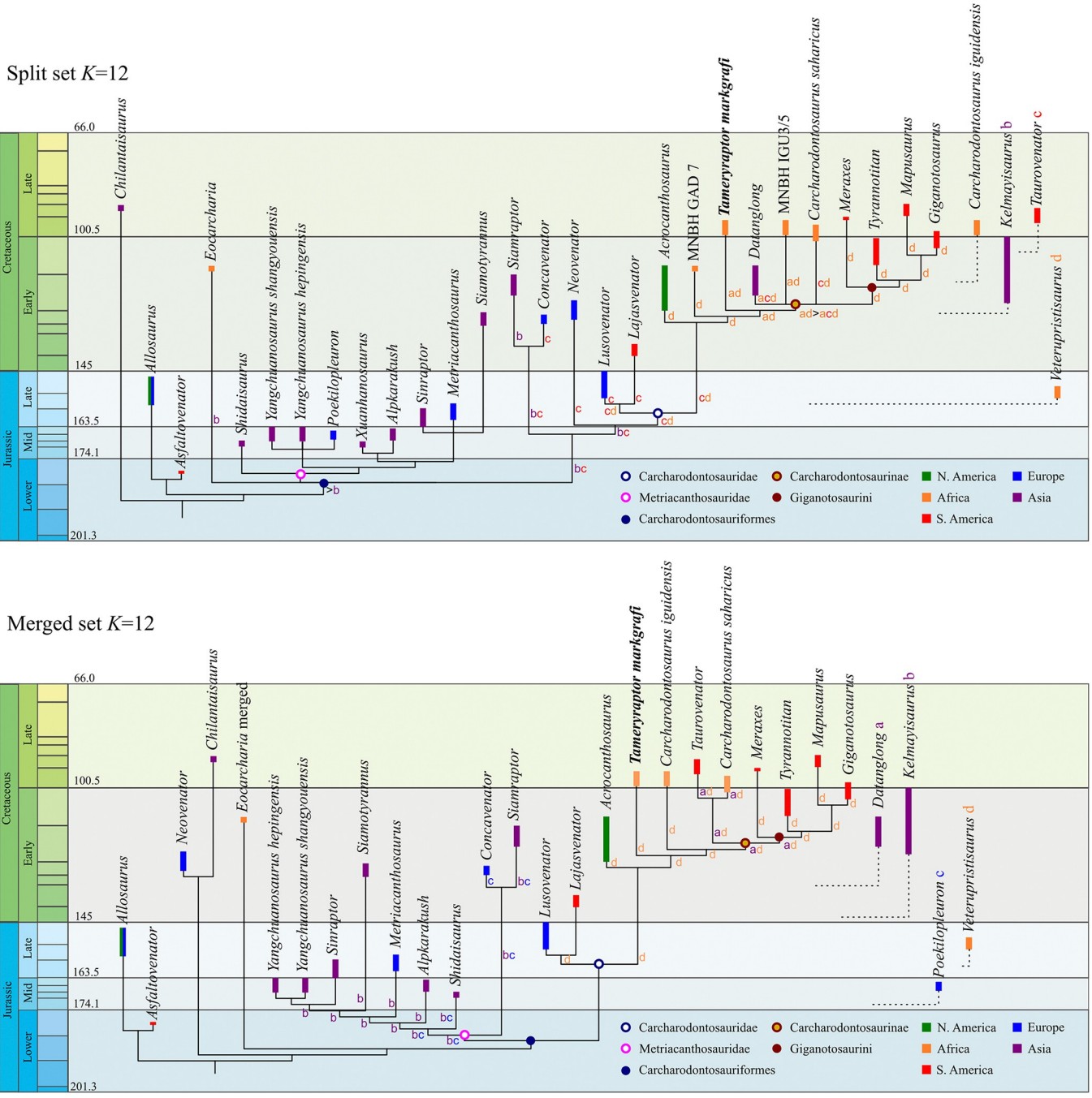

**Fig 13. Time calibrated reduced strict consensus tree.** Analyses done with *K* = 12 values.

12893)). In the Merged dataset it shares the following unambiguous synapomorphies: Maxilla with extensive grooved sculpturing on the main body (also shared with tyrannosauroids like *Tyrannosaurus* or *Tarbosaurus* [84] and abelisauroids such as *Majungasaurus* or *Carnotaurus* [46, 78]). It also shares the following ACCTRAN character transformation present for Charcharodontosauridae in both datasets: Fibula with anterior portion of proximal end subequal or greater in mediolateral width than posterior portion. SNSB-BSPG 1922 X 46 is recovered

**Table 2. Results of the different analyses.** For an explanation of the different constraint types see Materials and Methods.

| Dataset | Weight | Constraints | Score |
|---|---|---|---|
| Split | equal | no | 6488 |
| Split | equal | "*saharicus*" | 6490 |
| Split | equal | "*Carcharodontosaurus*" | 6491 |
| Split | equal | "merged" | 6493 |
| Split | equal | "*Shaochilong*" | 6492 |
| Split | equal | "*Datanglong* Megaraptora" | 6490 |
| Split | equal | "*Mapusaurus-Taurovenator*" | 6489 |
| Split | $K = 15$ | no | 235.50767 |
| Split | $K = 12$ | no | 272.70441 |
| Split | $K = 9$ | no | 324.21003 |
| Merged | equal | no | 6489 |
| Merged | equal | "*saharicus*" | 6490 |
| Merged | equal | "*Carcharodontosaurus*" | 6490 |
| Merged | equal | "*Shaochilong*" | 6493 |
| Merged | equal | "*Datanglong* Megaraptora" | 6491 |
| Merged | equal | "*Mapusaurus-Taurovenator*" | 6490 |
| Merged | $K = 15$ | no | 235.71117 |
| Merged | $K = 12$ | no | 272.71352 |
| Merged | $K = 9$ | no | 324.21469 |

outside of Carcharodontosaurinae because it lacks the following characters that are present within the group in both analyses: Nasals with prominent lateral shelf overhanging the antorbital fossa; frontal with dorsally vaulted surface across the frontal-prefrotnal contact; ossified interorbital septum; maxillary and dentary teeth with mesial carinae extending to the base of the tooth crown.

All specimens previously referred to the genus *Carcharodontosaurus* are here found in various derived positions within, and outside of Carcharodontosaurinae. A conclusive sister group relationship was not found for any of these specimens in any of the analyses. Constraining SNSB-BSPG 1922 X 46 into a sister group relation to *Carcharodontosaurus saharicus* requires two additional steps in the Split and one additional step in the Merged dataset. Running the "*Carcharodontosaurus*" constraint added one further step in the Split and no additional steps in the Merged dataset (Table 2).

**The phylogeny of Allosauroidea.** None of the analyses recovered a monophyletic Carnosauria as seen in Rauhut & Pol [87]; instead Allosauroidea, as defined by Rauhut and Pol [87], is consistently recovered as the sister group to Coelurosauria, as was recovered in previous studies (e.g.[61, 88]). In all of our analyses, Allosauroidea consists of Allosauridae, Metriacanthosauridae and Carcharodontosauridae, sometimes with some additional taxa placed outside of or in between these clades. However, some taxa or clades that were recovered as Allosauroids in other analyses were consistently found outside that clade.

Piatnitzkysauridae are recovered outside of Allosauroidea, although their position is quite variable, with the clade mostly being recovered within Megalosauroidea (Split: $K = 15$, $K = 12$ $K = 9$, Merged: $K = 12$, $K = 9$), or in variable positions among the base of Tetanurae (Split: equal weight, Merged: equal weight, $K = 15$).

In all Split and Merged runs Megaraptora is consistently recovered within the Tyrannosauroidea. These results mirror the more recent consensus on the position of Megaraptora, which seems to suggest a tyrannosauroid identity for the group [36, 89, 90], as opposed to the initial

identification as a part of Allosauroidea [61]. In both $K = 12$ datasets, Megaraptora is recovered within the Tyrannosauroidea, based on the following synapomorphies shared between the two datasets: Lacrimal with posterodorsal process present and much shorter than anterior process (a posterodorsal process is absent in most Allosauroids, e.g. *Acrocanthosaurus*, *Concavenator* and *Allosaurus* ([43–45] with the exception of metriacanthosaurids, where it is also short e.g. *Sinraptor*, *Yangchuanosauurs* [60, 91]. A similar condition is also observed in ceratosaurs like *Eoabelisaurus*, *Ceratosaurus* or *Majungasaurus* [46, 75, 92]); sagittal crest extending onto the frontals (This is absent in all allosauroids); mandible retroarticular process shorter than antero-posterior length of mandibular glenoid. Other characters generally present in Tyrannosauroidea and shared with Megaraptora in both sets are: Anteroposterior length of maxillary-jugal contact relative to total maxilla length less than 25%; basipterygoid processes about as long anteroposteriorly as proximodistally; basisphenoid recess partially divided into left and right compartment by a thin lamina; longitudinal groove on the ventral side of the cultriform process absent; ilium and ischium with peg and socket articulation.

Allosauridae is made up of *Allosaurus* and *Asfaltovenator*, though the latter is not always recovered in the Allosauridae, and instead sometimes (Merged: equal weight, $K = 15$; Split: equal weight) recovered within close proximity at the base of Allosauroidea, congruent with the results of Rauhut and Pol [87]. *Yunyangosaurus* is also recovered in Allosauridae in some analyses (Merged $K = 9$, Split $K = 9$), although it is recovered within the Megalosauridae or as a basal tetanuran in the other analyses.

Although the ingroup relations of the Metriacanthosauridae are insufficiently resolved, all analyses recover the Metriacanthosauridae within Allosauriodea more closely related to Carcharodontosauridae, than to *Allosaurus*. In all but one analysis (split $K = 12$) Metriacanthosauridae is recovered as the sister group to the Carcharodontosauria, or Carcharodontosauridae. In this last analysis, it is recovered in a polytomy with Carcharodontosauria and *Eocarcharia*, which is likely due to the uncertainty of the position of the latter taxon. So far, no name exists for this clade. The synapomorphies for this clade will be elaborated on in the discussion.

A Carcharodontosauria, as defined by Benson et al. [61], is recovered in some (Split: equal weight, $K = 15$, $K = 12$ and Merged: equal weight, $K = 15$), but not in all of our analyses. In the other analyses (Split: $K = 9$, and Merged: $K = 12$, $K = 9$), *Neovenator* is recovered in a more basal position within Allosauroidea, together with its then sister taxon *Chilantaisaurus*. Carcharodontosauridae is recovered within all analyses, sharing the following unambiguous synapomorphies in both $K = 12$ analyses: Foramina in cortical surfaces of centra of sacral vertebrae present in some centra; mid-caudal vertebrae, anterior entrance of neural canal funnel like (outside Carcharodontosauria this is only present in *Dubreuillosaurus* [93]); neural spines on distal caudals from a low ridge. The ingroup relationships within Carcharodontosauridae are relatively consistent, with only *Taurovenator* and *Kelmayisaurus* being variably recovered within or without the group (Table 3).

The early branching carcharodontosaurids *Lusovenator* and *Lajasvenator* are consistently recovered as sister taxa in all analyses, regardless of the weight used. Similarly, *Siamraptor* and *Concavenator* are recovered as sister taxa in all analyses, with the exception of the equal weight analyses, where their exact position could not be determined. *Siamraptor* and *Concavenator* are recovered as basal members of Carcharodontosauria in all analyses that recovered this clade. In datasets that don't recover this clade, they are instead recovered in a basal position within the Metriacanthosauridae.

*Acrocanthosaurus* is consistently recovered as a carcharodontosaurid and sister taxon to the clade consisting of Carcharodontosaurini, *Carcharodontosaurus iguidensis*, its referred material and SNSB-BSPG 1922 X 46. In the Split (equal weight, $K = 15$, $K = 12$, $K = 9$) datasets, this clade also includes the referred maxilla material of *Eocarcharia* (MNBH GAD 7).

**Table 3. Position of carcharodontosaurid taxa within Carcharodontosauria in individual runs: Note that Carcharodontosauria is recovered in 5/8 runs and "split" and "merged" taxa are only counted for the respective runs, so have a maximum of 4 occurrences.**

| | Carcharodontosauria 5/8 | Carcharodontosauridae 8/8 | Carcharodontosaurinae 8/8 | Giganotosaurini 8/8 |
|---|---|---|---|---|
| **Always included** | *Acrocanthosaurus*<br>*Carcharodontosaurus iguidensis*<br>MNBH IGU3/5<br>*C. iguidensis* (merged)<br>*C. saharicus*<br>*Datanglong*<br>*Giganotosaurus*<br>*Lajasvenator*<br>*Lusovenator*<br>*Mapusaurus*<br>*Meraxes*<br>*Neovenator*<br>***Tameryraptor***<br>*Tyrannotitan*<br>*Veterupristisaurus* | *Acrocanthosaurus*<br>*C. iguidensis*<br>MNBH IGU3/5<br>*C. iguidensis* (merged)<br>*C. saharicus*<br>*Datanglong*<br>*Giganotosaurus*<br>*Lajasvenator*<br>*Lusovenator*<br>*Mapusaurus*<br>*Meraxes*<br>***Tameryraptor***<br>*Tyrannotitan*<br>*Veterupristisaurus* | *C. saharicus*<br>*Giganotosaurus*<br>*Mapusaurus* | *Giganotosaurus*<br>*Mapusaurus* |
| **Sometimes included** | *Eocarcharia*: 0-2/4<br>*Eocarcharia* (merged): 0-1/2<br>*Kelmayisaurus*: 2-5/5<br>*Poekilopleuron*: 0-2/5<br>*Taurovenator*: 0-2/5 | MNBH GAD 7: 3/4<br>*Kelmayisaurus*: 0-1/8<br>*Taurovenator*: 4-8/8 | *C. iguidensis*: 0-2/4<br>MNBH IGU 3/5: 0-2/4<br>*C. iguidensis* (merged): 0-1/4<br>*Datanglong*: 0-8/8<br>*Meraxes*: 7-8/8<br>***Tameryraptor***: 0-1/8<br>*Taurovenator*: 4-8/8<br>*Tyrannotitan*: 7-8/8<br>*Veterupristisaurus*: 0-8/8 | *Datanglong*: 0-8/8<br>*Meraxes*: 7-8/8<br>*Tyrannotitan*: 7-8/8<br>*Veterupristisaurus*: 0-8/8 |
| **Always excluded** | MNBH GAD 7 | *Eocarcharia*<br>*Eocarcharia* (merged)<br>*Neovenator*<br>*Poekilopleuron* | *Acrocanthosaurus*<br>*Eocarcharia*<br>MNBH GAD 7<br>*Eocarcharia* (merged)<br>*Kelmayisaurus*<br>*Lajasvenator*<br>*Lusovenator*<br>*Neovenator*<br>*Poekilopleuron* | *Acrocanthosaurus*<br>*C. iguidensis*<br>MNBH IGU 3/5<br>*C. iguidensis* (merged)<br>*C. saharicus*<br>*Eocarcharia*<br>MNBH GAD 7<br>*Eocarcharia* (merged)<br>*Kelmayisaurus*<br>*Lajasvenator*<br>*Lusovenator*<br>*Neovenator*<br>*Poekilopleuron*<br>***Tameryraptor***<br>*Taurovenator* |

The subdivision of the clade within Carcharodontosaurinae and Giganotosaurini is recovered in all runs, although the content of these groups can once again vary (Table 3). Carcharodontosaurinae do not share any unambiguous synapomorphies between both $K = 12$ analyses, but other characters are present in the group for both analyses: Nasals with a prominent lateral shelf overhanging the antorbital fenestra; frontal, dorsal surface vaulted across frontal-prefrontal suture (This is only present in *Carcharodontosaurus saharicus*, *Giganotosaurus* and *Meraxes* (SGM-Din 1, MUCPv-Ch1, MMCh-PV 65)); infracondylar fossa of occipital condyle absent; ossified mesethmoid in adults (also independently present in abelisauroids like *Majungasaurus* or *Carnotaurus* [46, 78]); maxillary and dentary teeth, mesial carina extending to the base of tooth crown (Within Allosauroidea this is also present in metriacanthosaurids e.g. *Sinraptor* [56]). Giganotosaurini always includes *Giganotosaurus* and *Mapusaurus* as sister taxa. With the exception of one run (Split: equal weight), *Tyrannotitan* is recovered as the sister taxon to the two and *Meraxes* as sister taxon to the group formed by the former three taxa. These results are generally in line with previous results [11, 50]. Giganotosaurini share the following characters between both split and merged $K = 12$ datasets: Maxilla, palatal shelf extending posteriorly

below the antorbital fenestra; prootic with two separate exits for the facialis nerve; otosphenoidal crest on the prootic discontinuous between margin of stapedial groove and anterodorsal margin of anterior tympanic recess.

## Discussion

### The taxonomic distinction of SNSB-BSPG 1922 X 46 from *Carcharodontosaurus* and its implications

As pointed out in the introduction, Stromer [1] believed his theropod specimen SNSB-BSPG 1922 X 46 to represent the same taxon as the teeth originally described as *Megalosaurus saharicus* by Depéret and Savornin [14]. The latter specimens were found close to Timimoun, Algeria, in sediments later assigned to the "Continental Intercalaire" (e.g. [94]) and believed to be Albian in age by Depéret and Savornin [14, 15]. The two teeth described by Depéret and Savornin [14, 15] came from a trench excavated for the water supply of Timimoun, and were found at a depth of c. 36 m below the surface and some 3 km apart. In the original short communication, Depéret and Savornin [14] referred these teeth to the genus *Megalosaurus*, as the new species, *Megalosaurus saharicus*. In their more detailed account [15], they discussed the then known stratigraphic range of *Megalosaurus* and noted that the question of the phylogenetic relationship between the different species was made difficult by the fact that most of these species were based on isolated teeth. However, they noted some differences between teeth of different ages referred to this genus, and suggested a division of *Megalosaurus* into three genera (or sub-genera; this is not entirely clear from the paper) and thus referred their African taxon to the genus *Dryptosaurus*, as *Dryptosaurus saharicus*, apparently mainly on stratigraphic considerations, as the teeth of the actual *Dryptosaurus* (= *Albertosaurus*) are considerably more robust.

Stromer [1] pointed out important differences of the skeletal remains described by him from other theropods known at the time and argued that these differences justified the erection of a new genus, which he named *Carcharodontosaurus*. However, he also noted the close similarity of the teeth of his Egyptian specimen to those described by Depéret and Savornin [14, 15] and thus explicitly referred his material to the same species, as *Carcharodontosaurus saharicus*. After the destruction of the Egyptian specimen, the name *Carcharodontosaurus saharicus* was usually used in reference to this material, and few additional specimens were referred to this species, with the exception of isolated teeth, generally from northern Africa. A notable exception was the referral of isolated teeth and bones to *Carcharodontosaurus saharicus* by Lapparent [94]. This material, which included a cranial fragment, teeth, a poorly preserved cervical vertebra, several supposed dorsal, sacral and caudal vertebrae, remains of a humerus and a manual ungual, and a metatarsal and some pedal unguals, came from different localities from all over western Africa and showed very limited overlap with the materials described by Stromer [1]. The main reason for referral to the same taxon were obviously the teeth and the size of the remains, but no detailed taxonomic evaluation was presented by Lapparent [94]. However, for several of the specimens described by Lapparent [94], even the identification as theropod remains is questionable (e.g., the caudal vertebrae figured by Lapparent, [94], pl. VI, Fig 5 and pl. VII, Fig 6, rather seem to be sauropod elements), and, also due to the lack of overlap with the material described by Stromer [1], the referral to the same taxon seems unjustified; the material described by Lapparent [94] is in need of revision.

More than 50 years after the destruction of the Egyptian material, Sereno et al. [7] described a partial skull of a large theropod from the "Kem Kem Beds" of Morocco (now referred to the Douira Formation of the Kem Kem Group [18]) and also referred it to the species *Carcharodontosaurus saharicus*. The primary reason for the referral to this species was again the

morphology of the teeth, although Sereno et al. ([7]: 987) also noted similarities to the material described by Stromer [1]), such as a strongly developed ridge below the antorbital fossa and the sculpting of the maxilla and rugosities of the nasal. However, discoveries of more carcharodontosaurid taxa in the past 30 years have shown that the morphology of the teeth is not unique to the species *Carcharodontosaurus saharicus* [14], as teeth with the same characteristics are also found in *Giganotosaurus* (MUCPv-Ch1; [13]), *Mapusaurus* [12] and *Tyrannotitan* [50]. This was pointed out by Brusatte and Sereno [8], who also noted that the teeth described by Depéret and Savornin [14, 15] seem to be lost, and consequently designated the Moroccan skull as neotype for *Carcharodontosaurus saharicus*. However, these authors only noted that "a well preserved skull with teeth from Morocco . . . is identical to both the original holotypic teeth and Stromer's Egyptian material" (p. 904), but did not comment on any specific similarities between the neotype and the Egyptian material.

More recently, Ibrahim et al. [18] presented additional comparisons between the Moroccan skull and the Egyptian material. Apart from the characters already mentioned by Sereno et al. [7], they noted several other characters that supposedly united the two north African specimens from other carcharodontosaurids known at the time, including the prominence of the nuchal crest, the shape of the posterior parietal process, the width of the supraoccipital crest, the particularly deep maxilla, the position of the anteromedial process of the maxilla high on its medial side, a suture with the premaxilla that slopes at approximately 65˚ to the alveolar margin and is exposed in medial view, a rounded and everted rim of the maxillary antorbital fossa, ventrally directed linear rugosities on the lateral surface of the maxilla and a nasal that is gently waisted at midlength, has an extensive narial fossa anteriorly, and marked rugosities posteriorly [18]: 162–165. However, all of these characters are problematic, as they have a wider distribution within carcharodontosaurids or allosauroids in general, are poorly defined, or cannot be verified for SNSB-BSPG 1922 X 46 on the basis of the information published by Stromer [1]. Thus, for example, there is no marked difference in the development or height of the medial knob-like expansion of the nuchal crest in the neotype of *Carcharodontosaurus saharicus* (SGM-Din 1), the Egyptian specimen, the type of *Giganotosaurus* (MUCPv-CH-1), or *Meraxes* (MMCh-PV 65), and also the inclination of the parietal nuchal crest seems to be the same. As for the posterior process of the parietal, this was said to be triangular in the Moroccan skull and SNSB-BSPG 1922 X 46, but tongue-shaped in *Giganotosaurus*. However, this process is damaged in *Carcharodontosaurus saharicus* (SGM-Din 1), and nothing can be said about its exact shape in the Egyptian specimen on the basis of the drawings in the original description of Stromer (Fig 6). The supraoccipital crest is narrow in the Moroccan and Egyptian specimens, but this is also the case in *Meraxes* (MMCh-PV 65) and other allosauroids, such as *Allosaurus* [69] or *Sinraptor* [60], and the dorsally mediolaterally expanded crest in *Giganotosaurus* is probably an autapomorphy of this taxon [52]. Concerning the supposed similarities in the maxilla, the premaxillary contact was not preserved in the Egyptian specimen. Furthermore, both its angle and medial exposure in *Mapusaurus* is comparable to the situation in the neotype of *Carcharodontosaurus saharicus* (MCF-PVPH-108.169; [51]); the maxilla of *Giganotosaurus* is too poorly preserved to say anything about its morphology in this area. Likewise, the depth of the maxilla does not seem to be significantly different in *Carcharodontosaurus* and *Mapusaurus* (MCF-PVPH-108.169), whereas the maxilla of the Egyptian specimen seems slightly more slender and more comparable to that of *Carcharodontosaurus iguidensis* [8], or the recently described *Meraxes* [11]. The anteromedial process is placed at the dorsal rim of the anterior end of the maxilla in all carcharodontosaurids, but nothing can be said about its position or shape in the Egyptian specimen, as it was not preserved. An everted, rounded ridge bordering the antorbital fossa is variably present in carcharodontosaurids, including *Meraxes* (MMCh-PV 65), a juvenile maxilla of *Mapusaurus* (MCF-PVPH-

108.115; [12]), and *Carcharodontosaurus iguidensis* [8], although it is not as strongly developed as in *Carcharodontosaurus saharicus* or SNSB-BSPG 1922 X 46. However, whereas this ridge has its strongest expansion in the anterior part of the antorbital fossa in *Carcharodontosaurus saharicus*, Stromer [1] mentioned that its lateral expansion increased posteriorly in the Egyptian specimen, and the ridge does not seem to be any more conspicuous in its anterior portion than in *Meraxes* [11]. Likewise, ventrally directed linear rugosities are present on the lateral surface of the maxilla in *Giganotosaurus* (MUCPv-Ch1), *Meraxes* (MMCh-PV 65), and *Carcharodontosaurus saharicus* [18], but in none of these taxa are they as strongly developed as in the Egyptian specimen. Concerning the nasal, Ibrahim et al. [18] state that the nasals of *Carcharodontosaurus saharicus* and the Egyptian specimen showed a gentle constriction at midlength in lateral view. However, this is probably an error, as no lateral view of the nasal of SNSB-BSPG 1922 X 46 was included in Stromer's [1] figures, and the photograph of the specimen shows that such a constriction was absent. What could be interpreted as a subtle constriction in Stromer's 1936 sketch (Fig 5), is the emargination of the antorbital fossa, which is laterally covered by a bony lamina in *Carcharodontosaurus saharicus*. Thus, Ibrahim et al. [18] probably meant a constriction of the bone in dorsal view, which is present for both specimens. However, a constriction of the nasal in dorsal view is also present in *Meraxes* (MMCh-PV 65) and in the preserved portion of the nasals of *Mapusaurus* [12]. For *Giganotosaurus* this can not be validated due to the incompleteness of the preserved nasal. The presence of a marked supranarial fossa in *Carcharodontosaurus saharicus* rather seems to be a difference from the Egyptian specimen, as such a fossa is not indicated in Stromer's [1] drawings and not visible in the photograph of the specimen. In the description, Stromer only mentions a slight concavity in this position, but both the drawings and the photograph of the specimen do not show any indication of the marked rim seen in *Carcharodontosaurus saharicus* [18]. Furthermore, a marked, sharply rimmed supranarial fossa, as present in the latter taxon, is also present in other Allosauroids, such as *Asfaltovenator* (MPEF-PV 3440), *Sinraptor* [60] or *Acrocanthosaurus* (NCSM 14345). Finally, nasal rugosities are present in all derived carcharodontosaurids, although a nasal crest, or horn, is so far only known in SNSB-BSPG 1922 X 46. Thus, none of the characters put forward so far are evidence for a referral of the Moroccan skull and the Egyptian specimen to the same species or even genus.

On the contrary, it is clear, from the detailed description and the comparisons above and from the results of the phylogenetic analysis, that the Egyptian carcharodontosaurid represents a distinct and previously unrecognized taxon, differing from the Moroccan specimen (SGM-Din 1) in numerous characters, including the ventromedially inflected medial nasal articulations, absence of a strongly demarcated narial fossa on the nasals, presence of a nasal crest or horn, the emargination of the antorbital fossa being visible in lateral view on the nasals, the more pronounced rugosities in both the nasals and maxilla, more symmetrical maxillary teeth, a less expanded prefrontal contact on the frontals, a vaulted expansion on the midline of the frontals, and a relatively more enlarged cerebrum (Fig 14). Note, while some of these characters, especially those linked to possible display structures, like the presence of a nasal horn or the more pronounced rugosities, could possibly be the result of sexual dimorphism, no such form of dimorphism is known from theropods with comparable cranial ornamentations (like *Allosaurus*, *Ceratosaurus* or *Majungasaurus*), where multiple specimens are known. Recently, this exact issue was also brought forward by Pol et al. [95] in their paper describing the South American abelisaurid *Koleken*, as one of the features distinguishing it from the contemporary *Carnotaurus* is the absence of frontal horns. We come to the same conclusion as these authors in our case given that "To date, date, presence or absence of horns have not been documented as a case of sexual dimorphism in archosaurs, either extinct or extant [. . .]" and similar to the case in *Koleken* and *Carnotaurus*, the differenced between SNSB-BSPG 1922 X 46 and

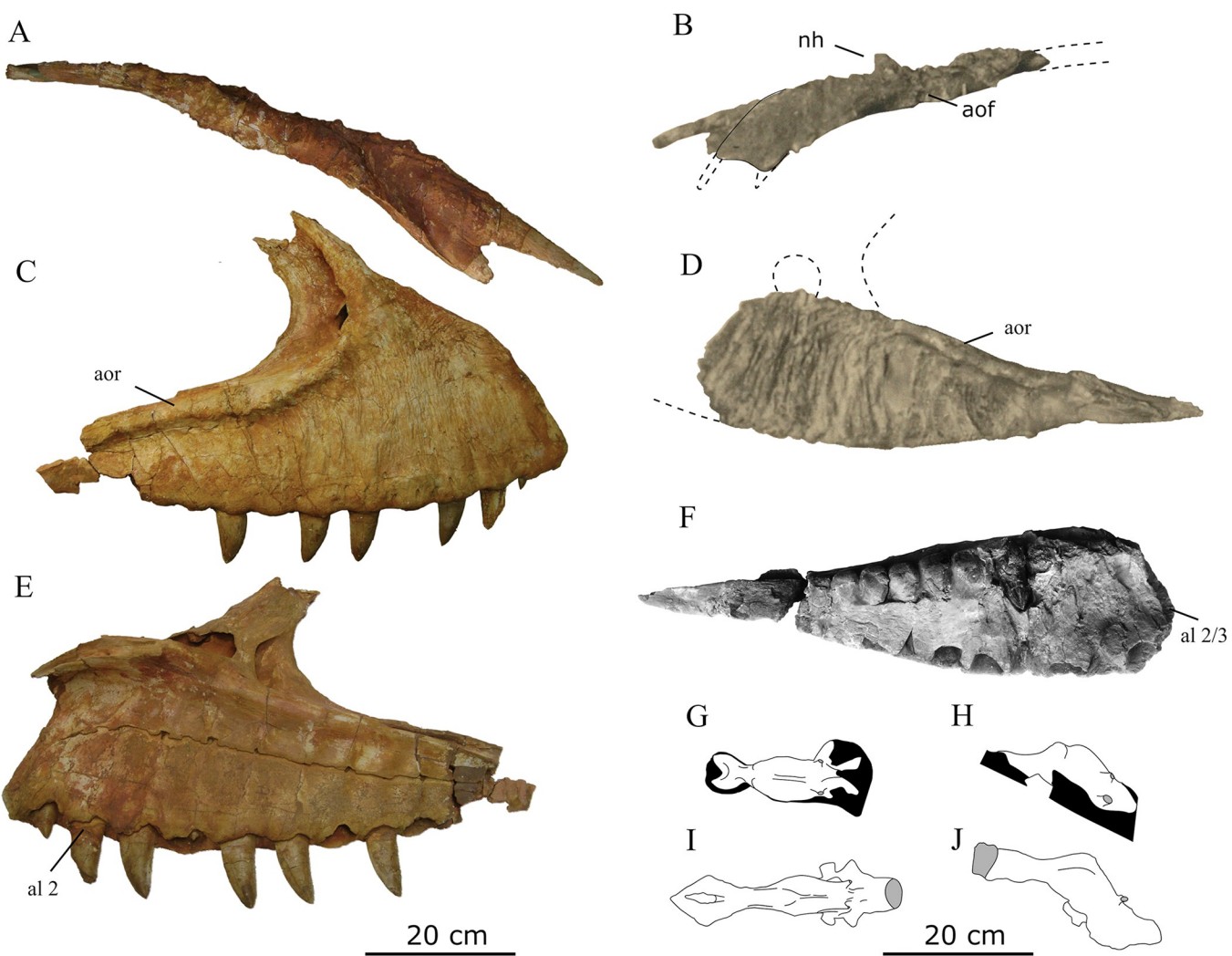

**Fig 14. Comparison between SNSB-BSPG 1922 X 46 and SGM-Din 1.** SNSB-BSPG 1922 X 46: (**B**) nasalia from left lateral, left maxilla from (**D**) lateral and (**F**) medial and endocast in (**G**) dorsal and (**H**) left lateral view; (**A**, **C**, **E**, **I**, **J**) and SGM-Din1: (**A**) right nasal in lateral view, right maxilla in (**C**) lateral and (**E**) medial view and endocast in (**I**) dorsal and (**J**) left lateral view; Sources: (**A**, **C**, **E**, **J**) [18], (**I**) modified from [54], (**B**, **D**) UAT 678/20/SNSB-BSPG 1922 X 46, (**F**) [1], (**G**, **H**) MB. R. 2056.

*Carcharodontosaurus saharicus* are not limited to the absence/presence of these possible display structures.

Stromer never formally designated SNSB-BSPG 1922 X 46 as the neotype specimen of *Megalosaurus saharicus*. Instead he referred his material to the same species as the Algerian teeth, changed the genus name and gave an emended diagnosis [1, 25]. In both his 1931 and 1934 papers he specifically noted, that he considered the smaller of the two teeth described by Depéret and Savornin ([15]: p. 259–260, pl. 12 Fig 1) to be the type specimen of *Carcharodontosaurus saharicus* ([1]: p. 9, [22]: p. 62). Thus, this tooth should be considered the lectotype of *Carcharodontosaurus saharicus* in accordance with ICZN article 74.5, and the species *Carcharodontosaurus saharicus* (Depéret and Savornin, 1925) has to be considered as the type species of the genus *Carcharodontosaurus*, in accordance with ICZN article 68.3. The genus and species name thus remains with the now lost Algerian teeth. In 2007 the Moroccan cranium was designated the neotype for *Carcharodontosaurus saharicus* by Brusatte and Sereno [8]. As

Brusatte and Sereno's nomenclatorial act fulfills the requirement of a neotype designation according to ICZN article 75.3, and this constitutes the first neotype designation, it would take priority over any possible neotype designation of the Egyptian material, in accordance with ICZN 75.4. One could argue that since carcharodontosaurid teeth are not diagnostic at the genus level, as already mentioned by Brusatte and Sereno [8], it is not possible to refer either the Egyptian or the Moroccan specimen to these teeth. However, as these teeth are lost, and a valid neotype designation exists, this is irrelevant from a formal taxonomic perspective. There is, however, a better case for the referral of the Moroccan material to the same taxon as the original teeth, as it not only comes from a geographically much closer location (thus fulfilling ICZN 75.3.6. "the neotype came as nearly as practicable from the original type locality"), but also shows teeth with a greater backwards curvature and an extreme labiolingual flattening, which are at least superficially more similar to the tooth described by Depéret and Savornin [15] and designated as lectotype by Stromer [22]. In the interest of preserving the widely used genus *Carcharodontosaurus* and family Carcharodontosauridae, and in accordance with ICZN article 75.4, we thus concur with the designation of SGM-Din 1 as the neotype of *Carcharodontosaurus saharicus* sensu Brusatte and Sereno [8].

The distinct features of the Egyptian specimen, combined with the results of the phylogenetic analysis, are enough support for the erection of a new genus and species of carcharodontosaurid dinosaur for this taxon. While this could be seen as problematic given recommendation 73B of the ICZN, which advises against designation of material that the author only knows from descriptions and illustrations, but has not studied first hand, there are exceptions where naming such a specimen is still possible, following declaration 45, recommandations 73G –J. We think that this is such an exceptional case where a nomenclatorial act is justified and outline our reasoning following the individual recommendations of declaration 45 below:

**"73G Specific reasons for designation of an unpreserved specimen as the name-bearing type":** This is followed by us showing that SNSB-BSPG 1922 X 46 is A) clearly distinct from the Moroccon neotype and other carcharodontosaurs and represents a separate species that can be diagnosed on the basis of a unique character combination as well as some additional possible autapomorphies. B) The specimen has undoubtedly been destroyed in WW2 and no comparable extant specimen is present. Consequently, direct observations are not possible, and a revision, as undertaken here, can thus only rely on the published descriptions and illustrations [1, 22, 25, 26] and the newly recovered photographs of the specimen. C) The formal naming of this distinct taxon is needed for the following reasons: 1) The material described by Stromer [1], although in earlier years often neglected, has nonetheless played a pivotal role for our understanding of carcharodontosaurid anatomy, phylogeny and evolution. 2) The Egyptian specimen is the only diagnosable African carcharodontosaurid from east of the Trans-Saharan seaway and thus also of considerable biogeographic interest. 3) The specimen described by Stromer is the only African carcharodontosaurid for which at least a partial post-cranial skeleton is known so far, and one of the most complete carcharodontosaurids known outside South America in general, making it an important reference taxon for carcharodontosaurid anatomy.

**"73H Assertion of due diligence":** This is fullfilled, by our archival and collection searches in Munich, Berlin and Tübingen, where we recovered as much information as possible on the specimen. Preservation or recovering of the original specimen is not applicable here, as the museum and most of its inventory were destroyed in World War II.

**"73I Consultation with specialists":** The first author of the paper has also worked on theropods since his bachelors degree in 2018 and has been part of the Mesozoic Tetrapod Working Group, developing a large phylogenetic dataset for theropods and is thus familiar with

theropod anatomy. The remaining two authors of the paper (EC and OWMR) are experts in theropod anatomy, taxonomy and systematics, and specifically qualified in this case due to their many years of experience working on theropods in general and carcharodontosaurids in particular.

**"73J Comprehensive iconography and measurements":** For this we provided a comprehensive description of the specimen that is as detailed as possible including anatomical descriptions, measurements in the supplementary material as well as the compilation of photographs and illustrations of the specimen. In particular, we would like to point out that all our observations and interpretations can be verified on the basis of the available descriptions and images; in cases where our interpretations are somewhat ambiguous, as they might be only based on the images, but not additionally explicitly supported by Stromers descriptions, this is indicated in our descriptive section. Note that the fact that Stromers descriptions and illustrations are highly accurate in specimens that can still be studied first hand, and the fact that the general features of his illustrations of the Egyptian specimen are in accordance with the photographs, indicate that this data is reliable. Nonetheless, the diagnostic characters chosen for our new taxon have been carefully selected and are all directly visible in the photographs of the specimen as well as additionally supported by Stromers descriptions and/or illustrations. A second section of additional diagnostic characters was added for characters that are highly unsual and directly supported by both Stromers illustrations and descriptions, but cannot be directly verified from the photograph due to either a lack of resolution or missing perspective.

## Systematic Paleontology

**Theropoda** Marsh, 1881

**Allosauroidea** Marsh, 1878

**Carcharodontosauriformes** new clade, comprising *Carcharodontosaurus saharicus*, *Sinraptor dongi* and all descendants of their most recent common ancestor

**Carcharodontosauridae** Stromer, 1931

Tameryraptor gen nov.

**Zoobank:** urn:lsid:zoobank.org:act:8B56E7DA-EEA4-49E4-9283-55C3064FEC96

**Etymology:** *Tameryraptor* meaning "thief from the beloved land" is a combination of one of the more informal ancient Egyptian names for Egypt, (ta-mery) meaning beloved land [96]; and the latin word for thief, raptor

**Diagnosis:** As for species

**Type species:** *Tameryraptor markgrafi* sp. nov.

**Zoobank:** urn:lsid:zoobank.org:act:D1D99215-0770-45FB-83D5-EB2F121BCF12

**Etymology:** *markgrafi* in reference to Richard Markgraf, the Austrian fossil collector who discovered most of the dinosaur remains described by Stromer.

**Holotype:** SNSB-BSPG 1922 X 46, now destroyed. The specimen included parts of the left and right maxilla, both nasals, parts of the braincase and endocast of the braincase, three cervical vertebrae, an anterior to anterior mid-caudal vertebra and chevron, the proximal part of a dorsal rib, a left ischium, right and left pubis, both femora and a left fibula. In the absence of the actual specimen, the descriptions and figures of Stromer [1, 25] and the photograph UAT 678/20 stand as representatives for the holotype, in accordance with ICZN article 73.1.4.

**Horizon and Locality**: SNSB-BSPG 1922 X 46 was found two kilometers from Ain Gedid on the Western foot of the Gebel Harra (Fig 1) in basal layers of hardened, gypsum-free marl (Layer p of Stromer's 1914 profile) of the Bahariya Formation, Cenomanian.

**Diagnosis:** *Tameryraptor markgrafi* gen. nov sp. nov. is a carcharodontosaurid theropod characterized by the following unique character combination (autapomorphies marked with a

\*): antorbital fossa visible on nasals in lateral view; small nasal horn on the medial rim of the nasals\*; maxilla with well-developed antorbital ridge; maxillary teeth highly symmetrical anteroposteriorly; prefrontal facet on frontal not expanded; frontals vaulted dorsally\*; femur with spike like accessory trochanter; proximal fibula more anteriorly than posteriorly expanded.

**Additional diagnostic features (see previous section for explanation):** internasal suture inclined ventromedially\*; anterior middle caudal vertebrae with well-developed pneumaticity including foramina posterior to prezygapophyses opening into spinoprezygpaophsyseal fossa\* and pneumatic foramina in vertebral bodies;

## Carcharodontosaurid systematics, phylogeny and evolution

Our understanding of the ingroup relationships of the Tetanurae has been continuously evolving ever since the clade was first named by Gauthier in 1986 [97]. Most commonly, Tetanurans are divided into three main lineages, Megalosauroidea, Allosauroidea and Coelurosauria [61, 88, 98–100]. Typically, Allosauroidea is recovered within the Avetheropoda, as sister group to Coelurosauria, although some analyses have found them as sister group to Megalosauroids within a monophyletic Carnosauria to the exclusion of Coelurosauria [47, 101, 102]. The ingroup relationships of Allosauroidea itself also remain poorly understood, with groups like Carcharodontosauria, Megaraptora, Piatnitzkysauridae and Metriacanthosauridae being variably recovered within, or outside of the Allosauroidea [61, 73, 87, 88, 103].

Close relationships between the Metriacanthosauridae and carcharodontosaurids were first hypothesized by Coria and Currie [52], who noted similarities between the braincase of *Sinraptor*, *Giganotosaurus* and *Carcharodontosaurus*. Among these shared characteristics were the strong posteroventral orientation of the braincase in relation to the frontal region of the skull roof, a supraocciptial knob at least twice the width of the foramen magnum, supratemporal fenestrae that open more anteriorly than dorsally and three other characters [52]. Other analyses supported this relationship and added further synapomorphies, such as the presence of an anterior spur on the ventral process of the postorbital, or the presence of an expanded ischial boot to the group [47, 98, 104]. However, more recent analyses usually recover the Metriacanthosauridae as the sister group to Allosauria, a clade containing *Allosaurus fragilis*, *Carcharodontosaurus saharicus* and all descendants of their last recent common ancestor [43, 61, 87, 88, 105]. The results of our analyses add further evidence for a close relationship between metriacanthosaurids and carcharodontosaurids, in a clade for which we suggest the name Carcharodontosauriformes. Carcharodontosauriformes are defined as a node-based clade that includes *Carcharodontosaurus saharicus*, *Sinraptor dongi* and all descendants of their most recent common ancestor. Based on our phylogenetic analysis, Carcharodontosauriformes can currently be diagnosed by the following unambiguous synapomorphies: Postorbital, supraorbital brow in dorsal view bulges notably laterally over the orbit; quadrate head partially hidden in lateral view by a bony lamina connecting the ventral and posterior process of the squamosal; nuchal crest discontinuous between squamosal and parietal with U-shaped incision; ventromedially opening recess on ectopterygoid wing of pterygoid confluent with ectopterygoid recess; exit of facialis (VII) nerve placed in a joint depression (prootic fossa) with the trigeminal foramen on the lateral side of the prootic; mesial and distal maxillary and dentary teeth weakly recurved with apex of the tooth positioned above the distal half; at least some distal maxillary teeth with straight distal carina in labial or lingual view; intercentrum of axis with anterodorsally tilted ventral surface; mid–caudal vertebrae with lateral ridge extending anteriorly from the dorsal surface of the base of the transverse process; lateral tubercle present on humeral shaft distal to the deltopectoral crest; tab on dorsomedial aspect of medial most metacarpal with phalanges overlapping adjacent metacarpal present; ventrolateral tab on

the proximolateral surface of first position metacarpal present. Due to the differing content of the clade in different analyses (mainly in regards to the position of *Neovenator* and *Eocarcharia*) Carcharodontosauriformes is not always defined by the same synapomorphies. For the list of unambiguous synapomorphies we decided to use the Split *K* = 12 dataset, as this dataset does not require any assumptions on the assignability of the split material. Possible additional characters under ACCTRAN and DELTRAN present in the Carcharodontosauriformes in both Split and Merged sets include the following: Basisphenoid recess opening posteroventrally rather than opening ventrally (This is possibly ancestral to Allosauroidea, since its condition is unclear in Allosauridae, as the basisphenoid recess opens ventrally in *Allosaurus* [106], but posteroventrally in *Asfaltovenator* [MPEF PV 3440]); dorsal ribs with pneumatic opening in the angle of the capitulum and tuberculum (this character is variably distributed within Averostra. Most tyrannosauroids such as *Alioramus*, *Eotyrannus*, *Murusraptor* and *Maip* exhibit this pneumatic opening [49, 64, 66, 107]. In ceratosaurus, it is present in *Ceratosaurus* [75], but absent in Abelisauroids like *Aucasaurus* or *Majungasaurus* [108, 109]. It is also absent in the probable megalosauroid *Yunyangosaurus* [110] and the megalosauroids like *Afrovenator* [MNBH TIG1], and *Wiehenvenator* [111]. Within Allosauroidea pneumatic openings in the angle of the capitulum and tuberculum are somewhat erratically distributed, being absent in *Asfaltovenator* [MPEF PV 3440], *Allosaurus*, *Alpkarakush* [45, 112] and the basal Carcharodontosaurids *Lusovenator* and *Lajasvenator* [71, 113], but present in *Sinraptor*, *Concavenator*, *Neovenator* and *Mapusaurus* [12, 57, 58, 60]).

Our analyses confirm the carcharodontosaurid affinities for the Late Jurassic *Lusovenator* [71] and *Veterupristisaurus* ([114]; see below), thus confirming a Jurassic origin for the clade, as is to be expected from their sister-group relationship with metriacanthosaurids. Furthermore, some analyses (Split and Merged, using equal weights) also recovered *Poekilopleuron* as a possible carcharodontosaurid, which would push the known record of the clade into the Middle Jurassic. Interestingly, *Lusovenator* is consistently recovered as the sister taxon to the second oldest secure carcharodontosaurid, *Lajasvenator* from the Valanginian of Argentina [113], indicating an early offshoot of the main carcharodontosaurid lineage. On the other hand, two other supposed Early Cretaceous carcharodontosaurids, *Concavenator* and *Siamraptor*, are found as the earliest branching carcharodontosaurians in the Split *K* = 12 analysis (and all other analyses that recovered a monophyletic Carcharodontosauria), but were recovered within metriacanthosaurids in the Merged *K* = 12 analysis (and other analyses that failed to recover Carcharodontosauria). This further highlights the close anatomical similarity between metriacanthosaurids and early carcharodontosaurs. This is mirrored by *Neovenator*, which can be placed within Carcharodontosauria, as sister taxon to Carcharodontosaudiae, or considerably more basal, outside Carcharodontosauriformes. Thus, there still seems to be considerable phylogenetic uncertainty at the base of the Carcharodontosauriformes.

In the higher carcharodontosaurids, *Acrocanthosaurus* and, probably, *Datanglong* represent the only Laurasian taxa in an otherwise Gondwanan Group. *Acrocanthosaurus* has originally been described on the basis of two fragmentary specimens from the Antlers Formation (Late Aptian-Early Albian) of south-eastern Oklahoma [53], but several other specimens from roughly contemporaneous units from several parts of the US have been referred to the same taxon over the years [2, 73, 115, 116], resulting in geographic range of this taxon from Texas via Wyoming to Maryland. Whereas the referral to *Acrocanthosaurus atokensis* was based on apomorphic characters for the partial to fragmentary specimens described by Harris [2], D'Emic et al. [115] and Carrano [116], no formal reason for the referral of the most complete specimen (NCSM 14345) has been published so far [43, 73], other than that it is of similar size and comes from the same formation as the holotype. We noted several differences between the type material (OMNH 10146) and specimen NCSM 14345, which might make a re-evaluation

of this referral necessary. These differences include: continuous antorbital fossa on the lacrimal (type) versus a separation of the dorsal and ventral portions of the fossa by an anterior expansion of the lateral lamina (NCSM 14345); wide U-shaped ventral margin of the orbit with the antorbital fossa reaching below the orbit on the jugal (type) vs. narrow orbital margin with antorbital fossa being restricted to the antorbital part of the jugal (NCSM 14345); supraoccipital crest narrow and undivided (type) vs. wide and bifurcated dorsally (NCSM 14345); slightly acute angle (type) vs. wide angle (NCSM 14345) between posterior skull roof and occiput in the braincase; a ridge extending from the dorsal rim of the paroccipital process medially below the posterior exit of the mid-cerebral vein is present in the type but absent in NCSM 14345; trigeminal foramen is a single opening (type) vs. completely split (NCSM 14345; see [8]); pneumatic pockets next to the occipital condyle are considerably wider in the type than in NCSM 14345; ventral indentation of the articular surface of the occipital condyle absent in the type, but present in NCSM 14345; distal shaft of tibia markedly flexed medially (type) vs. straight (NCSM). Although an evaluation of the taxonomic or phylogenetic significance of these differences is clearly beyond the scope of this paper, a reexamination of the North American material is warranted. Carcharodontosaurid diversity in the Early Cretaceous of North America might have been greater than currently recognized.

*Datanglong* is the only Laurasian carcharodontosaurid taxon more derived than *Acrocanthosaurus*. Although its exact position could not be resolved, it is usually recovered within, or as possible sister taxon to the Carcharodontosaurinae, an otherwise exclusively Gondwanan group. However, the highly fragmentary nature of the taxon (95% missing data), and more detailed comparisons of the characters used to unite this taxon with carcharodontosaurids cast some doubts on this position. Mo et al. [117] identified *Datanglong* as a member of Carcharodontosauria based on the presence of large external pneumatic foramina and internal spaces in the lateral surface of the ilium and an peg and socket articulation of the ischium and ilium. Within carcharodontosaurs, pneumatic structures have been described for the ilium of *Mapusaurus* and *Neovenator* [12, 58]. However, it is unclear if they are equivalent to the structures present in *Datanglong*. In *Neovenator* an internally subdivided, anteroposteriorly elongate, chamber is present immediately dorsal to the acetabulum, but neither the origin of a possible pneumatization nor the pneumatic nature could be established with certainty [58]. For *Mapusaurus*, Coria and Currie [12] describe an excavation into the interior of the ilium as originating from the brevis fossa. Furthermore, they describe "shallow but distinct pits" in the brevis fossa in the region between the base of the ischial peduncle and this excavation. In one of their Figures ([12]: p 99, Fig 26), these are labeled as pneumatic diverticulae, while they are described in the text as being "associated with the lateral caudofemoralis brevis musculature" [12]. Similar shallow pits are also present in the brevis fossa of *Giganotosaurus* (MUCPv-Ch1), but these do not appear to correspond to any internal pneumatic structures. None of these taxa exhibit pneumatic openings on the lateral side of the ilium as have been mentioned, but not described in detail in *Datanglong*. Mo et al. [117] directly compared the ilium of *Datanglong* to *Aerosteon*, which, along with the rest of Megaraptora, was recovered within Carcharodontosauria in their phylogenetic analysis. Highly pneumatic ilia are characteristic of Megaraptora, being present in *Aerosteon*, *Australovenator*, *Murusraptor* and *Tratayenia* [31, 66, 90, 118]. The distribution of pneumaticity in this group is, however, different from the pneumaticity observed in carcharodontosaurids. While both *Murusraptor* and *Aerosteon* also exhibit pneumatic openings within the brevis fossa, these structures are located significantly more posteriorly in these taxa [65, 66]. Additional pneumatic openings, somewhat similar to the structures described in *Datanglong*, are present on the lateral surfaces of the ilia of *Aerosteon*, *Murusraptor* and *Tratayenia* [31, 66, 90], which might suggest a closer connection to the latter group. However, both *Murusraptor* and *Aerosteon* also exhibit pneumatic openings

along the medial surface of the ilium, where they are mainly associated with the sacral ribs and, at least in *Murusraptor* directly originate from pneumaticity in the sacral vertebrae [31, 65, 66]. In contrast, the sacral vertebrae of *Datanglong* lack this pneumaticity, suggesting a different origin of the possible pneumatization in the ilium of *Datanglong*. Thus, it remains unclear if the pneumaticity in the ilium of *Datanglong* is homologous to the structures of carcharodontosaurids or megaraptorans. Furthermore, no internal pneumatic spaces in the ilium were mentioned or described for *Datanglong*, and the distribution of pneumatic foramina seems rather random. It is worth noting that structures resembling pneumatic openings can also be the result of bioerosion [119, 120]; however, without personally examining the specimen it is impossible to further evaluate the possible pneumaticity in *Datanglong*.

The peg and socket articulation of the ischium and ilium does not help to clarify the position of *Datanglong*, as it is not only present within Carcharodontosauria, but also in some ceratosaurs, like *Ceratosaurus* (BYU 12893) and abelisaurids, tyrannosauroids like *Tyrannosaurus* and *Alioramus* [49, 80], and notably in the possible megaraptoran *Siats* [70].

The similarities between *Datanglong* and megaraptorans have already been pointed out by Samathi et al. [121], who advised caution regarding the placement of this taxon within Carcharodontosauridae. Constraining *Datanglong* into the Megaraptora only requires two additional steps in both Split and Merged sets in our analyses. Given the fragmentary nature of *Datanglong* and its instability in the phylogeny, we concur with Samathi et al. [121]. A detailed review of *Datanglong* would be needed to further clarify its position, as none of the authors of this paper have personally seen the specimen.

Another Laurasian Early Cretaceous theropod, *Shaochilong maortuensis* has been classified as a carcharodontosaur ever since it was separated from the genus *Chilantaisaurus* [5]. It has since been recovered within the group in multiple other analyses [11, 50, 71, 88], all of which, however, were heavily focused on basal Tetanurans. Thus, only very few tests of the phylogenetic position of this taxon in a more global context of theropod interrelationships have been presented so far [5, 103], and all of these analyses have been based on the characters proposed as carcharodontosaurid synapomorphies present in this taxon by Brusatte et al. [122]. However, several of these characters are problematic or not limited to carcharodontosaurids. The following characters were proposed by Brusatte et al. ([122]: 38): a limited antorbital fossa on the lateral surface of the maxilla, deep interdental plates, fused frontal-frontal and frontal-parietal sutures, limited supratemporal fossae on the frontal, a curved muscle crest within the supratemporal fossa, postorbital-lacrimal contact above the orbit, paracondylar pneumatic foramina leading into a midline recess underneath the endocranial cavity, a largely anterior-posterior trending fenestra ovale, a trigeminal (V) nerve foramen located posterior to the nuchal crest, two foramina for the facial (VII) nerve, and a thickened antotic crest on the laterosphenoid.

Several of these characters are indeed shared between *Shaochilong* (IVPP V 2885) and carcharodontosaurids and are absent or rare in other theropods, such as high interdental plates (also present in megalosaurine megalosauroids; [98] and abelisaurids [46, 123]), a fused frontal-frontal and frontoparietal suture (also present in abelisaurids (e.g., [46]), and a limited supratemporal fossa on the frontal (also present in maniraptoriform theropods [124]).

A reduced antorbital fossa in the jugal ramus of the maxilla is also found in abelisaurids (e.g., [46]), many megalosauroids (e.g. [35, 110, 125]), and numerous coelurosaurs (e.g. derived tyrannosauroids [84]; therizinosaurids [126], troodontids (e.g., [127]), *Archaeopteryx* [128]). In the ascending process, the antorbital fossa is usually small in non-coelurosaurian theropods in general, but it is still present, and similar in size to other basal tetanurans, in other carcharodontosaurids. Furthermore, the complete lack of an antorbital fossa in the ascending process in *Shaochilong* might be due to damage as the medial side of the area, where the medial lamina delimiting this fossa would be expected, is damaged.

The "curved muscle crest within the supratemporal fossa" is a problematic character, as it is rather unclear in how far the structures identified as such are comparable between the taxa mentioned. Brusatte et al. [122] indicated a curved crest that flexes posterolaterally from the anterior rim of the supratemporal fossa on the lateral side of the frontal as this structure in *Shaochilong*. A similar crest is indeed present, though more medially and posteriorly placed and more anteroposteriorly than mediolaterally directed, in the braincase referred to *Carcharodontosaurus iguidensis* [8], and Brusatte et al. [122] noted that this crest was also present in *Carcharodontosaurus saharicus*, *Giganotosaurus* and *Acrocanthosaurus* (NCSM 14345), although is slightly different orientations than in these two taxa. However, our own observations fail to identify a similar crest in the latter taxa. *Carcharodontosaurus saharicus* (SGM Din 1) and *Giganotosaurus* (MUCPv-Ch1) generally differ from *Shaochilong*, the braincase referred to *C. iguidensis* and *Acrocanthosaurus* in that a true supratemporal fossa is absent, as both the anterior and medial rims of the supratemporal fenestrae overhang a steeply inclined to near vertical wall of this opening. Within the opening, a short, posteroventrally inclined ridge is present in *Giganotosaurus* in its anteromedial part, but such a ridge seems to be absent in *Carcharodontosaurus saharicus*. Instead, here, a slight swelling transverses the anterior wall of the supratemporal fenestra mediolaterally a short way below the overhanging anterior rim. This swelling seems to follow the (fused) frontoparietal suture and leads to the postorbital condylus of the laterosphenoid laterally. Likewise, in *Acrocanthosaurus* (OMNH 10146, NCSM 14345), there are two low and slightly curved ridges within the supratemporal fossa that extend mediolaterally some distance posterior to the anterior rim of the fossa. These ridges again seem to follow the sutures between the frontal and the parietal and the parietal and laterosphenoid, respectively. Thus, it is unclear if these ridges can in any way be homologized with the "crest" seen in *C. iguidensis* and *Shaochilong*. Furthermore, very similar, low, curved mediolateral ridges along the frontoparietal and parietal-laterosphenoid sutures are present in many other theropods, such as *Allosaurus* (SMA 0005, DINO 11541) and *Sinraptor* (IVPP V 10600), and the frontoparietal suture is often especially marked in tyrannosauroids (e.g. [48, 129]), including megaraptorids [130]. Thus, more detailed comparisons are needed to establish homologies between these structures and to decide if and at what level they might be synapomorphic for certain clades.

Since neither postorbital, prefrontal or a lacrimal are known for *Shaochilong*, the presence of a postorbital-lacrimal contact above the orbit cannot be directly verified. A small orbital notch, as described by Brusatte et al [122], is, however, present on the frontal. Narrow orbital notches are also present in the braincase of *Murusraptor*, *Sinraptor* or *Alioramus* [60, 129, 130]. In none of these taxa does the lacrimal contact the postorbital directly, although these bones are in close proximity in the articulated skulls. Brusatte et al. [122] argued that such a contact was present in *Shaochilong* and thus the frontal excluded from the orbital rim; however, there is a rather large, more anteriorly than laterally directed and dorsoventrally thin orbital rim between the anterior end of the postorbital facet and the prefrontal articulation in this taxon. Brusatte et al. [122] acknowledges the presence of this orbital notch, but argued that it would not have contributed to the orbital rim. Nevertheless, the anterolateral surface of this notch does not bear any articular facet, and thus the situation is very similar to the situation in other theropods that have a reduced contribution of the frontal to the orbital rim (e.g. *Murusraptor* [130]). Thus, we consider the presence of a lacrimal-postorbital contact to be rather unlikely in *Shaochilong*.

The presence of paracondylar pneumatic foramina leading into a midline recess underneath the endocranial cavity have been interpreted as a carcharodontosaurid synapomorphy every since this morphology was first described for *Giganotosaurus* by Coria and Currie [52]. However, the morphologies seen in different carcharodontosaurids differ. Thus, a large, round

foramen is present within the subcondylar recess in the anterolateral side of the base of the neck of the occipital condyle on either side in *Carcharodontosaurus saharicus* SGM Din 1), the braincase referred to *C. iguidensis* [8] and on the right side, but not the left side, in *Giganotosaurus* (MUCPv-Ch1). In *Acrocanthosaurus*, a large pneumatic recess extends from the subcondylar recesses mainly anteriorly, but not medially into the neck of the occipital condyle (OMNH 10146), being more comparable to the situation found in many tyrannosaurids in this respect. *Meraxes* lacks invasive recesses extending from the subcondylar recesses altogether (MMCh-PV 65). In *Shaochilong*, a pneumatic recess is present in the subcondylar recess below the occipital condyle, and invades the basioccipital medially; this recess is thus in a markedly different position than the recesses in the neck of the condyle in the carcharodontosaurids mentioned above. A similarly located recess is only present in *Struthiomimus* [131].

The anteroposteriorly trending fenestra ovale and the double foramina for the exit of the facialis nerve mentioned as carcharodontosaurid characters for *Shaochilong* by Brusatte et al. [122] are based on an erroneous interpretation of the foramina on the lateral wall of the braincase in this taxon. The openings interpreted as the foramina for the facialis nerve quite clearly represent the fenestra ovale and fenestra pseudorotunda, as shown by their position within a triangular cochlear recess at the base of the paroccipital process, flanked dorsally by a posterolateral process of the prootic (Fig 15), and the fact that a well-developed stapedial groove is present posterolaterally to this opening on the left side of the braincase. The actual foramen for the facialis nerve seems to be represented by a rather small, matrix-filled opening ventrally in between the trigeminal foramen and the cochlear recess, above the broken base of the preotic pendant (Fig 15). Thus, only a single facialis foramen is present in *Shaochilong*. Consequently, the foramen identified as the fenestra ovale by Brusatte et al. [122] represents the opening of the posterior tympanic recess, which is placed posterolateral to the cochlear recess within the stapedial groove (Fig 15), as it is the case in many coelurosaurs (e.g. [129, 132]).

The last character, a thickened antotic crest, is currently poorly defined, as it is somewhat unclear how the situation in carcharodontosaurids is supposed to differ from other theropods. However, most of the antotic crest is actually missing in *Shaochilong*, and only its base is preserved. Although Brusatte et al. [122] state that this base indicates that the crest was unusually stout in *Shaochilong*, the crest seems to rapidly become thinner towards the lateral break on the left side of the braincase. However, not enough of the crest is preserved to determine its development with any certainty.

In all of our analyses, *Shaochilong* was recovered within the Tyrannosauroidea. In both Split and Merged analyses, four additional steps are needed to move the taxon into Carcharodontosauria (Table 2). A position within the Coelurosauria is supported by the following synapomorphies in both $K = 12$ datasets: Presence of a sagittal crest (a sagittal crest is not known for any allosauroids); a narrow and groove like infracondylar fossa of the occipital condyle (broad in allosauroids, but also narrow in at least some carcharontosaurids e.g. *Acrocanthosaurus*); basal tubera undivided and equally formed by basioccipial and basispenhoid (in all Allosauroids these are subdivided a longitudinal groove into a lateral and medial part formed by basisphenoid and basiocciptial respectively); presence of caudal tympanic recess (this is absent in Allosauroids with the exception of *Sinraptor* [85]). A position within Tyrannosauroidea is supported by the following synapomorphies in both $K = 12$ datasets: Maxilla, with extensive grooved sculpturing of external surface in adult individuals; sagittal crest extending onto the frontal; quadrate foramen semioval and widely open laterally, neural spines on distal caudal vertebrae present as a low ridge.

Apart from *Datanglong*, all other higher carcharodontosaurids come either from Africa or from South America, with the African taxa forming a paraphyletic array towards a monophyletic, purely South American clade Giganotosaurini. *Carcharodontosaurus saharicus* is thus

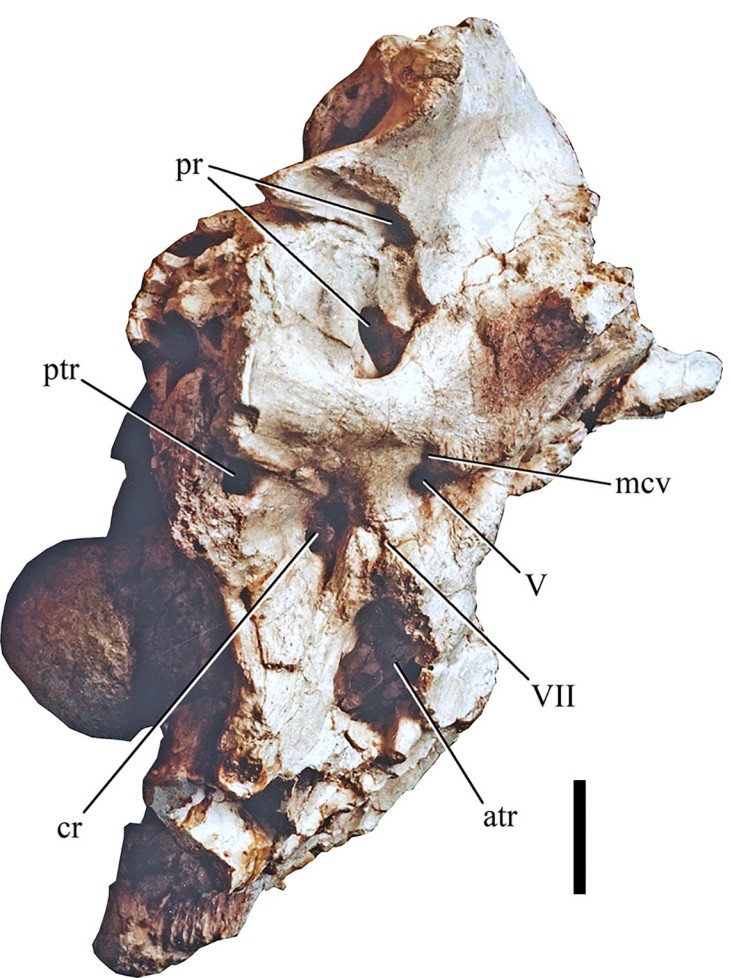

**Fig 15. Braincase of *Shaochilong maortuensis* (IVPP V 2885) in right lateral view.** Abbreviations: atr, anterior tympanic recess; cr, cochlear recess; mcv, foramen for the mid-cerebral vein; pr, pneumatic recesses of the dorsal tympanic recess; ptr, posterior tympanic recess. Roman numerals indicate the exits for the trigeminal (V) and facialis (VII) nerve. Scale bar is 2 cm.

found to be more closely related to the South American forms than to the Egyptian *Tameryraptor*. This might reflect the separation of eastern and western Africa by a Cretaceous seaway, which started in the Aptian Albian, and at the same time the close proximity of western Africa and South America up to the Lower-Upper Cretaceous boundary [133–135]. With a newly calibrated age of the oldest Giganotosaurini, *Tyrannotitan*, to the middle to higher Albian [136] this clade might thus have diversified in South America just about the time of the separation of South America from Africa, after a dispersal event from Africa. However, given the still very poor pre-"mid"-Cretaceous theropod fossil record from both Africa and South America, new finds are needed to test this hypothesis.

## The phylogenetic implications of merged codings and fragmentary taxa

Fragmentary taxa remain a big problem in phylogenies, introducing polytomies and rendering interpretation of results difficult. Despite this, some useful information can still be obtained, even when fragmentary taxa are included. Sometimes even very fragmentary material can be

placed, be it in a resolved position or in an approximate position through IterPCR [41]. Such placements should, however, only ever be seen as preliminary until further material of the taxon is found, and drawing far-reaching conclusions from such fragmentary material can be problematic. An example for this is *Ulughbegsaurus*, which was described on the basis of a maxillary fragment and identified as a carcharodontosaurid [137], representing the youngest record of the clade in Laurasia, with implications for the timing of faunal changes towards the tyrannosaurid-dominated latest Cretaceous faunas of Laurasia. Only two years later, this identification has been called into question by Sues et al. [138] upon the discovery of large dromaeosaurid material from the same formation. These authors noted that the supposed carcharodontosaurid characters are questionable and the specimen cannot be referred to any specific clade with any certainty.

Phylogenetic analysis can also be used to test the referral of non-overlapping material to a particular genus. This is not definitive, but referred specimens being recovered within close phylogenetic proximity to one another lends support for an identification as the same genus. A good example for this is *Carcharodontosaurus iguidensis*. *C. iguidensis* and its referred material MNBH IGU3/5 are consistently recovered in close relation to one another, even being possible sister taxa in all but one ($K = 15$) analysis, supporting the referral of MNBH IGU3/5 to *C. iguidensis*. In the split runs, both *C. iguidensis* and MNBH IGU3/5 are generally recovered either in close phylogenetic proximity to, or in a basal position within the Carcharodontosaurinae. When the codings for *Carcharodontosaurus iguidensis* are merged, it is almost always recovered outside of Carcharodontosaurinae, with the exception of the unweighted analysis, where it is recovered in a basal polytomy in the Carcharodontosaurinae, mirroring the positions of the split run. However, "*Carcharodontosaurus*" *iguidensis* is never recovered as sister taxon to *Carcharodontosaurus saharicus*, calling into question the referral of both species to the same genus. However, we refrain from proposing a new genus for this species, as more detailed comparisons and, probably, additional material for both taxa would be needed to firmly establish their distinction.

Similar results are also present for *Aerosteon* and the referred hind limb MCNA PV 3139. Both are always recovered within the Megaraptora, with close proximity to one another, or as possible sister taxa, making the referral of MCNA PV 3139 to *Aerosteon* more likely. When the codings are merged, *Aerosteon* is recovered in a derived position in the Megaraptora, often as a possible sister taxon to *Australovenator*.

*Megaraptor* and the referred juvenile specimen MUCPv 595 are consistently recovered within the Megaraptora. In three out of the four runs (equal weight, $K = 12$, $K = 15$), MUCPv 59 is recovered in a less derived position than *Megaraptor*, while in the other ($K = 9$) it is recovered within a clade consisting of *Murusraptor*, *Maip* and *Orkoraptor*. The former result is not unexpected, as juvenile taxa being recovered in a comparatively more basal position in comparison to mature individuals is something that has been observed multiple times before (e.g. [139–142]). Therefore, this result does not necessarily contradict the identification of MUCPv 595 as *Megaraptor*. The position of the juvenile specimen within a clade with *Murusraptor* and *Maip* is the effect of a single unambiguous synapomorphy, which might distinguish MUCPv 59 from the other specimens of *Megaraptor*. An accessory posterior centrodiapophyseal lamina in the anterior caudal vertebrae, initially described as an autapomorphy for *Maip* [64], is also present in *Murusraptor* (MCF-PVPH-411) and MUCPv 59, but notably absent in one of the other referred specimens of *Megaraptor* (MUCPv 341). How far this character is dependent on the position of the caudal vertebra remains unknown, as the caudal series of *Murusraptor*, *Maip* and MUCPv 59 are highly incomplete. It is possible that an accessory posterior centrodiapophyseal lamina was also present in MUCPv 341, as the specimen only preserves two caudal vertebrae. The referral of MUCPv 595 to *Megaraptor* is reasonable under these aspects.

When the two taxa are merged, *Megaraptor* is mostly recovered as the sister taxon to a derived clade of megaraptorans containing *Maip*, *Murusraptor*, *Orkoraptor*, *Australovenator* and *Aerosteon*.

A different situation is present for *Eocarcharia* and its referred Maxilla MNBH GAD 7. Both are consistently recovered in different positions in all analyses. The more informative MNBH GAD 7 (8.3% coded data compared to 5.4% coded data) is consistently recovered within Carcharodontosauriformes, either within Carcharodontosauria outside of Carcharodontosaurinae (equal weight, $K = 15$, $K = 12$) or within Metriacanthosauridae ($K = 9$). In contrast, the position of *Eocarcharia* is more variable, with the taxon being recovered either at the base of Carcharodontosauria ($K = 15$), variably outside or inside of Carcharodontosauriformes (equal weight, $K = 12$) or outside of Carcharodontosauriformes ($K = 9$). While this does not rule out an assignment of MNBH GAD 7 to *Eocarcharia*, it also does not support it. In the merged datasets *Eocarcharia* (merged) is more often than not recovered outside of Carcharodontosauria, either as sister taxon to the Carcharodontosauriformes ($K = 9$, $K = 12$), as a basal member of Metriacanthosauridae ($K = 15$), or in an uncertain position as either the basalmost carcharodontosaurid, or metriacanthosaurid (equal weight). The position of *Eocarcharia* as a carcharodontosaurid in previous analyses (e.g. [9, 50, 71]) seems to be strongly influenced by the codings of the referred maxilla, as the holotype seems to be less diagnostic.

Despite being the most incomplete taxon in the dataset (99% missing data), *Veterupristisaurus* is consistently recovered as a member of Carcharodontosauridae in all analyses, as was initially proposed by Rauhut [113]. Its exact position within the group could not be determined and a position at the base of the Carcharodontosauria, as was recovered in previous analyses [71], could not be confirmed in our analyses. *Veterupristisaurus* is recovered in the Carcharodontosauridae based on the presence of a lateral ridge extending anteriorly from the base of the transverse process and a funnel shaped anterior entrance of the neural canal, both of which are almost exclusively present in Carcharodontosauridae, with the exception of *Dubreuillosaurus*, which also exhibits both of these traits [94], *Coelurus*, which has the anterior ridge (YPM 1991), and *Poekilopleuron*, in which the ridge might been present based on the drawings [143]. It should be noted, however, that the main character used by Rauhut [113] to diagnose *Veterupristisaurus*, a double spinoprezygapophyseal lamina in the mid-caudal vertebrae, has since also been found in *Lusovenator* [71], so that the validity of the former taxon is questionable.

The phylogenetic position of the extremely fragmentary Asian *Kelmayisaurus* (~96% missing data) has been the subject of some debate. Initially, it was considered to be a tetanuran of uncertain affinities [144, 145], or even a nomen dubium by Rauhut & Xu [146]. A carcharodontosaurid identity has first been proposed for *Kelmayisaurus* by Brusatte et al. [147]. In our analyses *Kelmayisaurus* is recovered within varying positions among Allosauroidea. Only in some of the analyses (Split: equal weight, $K = 15$) it is exclusively recovered within the Carcharodontosauria, while in others it is recovered within the Metriacanthosauridae (Split: $K = 9$ and Merged: $K = 12$, $K = 9$), or in positions inside, or outside either of the two clades (Merged: equal weight, $K = 15$ and Split: $K = 12$). Again, the fragmentary nature of these remains do not allow for a secure placement at the moment.

The fragmentary North American taxon *Siats* (~91% missing data), initially regarded as a carcharodontosaurian neovenatorid [70] and sometimes recovered as a basal carcharodontosaur [71], is here recovered within the Megaraptora in all analyses. This is in line with results obtained by Naish & Cau [107] or Bell et al. [148].

*Chilantaisaurus* (~90% missing data) has had a varied phylogenetic history. Initially described as a Megalosauroid [149], it has since been assigned to several positions within Allosauroidea (e.g. [2, 61, 64, 70]), within Megaraptora [107, 148] or within Spinosauroidea [47].

Our analyses produced inconsistent results. It is recovered as the sister taxon to *Neovenator* (Split: $K = 9$ and Merged: $K = 12$, $K = 9$), in a basal position within Allosauroidea (Split: $K = 15$, $K = 12$), or as a tyrannosauroid with uncertain affinities (Split: equal weight, Merged: equal weight, $K = 15$).

*Taurovenator* is a taxon that is so far only based on a single postorbital from the Huincul Formation in Argentina [62]. Consequently, it is one of the most incomplete taxa of the analysis (~98% missing data). Despite this, it is usually recovered exclusively within the Carcharodontosauria or Carcharodontosauridae, with the exception of the equal weight analyses, where an additional position as the sister taxon to *Carnotaurus* is possible. The postorbitals of both taxa indeed share many features. Both are T-shaped and exhibit a highly rugose surface texture. Furthermore, they share the presence of a triangular intraorbital process and the presence of an orbital boss, or cornual process above the ventral process [62, 78]. These results are thus not unexpected, given the lack of data for *Taurovenator*. With increased character weight this relation disappears and *Taurovenator* is often recovered as closely related to *Carcharodontosaurus saharicus* (Split: $K = 9$ and Merged: $K = 15$, $K = 12$, $K = 9$). A synonymity of *Taurovenator* and *Mapusaurus*, as proposed by Coria et al. [112] is not supported by our analyses, but it only takes one additional step in both Split and Merged datasets to move the two taxa into a sister group relation. Given the fragmentary nature of the taxon, we therefore do not exclude this possibility.

Although Allain and Chure [143] noted many similarities between *Poekilopleuron* and Allosauroids, such as the sigmoidal humeral shaft and the L-shaped middle chevrons, it was tentatively referred to the Megalosauroidea. In all of our analyses, *Poekilopleuron* is instead recovered as an allosauroid, usually within Metriacanthosauridae (Split: $K = 15$, $K = 12$, $K = 9$, and Merged: $K = 15$, $K = 12$, $K = 9$). In both equal weight analyses, an additional position at the base of Carcharodontosauria is also possible. As the original material is no longer available, new finds would be needed to solve the phylogenetic relationships of this taxon.

## Implications for North African Dinosaur faunas

To this day, our understanding of the North African Dinosaur faunas remains limited. Most of what we know is based on fragmentary material from the Egyptian Bahariya Formation, the Moroccan Kem Kem Group and the slightly older Elrhaz Formation in Niger. The situation is further complicated due to the destruction of most of the Bahariya material. As the material from the Bahariya Formation represented the first fauna of Mid-Cretaceous terrestrial vertebrate faunas from Africa, it became a reference fauna for comparisons with other finds from Africa. Thus, especially dinosaur taxa known from the Bahariya Formation were often identified in other localities from all over northern Africa, often with little or no justifications other than that the material collected represented the same major clade and was of comparable size (e.g. [94, 150]). Many of these fossils come from the so called "Continental Intercalaire", an informal stratigraphic unit originally claimed for all terrestrial sediments in between the marine uppermost Carboniferous and the Cenomanian transgression [151]. Correlations and especially dating of the different units included in the "Continental Intercalaire" are difficult (see [152, 153]), and have largely been based on faunal comparisons of vertebrate faunas (e.g. [17, 21, 152, 154]), including dinosaurs. Thus, faunal comparisons with the relatively well-dated Bahariya Formation were used to assign a Cenomanian age to many of the higher parts of the "Continental Intercalaire" in western Africa (e.g. [18, 21]), many of which had previously been considered to be Early Cretaceous (often Aptian-Albian) in age (e.g. [94]).

Vertebrate fossils from the Moroccan Kem Kem Group are often directly compared to the lost material of the Bahariya Formation, with a special focus on the theropods of both

formations, which are often claimed to be conspecific (e.g. [7, 155]). At first glance, these comparisons appear reasonable based on the supposed similar age (Cenomanian age for the Bahariya Formation [156, 157] and Albian to Cenomanian age for the Kem Kem Group [17, 18, 158]) and similar faunal composition of both strata, although this is partially circular reasoning, as the similar faunal composition is used to infer the similar age. Both formations contain large spinosaurids, carcharodontosaurs, abelisaurs and additional enigmatic theropod remains [18, 25, 159]. While the abelisaurid is not yet named in either formation, the theropod species *Spinosaurus aegyptiacus*, *Sigilmassasaurus brevicollis* and *Carcharodontosaurus saharicus* are often claimed to be present in both formations (e.g. [18, 59]), with the neotypes of *Spinosaurus* and *Carcharodontosaurus* proposed from Moroccan material [8, 155]. Furthermore, the enigmatic Moroccan *Deltadromeus* was initially compared to *Bahariasaurus* by Sereno et al. [7], who also referred some of the Egyptian material described by Stromer in 1934 to *Deltadromeus*. Likewise, Ibrahim et al. [18] considered a large theropod femur from the Bahariya Formation (SNSB-BSPG VIII 1969), described by Stromer [22], to belong to *Deltadromeus*. However, comparisons between the material of the two strata rarely includes detailed direct comparisons between the specimens, both due to the destruction of the original material and due to the lack of detailed descriptions for the Moroccan specimens. As pointed out by Evers et al. [33], apart from the proposed conspecific theropod taxa, there are no shared tetrapod species between the Kem Kem Group and the Bahariya Oasis. Indeed, in crocodylomorphs, similar taxa are present, but so far, not even the same genus has been identified in both units [160–162]. Notably, peirosaurid, uruguaysuchid and pholidosaurid crocodyliforms, which have been described from the Kem Kem Group (see [18] and references therein) are so far absent from the Bahariya Formation. Likewise, no shared sauropod genera are present in the Kem Kem Group and the Bahariya Formation [18, 163–167], and rebbachisaurids, which seem to be common in the Kem Kem Group [18, 166], are so far unknown from all of eastern Africa. Thus, the similarities seem to be restricted to theropod dinosaurs, and in the absence of detailed comparisons of the theropod taxa found, the question is in how far this might simply represent similarities in the general theropod faunal compositions across Gondwana in places like Niger, Morocco, Brazil and Argentina, in respect to the frequent co-occurrence of abelisaurids, carcharodontosaurids and spinosaurids in geological units of differing age [12, 168–173].

The identification of SNSB-BSPG 1922 X 46 as a distinct genus and species from the Moroccan *Carcharodontosaurus saharicus* adds to the differences between the two strata. While our results do not necessarily exclude the possibility that either taxa are shared between the strata, as little is still known of the faunal compositions of both, it does exemplifies the problems caused by superficial comparisons. This has implications for the comparability of the other theropod taxa between the two strata, and warrants a detailed reexamination of the proposed presence of *Spinosaurus*, *Sigilmassasaurus*, *Deltadromeus* and *Bahariasaurus* in both strata [7, 33, 59, 174]. Indeed, many of these referrals are questionable if the specimens are compared in detail [175].

The results of our phylogenetic analysis further indicate a possible closer relationship of the western African carcharodontosaurids with taxa from South America. This is also found in other tetrapods from the Kem Kem Beds and other western African localities. Thus, uruguayosuchid, pholidosaurid and peirosaurid crocodyliforms are known from the Kem Kem Group and other western African localities, as well as South America (see [18, 161, 176–178], and references therein), but are absent or at least rare (in the case or peirosaurids) in eastern Africa. Likewise, rebbachisaurid sauropods are common in the Aptian-Cenomanian of South America [179] and western Africa [164, 166, 167, 180] but have not yet been recorded from eastern Africa. Thus, it seems that western Africa might have had closer biogeographical ties with

South America than with eastern Africa in the "Mid" Cretaceous, possibly due to the establishment of the trans-Saharan seaway during that time.

The recognition that the "Mid"-Cretaceous northern African terrestrial vertebrate faunas are not as similar as often assumed should also necessitate a re-evaluation of the ages of the different geological units across the continent. As pointed out above, the supposed close similarities between western African vertebrate faunas and the Bahariya Formation was often used to infer a Cenomanian age for the former (e.g., [21]). However, the actual lack of overlapping species, and the fact that older faunas, such as that of the Elrhaz Formation of Niger, are similar in general terms (i.e. in respect to the higher groups represented), leaves the possibility open that western African faunas might be slightly older than currently assumed. Thus, independent means of dating these units (i.e. apart from their vertebrate faunas) would be desirable. In that respect, another supposedly "Mid"-Cretaceous vertebrate fauna from eastern Africa is also of interest, the Wadi Milk Formation of Sudan [181–183]. This unit has originally also been dated as Cenomanian on the basis of palynology and the fish fauna (see [181] and references therein), but recent radiometric dates from detritional zircons indicate that this formation is actually Campanian or younger in age [184]. Rauhut [183] identified several large theropod vertebrae and a proximal metatarsal III from that formation as a carcharodontosaurid; if this identification can be confirmed, these remains would represent the youngest record of carcharodontosaurids, indicating that this clade survived considerably longer in Africa than in South America.

In addition, the increased diversity of North African carcharodontosaurs warrants a reevaluation of other material previously referred to the genus *Carcharodontosaurus*. This includes isolated material from across North Africa (e.g. [94, 185]) and especially from more distant localities such as Brazil [186], and also encompasses the type and referred material of *Carcharodontosaurus iguidensis*. As noted above, the latter, similar to *Tameryraptor*, differs in several aspects from the Moroccan *Carcharodontosaurus saharicus* and is not recovered as the sister taxon to *C. saharicus* in any of the Merged, or Split analyses. A detailed comparison of the two carcharodontosaurids would be needed to further clarify the relation of the two taxa.

The difficulty of referring isolated carcharodontosaurid material to a specific taxon is further amplified by the possibility of the presence of multiple carcharodontosaurids within a single formation, as is the case in the Huincul Formation, which includes the carcharodontosaurids *Meraxes*, *Mapusaurus* and potentially *Taurovenator* [11, 12, 62], if the latter proofs distinct from either of the former taxa. Beside *Carcharodontosaurus saharicus*, an additional carcharodontosaruid taxon, *Sauroniops*, has been proposed for the Kem Kem Group. *Sauroniops* is based on a single, commercially collected, frontal bone initially described in 2012 [187] and shortly afterwards named by the same authors [188]. The distinction of *Sauroniops* from *Carcharodontosaurus saharicus* was questioned by Ibrahim et al. [18], who pointing out the size differences between the two specimens, found the diagnostic features of *Sauroniops* presented by Cau et al. [187, 188] unconvincing and consequently regarded *Sauroniops pachytholus* as a junior synonym of *Carcharodontosaurus saharicus*. In a paper describing additional proposed carcharodontosaurid material from the Kem Kem formation, Paterna & Cau [189] reviewed the status of *Sauroniops*, further differentiating it from *Carcharodontosaurus saharicus*. Although we do agree that differences between *Sauroniops* and *Carcharodontosaurus saharicus*, as pointed out by Cau et al. and Paterna & Cau [187–189], are intriguing, we prefer the conclusion reached in the initial description of the material by Cau et al. [187], that it is inappropriate to erect a new taxon on this singe frontal bone. Pending more material, we consider *Sauroniops* a nomen dubium.

## Conclusion

Although similar at first glance to the Moroccan *Carcharodontosaurus saharicus*, the Egyptian carcharodontosaurid is here revealed to be a distinct genus and species of carcharodontosaur. *Tameryraptor markgrafi* gen. nov. sp. nov. (Fig 16) increases the currently recognized diversity of carcharodontosaurs in the Cretaceous of Northern Africa and further showcases differences in the faunal compositions of the Egyptian Bahariya Formation and the Moroccan Kem Kem Group. These results therefore contrast the often-proposed uniformity in the theropod faunal composition across seeminlgy similarly aged strata in the middle Cretaceous of North Africa. As a consequence, a closer comparison of other theropods claimed to be cospecific between North African strata, such as *Bahariasaurus*, *Deltadromeus* and *Spinosaurus* is warranted. This topic will be further examined in upcoming papers, one of which is already in preparation. Furthermore, a new clade of allosauroids, Carcharodontosauriformes, is established, enhancing our understanding on the interrelationships within Allosauroidea. Lastly, this research

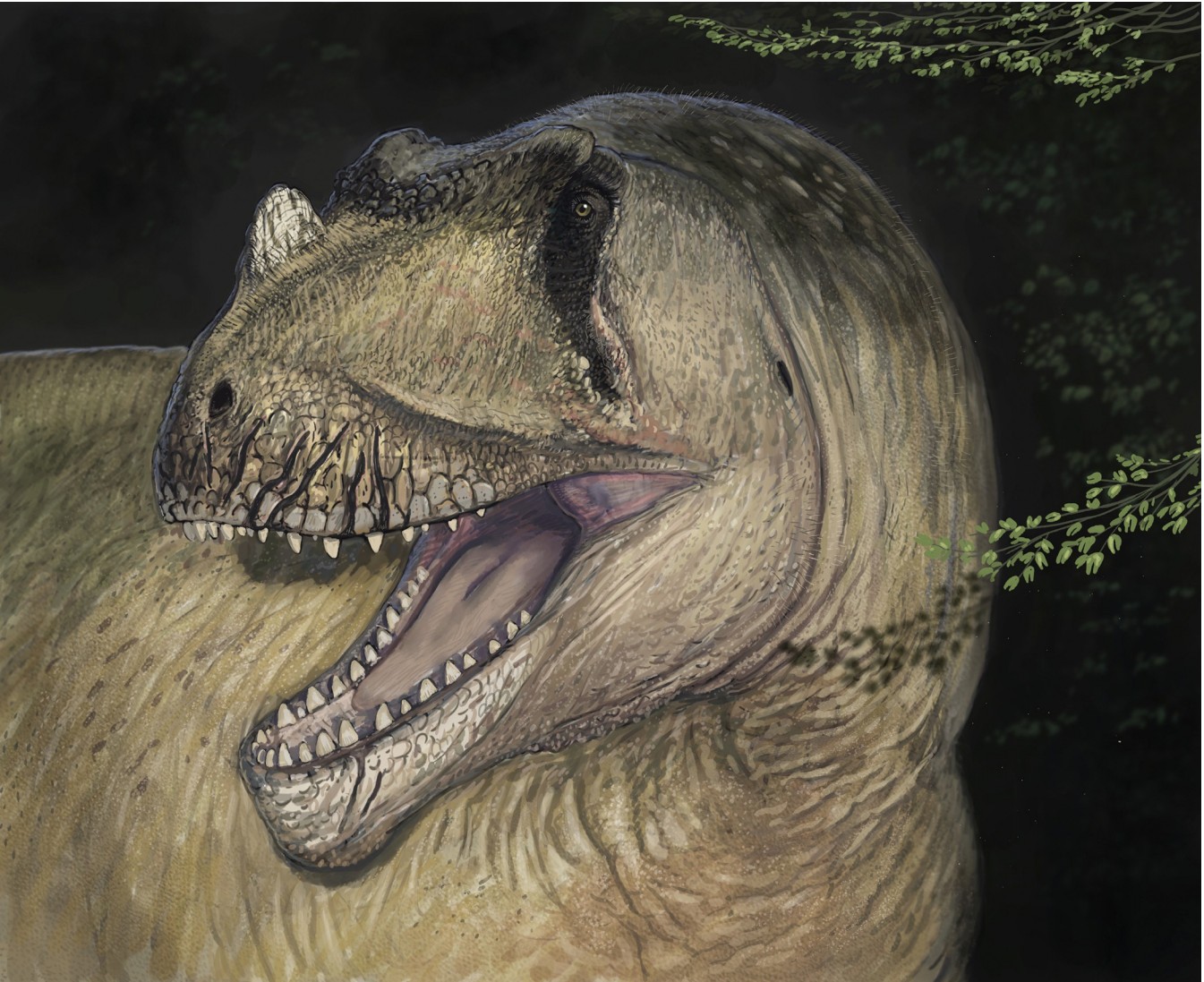

**Fig 16. Life reconstruction of *Tameryraptor markgrafi*.** By Joschua Knüppe.

highlights the problematic nature of naming new genera on highly incomplete material, an issue that was already recognized by Stromer, who, in a forward-thinking attitude for his time wrote the following in his 1934 paper: "However, I do not condone the abuse of creating new generic or specific names from such isolated or entirely incomplete remains, which unfortunately, due to the unsuitable priority rules of nomenclature for paleontology and as a result of its mindlessly pedantic administration, are obliged to serve as the basis for further designations."

## Supporting information

**S1 Dataset. Phylogentic datasets.**
(ZIP)

**S1 File. Phylogenetic results.**
(PDF)

**S1 Table. Select measurements of *Tameryraptor markgrafi* in mm.** * minimum measurement due to incompleteness of element;' measured from photograph or drawings; Data taken from [1].
(DOCX)

**S2 Table. Character support by node for results of pruned analysis in the split dataset (k = 12).** Unambiguous characters are listed with support under accelerated (ACCTRAN) and delayed (DELTRAN) transformations; autapomorphic values are not listed.
(DOCX)

**S3 Table. Character support by node for results of pruned analysis in the merged dataset (k = 12).** Unambiguous characters are listed with support under accelerated (ACCTRAN) and delayed (DELTRAN) transformations. Autapomorphic values are not listed.
(DOCX)

**S1 Fig. Node numbers for S2 and S3 Tables.**
(TIF)

## Acknowledgments

We would like to thank all the other members of the Mesozoic Tetrapod working group for their support and discussions. This paper uses the Mesozoic Tetrapod Theropod Matrix, a large matrix for analyzing theropod relationships that is being developed at the Bayerische Staatssammlung für Paläontologie und Geologie in Munich. Thus, we thank all the other contributors to the Mesozoic Tetrapod Theropod Matrix, mainly Christian Foth, João Kirmse, and Rafael Delcourt. Without the combined effort of all team members, reviewing the codings and formulating character definitions for such a large phylogenetic dataset would not have been possible. We would also like to extend my gratitude to the workers at the Universitäts Archiv Tübigen and Daniela Schwarz from the Museum für Naturkunde in Berlin for providing access to their collections and archival materials. Furthermore, we have to thank Juan Ignacio Canale, Damiano Palombi, and Mattia Antonio Baiano for providing pictures and answering questions about *Meraxes* and *Lajasvenator*. Furthermore, we thank Juan Canale, Rodolfo Coria, Damiano Palombi, and Eduardo Ruigomez for access to carcharodontosaurid material under their care. Without those, accurate scoring in the dataset would not have been possible. Last, but not least, we thank Tom Holtz and two anonymous reviewers, whose critical comments have greatly helped to improve the manuscript.

## Author Contributions

**Conceptualization:** Maximilian Kellermann, Oliver W. M. Rauhut.

**Data curation:** Maximilian Kellermann, Elena Cuesta, Oliver W. M. Rauhut.

**Formal analysis:** Maximilian Kellermann.

**Funding acquisition:** Oliver W. M. Rauhut.

**Investigation:** Maximilian Kellermann, Elena Cuesta, Oliver W. M. Rauhut.

**Methodology:** Maximilian Kellermann, Elena Cuesta, Oliver W. M. Rauhut.

**Project administration:** Oliver W. M. Rauhut.

**Resources:** Oliver W. M. Rauhut.

**Supervision:** Oliver W. M. Rauhut.

**Validation:** Maximilian Kellermann, Elena Cuesta, Oliver W. M. Rauhut.

**Visualization:** Maximilian Kellermann.

**Writing – original draft:** Maximilian Kellermann, Oliver W. M. Rauhut.

**Writing – review & editing:** Maximilian Kellermann, Elena Cuesta, Oliver W. M. Rauhut.

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
