## [Decision Letter · Decision Letter 0]

24 May 2024

PONE-D-24-16876Re-evaluation of the Bahariya Formation carcharodontosaurid (Dinosauria: Theropoda) and its implications for allosauroid phylogenyPLOS ONE

Dear Dr. Kellermann,

Thank you for submitting your manuscript to PLOS ONE. After careful consideration, we feel that it has merit but does not fully meet PLOS ONE’s publication criteria as it currently stands. Therefore, we invite you to submit a revised version of the manuscript that addresses the points raised during the review process.

We look forward to receiving your revised manuscript.

Kind regards,

Olga Spekker, Ph.D.

Academic Editor

PLOS ONE

Journal Requirements:

2. In your manuscript, please provide additional information regarding the specimens used in your study. Ensure that you have reported human remain specimen numbers and complete repository information, including museum name and geographic location. 

For more information on PLOS ONE's requirements for paleontology and archeology research, see https://journals.plos.org/plosone/s/submission-guidelines#loc-paleontology-and-archaeology-research.

3. Please take this opportunity to be sure you have met all of our guidelines for new species. For proper registration of a new zoological taxon, we require two specific statements to be included in your manuscript.

1) In the Results section, the globally unique identifier (GUID), currently in the form of a Life Science Identifier (LSID), should be listed under the new species name, for example:

Anochetus boltoni Fisher sp. nov. urn:lsid:zoobank.org:act:B6C072CF-1CA6-40C7-8396-534E91EF7FBB

Another LSID for the manuscript itself should also appear within the Nomenclature statement. You will need to contact Zoobank (zoobank.org/About) to obtain a GUID (LSID). You should receive one LSID for your manuscript and a separate, unique LSID for the new species. 

2) Please also insert the following text into the Methods section, in a sub-section to be called "Nomenclatural Acts":

The electronic edition of this article conforms to the requirements of the amended International Code of Zoological Nomenclature, and hence the new names contained herein are available under that Code from the electronic edition of this article. This published work and the nomenclatural acts it contains have been registered in ZooBank, the online registration system for the ICZN. The ZooBank LSIDs (Life Science Identifiers) can be resolved and the associated information viewed through any standard web browser by appending the LSID to the prefix ""http://zoobank.org/"". The LSID for this publication is: urn:lsid:zoobank.org:pub: XXXXXXX. 

The electronic edition of this work was published in a journal with an ISSN, and has been archived and is available from the following digital repositories: PubMed Central, LOCKSS [author to insert any additional repositories].

All PLOS ONE articles are deposited in PubMed Central and LOCKSS. If your institute, or those of your co-authors, has its own repository, we recommend that you also deposit the published online article there and include the name in your article.

Following a recent ruling by the International Commission on Zoological Nomenclature, electronic journals are now a valid format for publication of new zoological taxa. In order to ensure the valid publication of your new species, please be sure to include the updated version of Nomenclatural Acts (above). 

A complete explanation of our guidelines for publishing new species can be found on our website: http://www.plosone.org/static/guidelines#zoological.

4. Thank you for stating the following in your Competing Interests section: "There is no competing interest"

6. We note that Figure 1 in your submission contain map images which may be copyrighted. All PLOS content is published under the Creative Commons Attribution License (CC BY 4.0), which means that the manuscript, images, and Supporting Information files will be freely available online, and any third party is permitted to access, download, copy, distribute, and use these materials in any way, even commercially, with proper attribution. For these reasons, we cannot publish previously copyrighted maps or satellite images created using proprietary data, such as Google software (Google Maps, Street View, and Earth). For more information, see our copyright guidelines: http://journals.plos.org/plosone/s/licenses-and-copyright.

1) You may seek permission from the original copyright holder of Figure 1 to publish the content specifically under the CC BY 4.0 license.  

2) If you are unable to obtain permission from the original copyright holder to publish these figures under the CC BY 4.0 license or if the copyright holder’s requirements are incompatible with the CC BY 4.0 license, please either i) remove the figure or ii) supply a replacement figure that complies with the CC BY 4.0 license. Please check copyright information on all replacement figures and update the figure caption with source information. If applicable, please specify in the figure caption text when a figure is similar but not identical to the original image and is therefore for illustrative purposes only.

7. We are unable to open your Supporting Information file [S5 Phylogenetic Dataset.zip]. Please kindly revise as necessary and re-upload.

**Additional Editor Comments:**

Dear Dr. Kellermann,

We appreciate you submitting your manuscript to PLOS ONE and thank you for giving us the opportunity to consider your work.

I have completed my evaluation of your manuscript, which has been reviewed by two highly qualified reviewers all of whom agree it is worth to be published in PLOS ONE. Nevertheless, they have suggested some changes that will help to improve the paper.

Therefore, I invite you to resubmit your manuscript after addressing the reviewers’ comments below. When revising your manuscript, please consider all issues mentioned in the reviewers' comments carefully: please, outline every change made in response to their comments and provide suitable rebuttals for any comments not addressed. Please, note that your revised submission may need to be re-reviewed.

PLOS ONE values your contribution and I look forward to receiving your revised manuscript.

Yours sincerely,

Dr. Olga Spekker

Reviewers' comments:

Reviewer's Responses to Questions

**Comments to the Author**

1. Is the manuscript technically sound, and do the data support the conclusions?

Reviewer #1: Yes

Reviewer #2: Yes

2. Has the statistical analysis been performed appropriately and rigorously? 

Reviewer #1: N/A

Reviewer #2: Yes

3. Have the authors made all data underlying the findings in their manuscript fully available?

Reviewer #1: Yes

Reviewer #2: Yes

4. Is the manuscript presented in an intelligible fashion and written in standard English?

Reviewer #1: Yes

Reviewer #2: Yes

5. Review Comments to the Author

Reviewer #1: In this manuscript, the authors re-evaluate the status of a large-bodied predatory dinosaur skeleton described in 1931 and destroyed in 1944 during WWII. The authors also introduce a novel phylogenetic analysis of the predatory dinosaurs which adds - among others - new elements on the relationships among the large-bodied group Allosauroidea (i.e., the lineage including the specimen destroyed in 1944).

The phylogenetic analysis and the implications of that analysis are well-argumented and would surely improve our knowledge on the evolution of these dinosaurs, and I do not request any significant edit of that section of the manuscript (see just a few notes and comments in the pdf attached to this revision). Yet, I have several concerns about the taxonomic interpretation of the particular specimen lost in 1944, which is claimed by the authors to represent a new genus and species ("Alllissaurus markgrafi") and has been removed from the well-known taxon Carcharodontosaurus saharicus.

Since my concerns are both epistemological and methodological and stem from the logical and chronological sequence of the events involved, I first provide, below, a brief chronology of them (see the manuscript for the details of each step).

1) Depéret and Savornin (1925-27) erected Megalosaurus saharicus, based on two non-associated teeth from the "mid-Cretaceous" of Algeria.

2) Stromer (1931) erected Carcharodontosaurus, based on a partial skeleton from the "mid-Cretaceous" of Egypt. Stromer considered Carcharodontosaurus and "Megalosaurus saharicus" the same species and diagnosed Carcharodontosaurus saharicus from the Egyptian specimen.

3) The Egyptian specimen was destroyed during WWII.

4) Sereno et al. (1996) described a partial skull from the "mid-Cretaceous" of Morocco and referred it to Carcharodontosaurus saharicus, based on Stromer (1931).

5) Brusatte and Sereno (2007) explicitly designated the Moroccan specimen as neotype of C. saharicus. This decision was based on the following arguments: 5a) the two teeth described by Depéret and Savornin (1925-27) likely pertain to different individuals; 5b) the diagnostic tooth characters originally cited are no longer diagnostic at the specific or generic level; and 5c) both Egyptian and Algerian specimens cannot be used as type material; 5d) the Moroccan specimen "is identical to both the original holotypic teeth and Stromer’s Egyptian material".

In this new manuscript, the authors introduce two recently rediscovered original photos of the Egyptian specimen prior to its destruction, and list a series of features which could differentiate that material from the Moroccan specimen. Despite the specimen being destroyed, the authors consider the published material sufficient for erecting a new genus and species, distinct from Carcharodontosaurus saharicus. Note that these statements challenge the statements 5c) and 5d) by Brusatte and Sereno (2007).

The thesis discussed in the manuscript is thus the following:

6) The Moroccan specimen does not belong to the same genus and/or species of the Egyptian specimen.

7) Even if the Egyptian specimen is lost, its description and the new photographs justify it as holotype for the formal erection of a genus and species .

Both 6) and 7) are problematic, and I encourage the authors to reconsider them carefully:

Statement 7) is explicitly challenged by the International Code of Zoological Nomenclature (ICZN, 1999), Recommendation 73B: "An author should designate as holotype a specimen actually studied by him or her, not a specimen known to the author only from descriptions or illustrations in the literature". The Egyptian material, actually lost in 1944, should not be used to erect any new taxon.

The ICZN recommendation must be followed for a series of epistemological reasons. The first is the non-reproducibility of the observations reported in support of the new species: the irreversible destruction of the specimen prevents any objective evaluation of the diagnostic features introduced. Accordingly, any emendation of the diagnosis cannot be falsified. The second is the non-testable nature of a series of ambiguous interpretations of the published photos used by the authors to suggest some features not explicitly reported by Stromer (1931).

For example, the authors wrote:

"An expansion of the narial fossa to the nasal in the form of a supranarial fossa, as it is present

in Acrocanthosaurus, Concavenator, Carcharodontosaurus and Meraxes [11,18,43,44] is not

clearly visible in the photo of SNSB-BSPG 1922 X 46, but Stromer [1] described the area of

the anterior end of the nasal lateral to the median premaxillary process as slightly concave.

However, a clearly defined rim of such a fossa, as it is present in Carcharodontosaurus ([18]:

Fig 134A) is not at all indicated in Stromers figures, nor visible on the photograph."

The authors implicitly remark that the development of key features is ambiguous once the literature is compared with the photos/illustration. What should be considered the valid source? In this case, the narial fossa is described by Stromer (1931) as a slight concavity, yet such feature is not illustrated in the figures/photos.

The authors also wrote:

"The antorbital fossa extends onto the lateral surface of the nasals, which represents an

allosauroid synapomorphy [7,47], and is clearly visible in lateral view on the nasals of SNSBBSPG

1922 X 46. This is visible in the photograph of the specimen and also indicated in

Stromer’s reconstruction of the skull (Figs 2 and 4B, D)"

Contrary to the authors, I hardly differentiate an antorbital fossa of the nasal in the photo of the Egyptian specimen from the homologous part in the Moroccan specimen. The description by Stromer (1931) is ambiguous on such feature, so in this case we have to trust the interpretation of the text and figures by the authors. Note that the lack of such fossa is in agreement with the other late-diverging carcharodontosaurids. Maybe, I am over-conservative and prefer not speculate on elements of an old photo which was not explicitly created for illustrating anatomical details. The authors might be right in their interpretation of the images, but without direct observation of the specimen, there is no way for eventually falsifying their statement. If they are wrong, such error is fixed in their diagnosis of the taxon and could not be corrected by any independent observation. The drawings by Stromer do not match all details claimed from the photos, so what of the two images should we trust? Some features claimed by the authors could be non-biological in origin, due to taphonomy or damage, or even just photographic artifacts. If the authors have "over-interpreted" some features in the only available images of the now destroyed Egyptian specimen, nobody could falsify them. Thus, the diagnosis of "Alllissaurus markgrafi" is epistemologically problematic and cannot be considered a proper scientific statement.

Stated in a simpler way: if the authors are wrong in some interpretations, how could them be falsified without an available specimen? A couple of old black-and-white photos alone (one of which clearly was not meant to illustrate anatomical details but just the whole mounted material) is not sufficient evidence that all features listed by the authors as diagnostic are actually natural apomorphies.

Other features of the Egyptian specimen, even if genuine, could not be significant from a taxonomic perspective. For example, the "nasal horn" and the more extensive maxillary subcutaneous ornamentation (claimed to be absent or less developed in the Moroccan specimen) could be ontogeny-related and/or sexually-dimorphic features widespread among the carcharodontosaurids once a richer sample is available.

Statement 6) is an even more complex issue. The authors apparently seem unaware of the full implications of their main hypothesis: if we accept that the lost Egyptian specimen could be used as valid type of a taxon, and also endorse the taxonomic distinction of the Egyptian and Moroccan specimens, this has relevant nomenclatural implications:

First, if the Moroccan specimen does not belong to the same taxon as the Egyptian specimen, then the rationale for erecting a neotype of C. saharicus by Brusatte and Sereno (2007) from the former specimen is no more valid. In fact, the neotype was introduced under the assumption that all specimens mentioned belong to a single taxon. If the two specimens belong to different taxa, one cannot be used as the neotype which replaces the destruction of the other: the neotype sensu Brusatte and Sereno (2007) is thus invalidated! Furthermore, if the lost Egyptian specimen is now considered valid as holotype of "Alllissaurus", then it means that it has always been the valid type for a carcharodontosaurid taxon lacking a neotype: in that case, it has always been the most appropriate neotype of Carcharodontosaurus in spite of the Moroccan specimen, because Stromer (1931) used it to erect and diagnose that genus. Accordingly, the genus name "Alllissaurus" is automatically a junior synonym of the genus name Carcharodontosaurus.

The authors might reply that Carcharodontosaurus is now anchored to the Moroccan specimen, but such nomenclatural act was historically based and conditioned to a taxon sample the authors here have just demonstrated being wrong. They wrote: "In the interest of preserving the widely used genus Carcharodontosaurus and family Carcharodontosauridae, we concur with the designation of SGM-Din 1 as the neotype of Carcharodontosaurus saharicus sensu Brusatte and Sereno". Yet, such statement is contradicted by the main argument of their manuscript, because the authors have just (implicitly) demonstrated that the rationale for designating SGM-Din 1 [the Moroccan specimen] as valid neotype for Carcharodontosaurus is no more valid, because the Moroccan and Egyptian specimens do not belong to the same genus.

The best way to preserve both Carcharodontosaurus and Carcharodontosauridae should be to designate the Egyptian specimen as neotype of C. saharicus because:

1) The Egyptian specimen - assumed by the authors as a potentially valid type specimen for a taxon - has been the explicit source for the diagnosis of Carcharodontosaurus since the erection of the genus in 1931, whereas the Moroccan specimen was introduced only in 1996 and has been (arbitrarily) considered the neotype of the same taxon of the Egyptian specimen only in 2007 to replace the destroyed material.

2) All literature prior to 2007 assumed the Egyptian specimen as the main source of Carcharodontosaurus diagnosis (and, so far, it keeps being the only available source for all postcranial features of this taxon): even the referral of the Moroccan specimen to that taxon (regardless of it being right or wrong) was merely a consequence of comparing it with the Egyptian material. Removing the latter from Carcharodontosaurus is everything but an act "preserving" Carcharodontosaurus stability and consistency.

There is another reason which justifies the removal of the Moroccan specimen from C. saharicus instead of the Egyptian one: as stated above, according to the ICZN Recommendation 73B, "an author should designate as holotype a specimen actually studied by him or her, not a specimen known to the author only from descriptions or illustrations in the literature". Given that, of the two specimens, only the Moroccan one is currently housed in a public institution, only the latter could be eventually studied by a researcher, and so it is the only one of the two which is available if a second taxon is eventually split from the Carcharodontosaurus hypodigm.

A fourth reason for challenging the exclusion of the Egyptian specimen from Carcharodontosaurus is that the support of the phylogenetic results is weak. As shown in detail by the authors, when compared to the shortest scenario found by the analyses, only two additional steps are necessary for producing a Carcharodontosaurus saharicus node including both Egyptian and Moroccan specimens, and only three additional steps for reconstructing a node with all Carcharodontosaurus specimens in the monophyletic genus (both C. saharicus and C. iguidensis material). I replicated the analyses and confirm the results. Such small step differences are not statistically significant once evaluated with the Templeton test: we cannot exclude that they are merely due to lack of overlapping data and/or even a few coding artifacts (eventually biased by some miscoding due to misinterpretation of the published source). This means that the monophyly of Carcharodontosaurus keeps being a potentially valid topology. I would not consider the phylogenetic results sufficiently robust for supporting the splitting of the Egyptian specimen from Carcharodontosaurus. The radical taxonomic revision of a key historical specimen based on such weak results is thus poorly supported and should be avoided.

In conclusion:

Nomenclatural and epistemological reasons strongly discourage the erection of a new taxon from a specimen now lost and never studied directly by anyone after 1944.

The ICZN explicitly recommends to not erect a taxon like this.

If the Egyptian and Moroccan specimens belong to distinct genera, then the rationale followed in 2007 for designating the latter as "neotype" of Carcharodontosaurus is no more valid. Note that if the authors are correct in differentiating the Egyptian and Moroccan morphs, then "Carcharodontosaurus sensu Brusatte and Sereno 2007" is a chimera based on two (or more) taxa, and the only valid (non chimerical) "version" of Carcharodontosaurus is only the one prior 1996, based uniquely on the Egyptian specimen: any argument used in 2007 to define the Moroccan specimen as the neotype of the same taxon of the Egyptian specimen is thus no more valid, and cannot be used to justify the removal of the Egyptian specimen from the genus which was explicitly erected and diagnosed from its morphology.

If the authors are right in the taxonomic distinction of the two specimens, then the Moroccan specimen was not Carcharodontosaurus since the beginning: it could be provisionally referred to an indeterminate carcharodontosaurid species (and mentioned by its collection number, like other OTUs used by the authors in their phylogenetic analysis), pending an explicit revision of its status and that of "C. iguidensis", based on direct first-hand analysis. Removing the Egyptian specimen from Carcharodontosaurus would arbitrarily invalidate over 90 years of literature based on the taxon erected by Stromer in 1931, a result that cannot be endorsed just in order to conserve an arbitrary neotype definition which the new manuscript itself has demonstrated being based on wrong assumptions.

The phylogenetic test showed that only a few steps differentiate the splitting of all specimens from the traditional lumping of them into Carcharodontosaurus. Thus, pending more robust and less problematic evidence, any taxonomic revision should be avoided. Terms like "cf. Carcharodontosaurus" or "?Carcharodontosaurus" are valid terms for the material from Morocco and Niger to remark the potential non-monophyletic nature of that name until a consensus is reached.

The authors should thus remark the contradiction in Brusatte and Sereno (2007) relative to their new study, should state the problematic use of the Moroccan specimen as "neotype" of what actually was a chimaera, have to moderate the interpretation of the features based only on the photos of the lost fossil, and must avoid the creation of an intrinsically problematic taxon which is explicitly challenged by the ICZN.

Note that the taxonomic section of the manuscript lacks any record of ZooBank registrations, making any nomenclatural act - regardless of their substantial validity - as formally invalid (e.g., if the manuscript is accepted in the current version, "Alllissaurus markgrafi" is a nomen nudum).

Reviewer #2: This paper seeks to correct a taxonomic conundrum hiding in plain sight but which the community was essentially unaware: the fact that Stromer’s Egyptian material of Carcharodontosaurus saharicus (since destroyed) was likely not referrable to that Algerian tooth taxon at all. The authors provide previously unpublished photographic evidence of the original type, which allows a more detailed comparison to fossils discovered in the late 20th and 21st Century. This comparison reveals autapomorphies to the Egyptian dinosaur, requiring a new taxon name.

The authors have done an excellent review of the (oh so many!) questionable referral of various bones to taxa without overlapping material, when the different bones are sometimes separated by kilometers. If the information in p. 10 ¶3 could be summarized in a table (in particular, the OTU used in each “separate” and “merged” configuration), that would be appreciated.

But please note that they themselves repeat an invalid referral by taking some of Stromer’s writings at their word. Stromer’s (1931, 1934) comparisons of various African dinosaurs to “Dryptosaurus” is not to the New Jersey tyrannosauroid Dryptosaurus aquilunguis; rather, it is to the dinosaur universally accepted today as Albertosaurus sarcophagus (but which was in the late 19th and early 20th Century referred to the tooth taxon Laelaps/Dryptosaurus incrassatus. See Stromer’s footnote on the bottom of page 56 in his 1934 paper:

“Nach PARKS (1928, S. 5-7) ist Dryptosaurus LAMBE (1904, S. 20, Raf. VI, Fig. 6) warhscheinlich mit Albertosaurus sarcophagus OSBORN identisch.”

And his reference in the 1931 paper to the North American Dryptosaurus refer to Lambe’s 1904 paper of the skulls which became the type and coptype of Albertosaurus sarcophagus in Osborn 1905. Thus, in this paper of confusing theropod taxonomy, some proper clarification should be made that Stromer’s use of “Dryptosaurus” is actually Albertosaurus. (See p. 20 ¶3)

(Parenthetically, it seems bizarre to me that Depéret and Savornin included the most labiolingually compressed large theropod teeth known into the same genus as dinosaurs with the most labiolingually expanded teeth (source of the trivial nomen “incrassatus”: thickened or inflated)).

An aside: I appreciate the evocative name Alllissaurus (marauder reptile), but note in advance that this name is destined to be consistently misspelled (triple “l”s are rare), and will almost assuredly be autocorrected (either in word processors or web searches) to “Allosaurus”! This isn’t a reason not to use the name by any means, but simply looking ahead

p. 35, ¶3 The authors should make clear the definition of “Allosauroidea” they are using. I am assuming (given the authorship of this manuscript) they follow Rahut and Pol (2019), namely the most inclusive clade containing Allosaurus fragilis but not Megalosaurus bucklandii or Neornithes. Some previous node-based definitions would be potentially synonymous with Allosauria or the new Carcharodontosauriformes depending on the configuration of the allosaurs they found.

Cladograms: Note that Alpkarakush has yet to be published; please make certain of the timing of the publication of this paper vs. the Alpkarakush paper so that the authors do not trump themselves.

6. PLOS authors have the option to publish the peer review history of their article (what does this mean?). If published, this will include your full peer review and any attached files.

Reviewer #1: No

Reviewer #2: **Yes: **Thomas R. Holtz, Jr.

---

## [Author Response · Author response to Decision Letter 0]

12 Jul 2024

We kindly thank the reviewers for their critical and helpful comments, which have greatly helped to improve the manuscript. Most of the minor changes (especially those included in the annotated manuscript) were incorporated, and we will thus here offer mainly comments on some of the more severe criticisms and comments that we do not agree with.

Reviewer 1:

We will first reply to the reviewer’s main concern in general, and later to some of his or her sub-arguments in detail, as much of this is relevant for the question whether to base a new taxon on the now destroyed Egyptian material. For clarity, we will not copy all of the reviewers statements, but only those that we deem important for the main conclusions. Omitted parts are indicated by “…”

Yet, I have several concerns about the taxonomic interpretation of the particular specimen lost in 1944, which is claimed by the authors to represent a new genus and species ("Alllissaurus markgrafi") and has been removed from the well-known taxon Carcharodontosaurus saharicus.

…

5) Brusatte and Sereno (2007) explicitly designated the Moroccan specimen as neotype of C. saharicus. This decision was based on the following arguments: 5a) the two teeth described by Depéret and Savornin (1925-27) likely pertain to different individuals; 5b) the diagnostic tooth characters originally cited are no longer diagnostic at the specific or generic level; and 5c) both Egyptian and Algerian specimens cannot be used as type material; 5d) the Moroccan specimen "is identical to both the original holotypic teeth and Stromer’s Egyptian material".

…

Despite the specimen being destroyed, the authors consider the published material sufficient for erecting a new genus and species, distinct from Carcharodontosaurus saharicus. Note that these statements challenge the statements 5c) and 5d) by Brusatte and Sereno (2007).

…

6) The Moroccan specimen does not belong to the same genus and/or species of the Egyptian specimen.

7) Even if the Egyptian specimen is lost, its description and the new photographs justify it as holotype for the formal erection of a genus and species .

Statement 7) is explicitly challenged by the International Code of Zoological Nomenclature (ICZN, 1999), Recommendation 73B: "An author should designate as holotype a specimen actually studied by him or her, not a specimen known to the author only from descriptions or illustrations in the literature". The Egyptian material, actually lost in 1944, should not be used to erect any new taxon.

…

The best way to preserve both Carcharodontosaurus and Carcharodontosauridae should be to designate the Egyptian specimen as neotype of C. saharicus because:

In regards to statement 5): 

Although the reviewer is correct with his/her historic review of the history of Carcharodontosaurus, we disagree with his/her later stated consequences from these findings. First, please note that Brusatte & Sereno (2007) explicitly state that the original syntypic teeth of Megalosaurus saharicus (named “holotype” in their paper) are not diagnostic, but they do not discuss anything about the potential diagnosabilty of the Egyptian material. Thus, the reviewers conclusion 5c (both Egyptian and Algerian specimens cannot be used as type material) is not based on an actual statement in this work, but was apparently deduced from the fact that Brusatte & Sereno chose the Moroccon skull (the only material still in existence) as neotype. However, this choice by Brusatte & Sereno is based on their assumption that the original type teeth, the Egyptian material and the Moroccon skull belong to the same taxon (the reviewer’s point 5d). The main point of our work is the interpretation that the Egyptian material represents a taxon distinct from the Moroccon skull. If our interpretation is correct (and we agree with Brusatte & Sereno that the original syntypic teeth are not diagnostic and probably lost), we are left with four possibilities

1) We ignore the Egyptian material, as it has been destroyed.

This is not really an option, as the material described by Stromer (1931, 1934, 1936) has been referred to repeatedly, especially in the more recent literature on Cretaceous Gondwanan theropods, and multiple authors have found Stromer’s descriptions and figures to be useful for anatomical and often even taxonomic comparisons and systematic/phylogenetic inferences. Thus, the interpretation that this material might represent a taxon different from other reference material currently referred to the same taxon is of interest to theropod workers, and simply ignoring it would perpetuate possible errors in the literature (e.g. basing phylogenetic codings on a mixture of the Moroccan skull and the material described by Stromer). 

2) We change the neotype for Carcharodontosaurus saharicus from the Moroccon skull proposed by Brusatte & Sereno (2007) to the Egyptian specimen. 

Although this seems to be a preferred option for the reviewer (see also below), we consider it to be a worse possibility than the one we adopted for several reasons. First, the designation of the Egyptian specimen as neotype for Carcharodontosaurus saharicus would only shift the focus of one of the main concerns of the reviewer, namely that a taxon should not be diagnosed on the basis of material that is no longer in existence. If we were to use the Egyptian material as neotype, we also would need to give a diagnosis for Carcharodontosaurus saharicus, based on Stromer’s descriptions and the existing illustrations and figures – the same as for the erection of a new taxon. 

Second, the designation of the Egyptian material as neotype would directly violate two of the qualifying conditions for neotype designation laid out by the ICZN (articles 75.3.6 and 75.3.7). Article 73.3.6 states that a requirement needed is “evidence that the neotype came as nearly as practicable from the original type locality … and, where relevant, from the same geological horizon or host species as the original name-bearing type”. The Egyptian specimen came from a locality approximately 2800 km distant from the original type locality of Megalosaurus saharicus at Timimoun in Algeria, and the beds there were originally assigned to the Albian by Depéret & Savognin (1925, 1927), whereas the Baharyia Formation has since Stromer’s times been considered to be Cenomanian in age. In contrast, the locality in the Douira Formation from where the skull that was made the neotype by Brusatte & Sereno (2007) was collected, is only about 470 km away from Timimoun and is placed in sediments that are generally (at least roughly) correlated with the rocks outcropping at the type locality (see Lapparent 1960; Mateer et al. 1992; Ibrahim et al. 2020); the interpretation that all of these units are Cenomanian in age (as the Baharyia Formation) is actually based on faunal comparisons with the Baharyia-Formation – which we argue to be problematic in the discussion of our paper. Article 75.3.7 states that “…the neotype is, or immediately upon publication has become, the property of a recognized scientific or educational institution…”. As the material from Egypt is lost, it cannot be placed in a recognized scientific or educational institution – thus precluding it from becoming a neotype on this basis alone (unless, again, one considers the descriptions and illustrations by Stromer and the photographs in the BSPG and UT archives as a valid type – in which case there is no reason why not to base a new taxon on these). 

Third, the qualifying condition of article 73.3.5 “evidence that the neotype is consistent with what is known of the former name-bearing type from the original description and from other sources” is also problematic for the Egyptian material. In this context, it might be important to point out that Stromer (1934: p. 62) designated the smaller tooth described by Depéret & Savornin (1925, 1927) as “type” of Carcharodontosaurus saharicus, which we consider to be a valid lectotype designation according to article 74.5 of the ICZN. Stromer (1931: p. 9) compares this tooth with a tooth that was found isolated in the locality of his specimen, which he thought to represent the same taxon, namely Carcharodontosaurus saharicus. However, he also pointed out that this tooth differed in the stronger recurvature from the teeth found in situ in the maxilla of the specimen he described. Given that the general features of carcharodontosaurid teeth are now known to be not diagnostic on species or even genus level, it cannot be ruled out that the isolated tooth found with Stromer’s specimen came from another carcharodontosaurid taxon scavenging on the carcass. Thus, there is a – admittedly small and of currently unknown taxonomic significance – difference in curvature between the type tooth of Carcharodontosaurus saharicus and the teeth found in situ in the Egyptian specimen. In contrast, the smaller tooth described and figured by Depéret & Savornin (1927: pl. 13, fig, 1) is in complete agreement with teeth found in the Moroccan skull.

Fourth, as we consider the neotype designation of the Moroccan specimen by Brusatte & Sereno (2007) to be valid (see below), a new neotype designation would directly violate article 75.4 of the ICZN: “The first neotype designation published for a nominal species-group taxon in accordance with the provisions of this Article is valid and no subsequent designation, except one made by the Commission under the plenary power…, has any validity”.

Thus, we reject this option for all the reasons outlined above.

3) We leave the Moroccon skull as neotype for Carcharodontosaurus saharicus, simply state that we think that the Egyptian material represents a different taxon, but do not propose a new binomen for it.

This would be an option, but we consider it less good than the one we chose. If one considers Stromer’s descritpions (which are quite detailed for its time) and his illustrations to be accurate, the material from Egypt is clearly diagnosable. That Stromer’s descriptions and figures are accurate is supported by his descriptions of other materials that are still around and can be compared with the descriptions, such as the type specimens of Lybicosuchus and Aegyptosuchus (both kept in the BSPG collections), and by the newly found photographs of the materials that show many of the features that he describes and generally agree with the drawings of his publication. Thus, we would be left with a clearly diagnosable new taxon without a formal taxonomic name. This we consider problematic for two rather practical reasons. First, a named taxon receives considerably more attention than an unnamed taxon, and as the Egyptian material currently represents the only diagnosable carcharodontosaur from Africa east of the inner-Saharian seaway, it is an important reference for Cretaceous Gondwanan theropod faunas. Second, if one considers the material diagnosable (for which we, hopefully, make a strong case), it will be only a matter of time until the taxon will be named, if not by us then by subsequent authors who might not be aware or not agree with the reasons not to name a new taxon on diagnosable material, even if it is no longer around. 

4) We leave the Moroccon skull as neotype for Carcharodontosaurus saharicus and propose a new taxon for the (now destroyed) Egyptian material.

This is the option we chose, for all the reasons explained above. As the neotype designation by Brusatte & Serene has to be regarded as valid (see below) and we regard the Egyptian material as clearly distinct from other carcharodontosaurids and also diagnostic, this seems to be the best option.

Statement 7) is explicitly challenged by the International Code of Zoological Nomenclature (ICZN, 1999), Recommendation 73B: "An author should designate as holotype a specimen actually studied by him or her, not a specimen known to the author only from descriptions or illustrations in the literature". The Egyptian material, actually lost in 1944, should not be used to erect any new taxon.

The ICZN recommendation must be followed for a series of epistemological reasons. The first is the non-reproducibility of the observations reported in support of the new species: the irreversible destruction of the specimen prevents any objective evaluation of the diagnostic features introduced. Accordingly, any emendation of the diagnosis cannot be falsified. The second is the non-testable nature of a series of ambiguous interpretations of the published photos used by the authors to suggest some features not explicitly reported by Stromer (1931).

As noted by the reviewer himself, the ICZN statement is a recommendation, not a binding article. Of course, it is always preferable to base taxa on material that can be studied firsthand and that is available for subsequent study. However, as this material has been destroyed a very long time ago, this is obviously not an option in this case. Nevertheless, regardless of the recommendation cited by the reviewer, ICZN code 73.1.4 generally allows description of a new nominal taxon based on previous descriptions and illustrations, and there are numerous examples both in palaeontology and biology where this has been done. Thus, in dinosaurs, for example, the sauropod taxon Nopcsaspondylus alarconensis Apesteguía, 2007, was based on a single sauropod vertebra described and figured in three views by Nopcsa (1902) and was apparently lost long before the erection of a new taxon on it. Similarily, Amphicoelias fragillimus Cope, 1878, was described on a single neural arch for which only a field sketch existed, and Carpenter (2018) even changed the generic name for this taxon to Maraapunisaurus based on this illustration alone. 

More commonly, new species are described on specimens that might still have been studied by the authors but were not collected or lost prior to the description of a new species. Thus, for example, the Devonian lungfish Apatorhynchus opistheretmus Friedman and Daeschler, 2006, is based on a partial skull roof that was never collected, so that only photographs and a cast exist. More recently, Brum et al. (2021) described the unenlagiine theropod Ypupiara lopai on the basis of a partial maxilla that was destroyed previously in the fire that destroyed the Museu Nacional in Rio de Janeiro in 2018. Furthermore, in some types of fossils, for example vertebrate tracks, it is common practice that the actual specimens are not collected, but new ichnotaxa are described on the basis of photographs, photogrammetry or casts.

In all of these cases, the epistemological reason raised by the reviewer, that the lack of the actual specimen precludes verification of the finds, would also apply. Nevertheless, it is generally considered to be acceptable that the documentation presented in the papers is sufficient to characterize the new taxon. In our case, the anatomical interpretations and diagnostic features are all based on the description by Stromer, the detailed drawings included in his publications, and the newly found photographs, and thus, all of our statements can be verified or contested on the basis of the same data. We have been careful in our interpretations of the available data and have always indicated uncertainty whenever it is present in our descriptions. Whenever this uncertainty is not stated, our observations can be clearly verified from the descriptions, figures or images of the specimen, or a combination of either sources (see comments on features contested by the reviewer below). Furthermore, the ICZN gives recommendations for cases where the designation of a new taxon based on description and illustrations of a non-existing type, which we followed:

“Recommendation 73G. Specific reasons for designation of an unpreserved specimen as the name-bearing type. An author should provide detailed reasoning why at least one preserved specimen, whether a complete individual organism or a part of such an individual, was not used as the name-bearing type for the new taxon and why the formal naming of the taxon is needed at a point in time when no preserved name-bearing type will be available.” 

- As can be shown from Stromer’s descriptions and illustrations, the Egyptian carcharodontosaurid is distinct from any other known carcharodontosaur species. As this was the only sp

---

## [Decision Letter · Decision Letter 1]

2 Aug 2024

PONE-D-24-16876R1Re-evaluation of the Bahariya Formation carcharodontosaurid (Dinosauria: Theropoda) and its implications for allosauroid phylogenyPLOS ONE

Dear Dr. Kellermann,

Thank you for submitting your manuscript to PLOS ONE. After careful consideration, we feel that it has merit but does not fully meet PLOS ONE’s publication criteria as it currently stands. Therefore, we invite you to submit a revised version of the manuscript that addresses the points raised during the review process.

**ACADEMIC EDITOR:**

Dear Dr. Kellermann,

We appreciate you submitting your manuscript to PLOS ONE and thank you for giving us the opportunity to consider your work.

I have completed my evaluation of your manuscript, which has been reviewed by two highly qualified reviewers all of whom agree it is worth to be published in PLOS ONE. However, I agree with Reviewer 1 that the taxonomic part erecting a new species is problematic. Therefore, I kindly ask you to review that part and resubmit your manuscript after addressing Reviewer 1’s comments below. Please, note that your revised submission may need to be re-reviewed.

PLOS ONE values your contribution and I look forward to receiving your revised manuscript.

Yours sincerely,

Dr. Olga Spekker

We look forward to receiving your revised manuscript.

Kind regards,

Olga Spekker, Ph.D.

Academic Editor

PLOS ONE

Reviewers' comments:

Reviewer's Responses to Questions

**Comments to the Author**

1. If the authors have adequately addressed your comments raised in a previous round of review and you feel that this manuscript is now acceptable for publication, you may indicate that here to bypass the “Comments to the Author” section, enter your conflict of interest statement in the “Confidential to Editor” section, and submit your "Accept" recommendation.

Reviewer #1: (No Response)

Reviewer #2: (No Response)

2. Is the manuscript technically sound, and do the data support the conclusions?

Reviewer #1: Partly

Reviewer #2: Yes

3. Has the statistical analysis been performed appropriately and rigorously? 

Reviewer #1: N/A

Reviewer #2: Yes

4. Have the authors made all data underlying the findings in their manuscript fully available?

Reviewer #1: Yes

Reviewer #2: Yes

5. Is the manuscript presented in an intelligible fashion and written in standard English?

Reviewer #1: Yes

Reviewer #2: Yes

6. Review Comments to the Author

Reviewer #1: I thank the authors for the detailed reply to my comments.

I am sorry for not being completely able to follow their final conclusion, but the epistemological issues I remarked in the first review have not been addressed in a satisfactory way, and thus I cannot endorse the erection of a new genus/species based on the material destroyed in 1944.

The authors are right in listing four possible solutions to the controversy. They (the authors) have been very rigorous in following the ICZN rules, and I applaude their detailed argument. Yet, a recommendation (in this case, to not erect a new taxon) is not followed/ignored because of the Code itself, but on the basis of the Precautionary Principle, which is logically and epistemologically superior to the code itself. Accordingly, the only conclusion I endorse is the #3 among those listed by the authors (quoted below):

"We leave the Moroccon skull as neotype for Carcharodontosaurus saharicus, simply state that we think that the Egyptian material represents a different taxon, but do not propose a new binomen for it.

Just because the Egyptian material cannot be referred to the same taxon of the Moroccan material (a conclusion I can endorse) does not automatically mean that the former could be used to erect a new taxon. The authors have not addressed the main question of my review: what if the species they erect was not a real taxon, and based on erroneous interpretation of Stromer's publications? What if Stromer's words used to define such novel taxon are (even in part) erroneous? How could such taxon be falsified/revised without a testable specimen to work with? Any new species is a particular taxonomic hypothesis anchored to a testable type specimen. This is the reason for introducing type specimens housed in valid institutions: to avoid the proliferation of untestable taxonomic hypotheses. Without a testable specimen, any detail of such hypothesis is in an epistemological limbo, because we cannot replicate Stromer's observations used by the authors to erect that taxon. If two distinct authors disagree on some of the statements by Stromer used here to define the new species, how could such disagreement be solved? The objectivity principle dictates replicabile statements based on accessible specimens: in absence of such specimen, and pending the discovery of a second specimen supporting Stromer's description, no new species should be erected.

I invite the authors to follow the most conservative and most robust conclusion among those they listed, the less speculative and the one with potentially the less negative impacts: to discuss the possible differences between the Egyptian and Moroccan carcharodontosaurs according to the published material, but to avoid the formal erection of a new taxon.

The authors seem unaware that the formal erection of a new species based on lost material cannot be falsified pending additional material, being such taxon based only on Stromer's words: they based the diagnosis on Stromer's authority and not on replicable observations. The authors stated that Stromer's publications are a valid source of information: in principle, I do not question it. What I remark is that without an objective evidence (a fossil specimen) we cannot properly test the statements in those publications using an independent source.

The authors are invited to address my concern: how could we falsify Stromer's words in the case they (the words) were wrong? Only a second specimen could falsify those words. But if the species is not a real taxon, such second specimen does not exist, and would never be found. In particular, in the case of non-existence of such taxon, no additional specimen actually exists, and we would never be able to say anything about the validity/revision of that taxon. In that case, the species would merely be a non-falsifiable hypothesis included in our taxonomies and phylogenies. This must be avoided.

The authors listed a series of fossil taxa based on lost material in support of their preferred option (e.g., Maraapunisaurus). Yet, this is not a solid argument. First, I note that all of them are based on very fragmentary material (so, according to the conclusion of the manuscript, all of them would be considered nomina dubia even if based on accessible material). Second, all of the listed taxa are very problematic and barely mentioned in literature, thus of limited impact and significance. The exclusion of such taxa from the realm of the valid taxa would not impact in a significant way the progress of the palaeontological knowledge.

I encourage the authors to ask themselves: is the erection of this new taxon really necessary? Do we need another problematic African theropod to the list of the controversial dinosaur taxa? The authors provided examples of already named taxa based on available material from North Africa (so, taxa which at least epistemologically are more robust than the new species they wish to introduce) which are of controversial status: is the inclusion of another problematic species really necessary? I conclude that the taxonomy and inclusiveness of the North African theropods is sufficiently controversial and does not need another problematic species. Such new taxon would likely ignite a long debate on its validity, a debate unfortunately impossible to resolve if the species is not a real taxon and no material is available. Is another (potentially endless) debate on the validity of some dinosaur really welcome?

In summary, if the authors are right, only a second specimen would support their hypothesis: unfortunately, such second specimen does not exist, and pending its discovery, we cannot accept the institution of a taxon based uniquely on two photographs and the untestable acceptance of Stromer's words.

Note that from a palaeontological perspective, I have not a priori objections to the hypothesis that the Egyptian carcharodontosaur could be distinct from the Moroccan one. What I question is the epistemological basis for such hypothesis. In conclusion, the main elements of the manuscript in its current form, excluding the erection of a new species, are sufficient for a solid and significant paper: removing any mention to a new taxon would not impact the relevance and importance of the manuscript, and would likely be considered more positively than a manuscript introducing such a controversial and problematic species actually based on a destroyed fossil and just a few photos.

Reviewer #2: This version seems to have addressed all the issues that I and the other reviewer had previously suggested.

I did come across two typographical errors:

p. 25, ¶3 “Mapusaurus” is misspelled in the middle of the paragraph.

p. 26, ¶1 “carcharodontosaurids” is misspelled on the first line of the page.

7. PLOS authors have the option to publish the peer review history of their article (what does this mean?). If published, this will include your full peer review and any attached files.

Reviewer #1: No

Reviewer #2: **Yes: **Thomas R. Holtz, Jr.

---

## [Author Response · Author response to Decision Letter 1]

10 Aug 2024

We thank the reviewers for their additional revisions and critiques. In the following, we will once more address the criticism of the reviewers separately. 

Reviewer 1:

We thank reviewer 1 for the helpful reviews and we appreciate the concerns brought forward about possible epistemological issues that may arise from naming a new taxon. We have been debating this issue ourselves for a while and yet remain with our conclusion, that naming the Egyptian carcharodontosaur is a preferable solution over leaving a clearly diagnostic and scientifically significant taxon unnamed. 

The main concern of reviewer one “how could we falsify Stromer's words in the case they (the words) were wrong?” is valid, but has been thoroughly considered by us. In choosing the diagnostic features of our taxon, we did not use any features based solely on Stromers statements alone (Even though these are generally very accurate, as can be verified by crossreferencing Stromers publications with still existing material housed in the BSPG, like for example Lybicosuchus). For our taxon diagnosis we used only characters that are either

A) Directly visible on the picture of the specimen and are further supported by Stromers descriptions and/or illustrations of the specimen 

B) highly unusual features that would be hard to misinterpret and are clearly described by Stromer and clearly visible in the illustrations of the specimen

For a overview of this see the Table in the Response to Reviewers file.

A) Should not be problematic at all, as observations on the photographs of the specimen are reproducible

B) This is the only point that could be considered slightly more problematic, as the characters given here are not clearly visible in any photograph of the specimen. However, these characters represent quite unusal morphologies and are thus hard to misinterpret (like the foramina in the caudal vertebrae). They are further mentioned multiple times throughout different publications, further strengthening our use of these characters. Nonetheless we did consider the critique and split these off into a separate section of the diagnosis.

Even if we were to take out the characters falling under section B, the specimen would still be diagnostic simply based on all characters directly observable in A. The following point “If two distinct authors disagree on some of the statements by Stromer used here to define the new species, how could such disagreement be solved?” is thus moot, as the main arguments used by us to define said species are based on verifiable evidence.

As for the second point: 

“Just because the Egyptian material cannot be referred to the same taxon of the Moroccan material (a conclusion I can endorse) does not automatically mean that the former could be used to erect a new taxon.”

While this statement is generally true, a strong case can be made for the erection of a new taxon for the Egyptian carcharodontosaur. For one, the Egyptian taxon remains the only African carcharodontosaur with clearly associated cranial and postcranial elements, giving us the only source of information on the postcranial anatomy of the African members of this group. Furthermore, it remains one of the most informative African carcharodontosaurs, even with the specimen being destroyed. In our dataset, the Egyptian taxon is the second most completely coded African carcharodontosaur (78% missing data), after the neotype skull of Carcharodontosaurus saharicus, which is only slightly better coded (~76% missing data). All other African taxa in this group, and many of the named carcharodontosaurs for that matter, are significantly less complete (C. iguidensis 96% missing data, Eocarcharia ~94.5% missing data, Lajasvenator 86% missing data, Lusovenator 80% missing data, Siamraptor 82% missing data, Taurovenator 97.8% missing data, Veterupristisaurus 99% missing data). Even some of the better known South American taxa like Tyrannotitan only have slightly more phylogenetic information (77% missing data). In fact, the Egyptian carcharodontosaur is more completely coded than 33% of the OTUs used in our dataset. Given all of this we think it preferable to erect a new taxon, as we have already elaborated on further in our last Response to the reviewers.

Lastly, we want to address the statement of possible falsifiability of our taxonomic hypothesis:

“Only a second specimen could falsify those words. But if the species is not a real taxon, such second specimen does not exist, and would never be found. In particular, in the case of non-existence of such taxon, no additional specimen actually exists, and we would never be able to say anything about the validity/revision of that taxon. In that case, the species would merely be a non-falsifiable hypothesis included in our taxonomies and phylogenies. This must be avoided.”

This is an argument that, given the nature of morphospecies, is applicable not only to specimens that no longer exist but also to specimens that do still exist. Given the highly fragmentary nature of the fossil record, one can always make the argument that a given specimen is just an aberrant morphology of a known taxon. Unless a significantly large sample of fossils is found showing multiple transitionary forms between two different morphs (let’s say A and B), one can always make the argument, that A and B simply represent differing morphologies of the same taxon. Similarly, even if multiple specimens with the exact same morphologies of A and B are found but no transitionary forms, we still would not be able to confidently discredit an argument that we are simply still missing the transitionary forms. This is true whether or not we are referring to specimens that still exist or specimens that were destroyed. It is thus not a good argument against naming a species on no longer existing material.

“The authors listed a series of fossil taxa based on lost material in support of their preferred option (e.g., Maraapunisaurus). Yet, this is not a solid argument. First, I note that all of them are based on very fragmentary material (so, according to the conclusion of the manuscript, all of them would be considered nomina dubia even if based on accessible material). Second, all of the listed taxa are very problematic and barely mentioned in literature, thus of limited impact and significance. The exclusion of such taxa from the realm of the valid taxa would not impact in a significant way the progress of the palaeontological knowledge.”

We listed these taxa to underline our point that it is, in fact, possible to erect a new taxon based on no longer existing specimens. These examples were not meant as positive examples of such a practice, and we do agree with Reviewer 2 that these taxa should be considered nomina dubia even if they were based on accessible material. However, in our Manuscript we present 13 pages of detailed description of our proposed taxon, based both on a previous description by an author whose descriptions are generally considered to be accurate and on the illustrations (which, based on existing material figured by the same author, are probably also accurate) and on existing photographs of the specimen, which represents a substantial amount of information. This information is considerably more abundant than for the taxa mentioned above and, indeed, for many theropod dinosaurs known from still existing material, as outlined above. Disregarding this taxon from future phylogenies would be excluding a significant amount of information. (See also our previous comments addressing the second point)

“In conclusion, the main elements of the manuscript in its current form, excluding the erection of a new species, are sufficient for a solid and significant paper: removing any mention to a new taxon would not impact the relevance and importance of the manuscript, and would likely be considered more positively than a manuscript introducing such a controversial and problematic species actually based on a destroyed fossil and just a few photos.”

In theory, we agree with the main argument of this statement, that removing the erection of a new taxon and continuously referring to it with its specimen number would not significantly impact the relevance of this research in an ideal case. In practice, however, not naming this taxon would likely result in the following:

1. The Egyptian carcharodontosaur and its unique character combination would be more prone to be excluded from future phylogenies and other studies on the evolution, diversity and biogeography of carcharodontosaurid theropods. Unnamed species do not receive as much attention as named ones and are less likely to be considered in future works. If this happens, we lose a lot of anatomical information, and known carcharodontosaur diversity will be underestimated

2. As the reviewer himself admits, a convincing case is made that this specimen does represent a different taxon and (as shown above) it can be clearly diagnosed by a unique character combination and probably even several autapomorphic characters. Thus, it would be simply a matter of time before this taxon would be named by somebody.

Reviewer 2:

We thank reviewer 2 for his comments and corrected the respective typographical errors.

In conclusion:

We are acutely aware of the difficult situation we find ourselves in, and we acknowledge that erecting a new taxon for the Egyptian carcharodontosaur can be seen as problematic as the original specimen is no longer available. Nonetheless, we still think that making this material the type of a new taxon is the most preferable out of all the possible solutions for the tricky situation of this specimen, given the following points 

- The Egyptian taxon represents one of the most complete African carcharodontosaurs known

- It is the only specimen out of eastern Africa and the only African specimen combining cranial and postcranial material

- The diagnosis of our species is based overwhelmingly on photographically documented and thus verifiable characters, even if we were to completely disregard Stromers descriptions and drawings

- If, as both reviewers agree, we made a convincing argument that the Egyptian carcharodontosaur represents a distinct taxon and can be diagnosed (as we now hope to have sufficiently demonstrated), it is only a matter of time before this taxon will be named.

On a related note, regarding the validity of using observations based on illustrations, photographs and descriptions for the diagnosis of a taxon:

Any referral to a now lost specimen, be it for the erection of a neotype, the referral of a taxon, or for simple comparative reasons has to be done without a specimen. This is common practice in paleontology and has been done for some of Stromers dinosaurs, like Spinosaurus, but also for comparisons of “Carcharodontosaurus”, the specific specimen in question here. If one questions the validity of illustrations, photographs and descriptions of a specimen, then as a consequence, we would have to disregard lost specimens completely. Any comparisons to still existing specimens, or even the referring of neotypes for lost specimens would be impossible, as there would be no way to falsify these taxonomic hypothesis with the original specimen being destroyed.

Stromers Spinosaurus, for example, has been commonly been referred to in the literature, used for comparsions to other spinosaurids, in phyologenies and has had a neotype proposed for it in 2014. If we discard all of Stromers writings as unfalsifiable evidence, we would consequently loose a good part of the information on this taxon and there would be no way to ever erect a neotype for this taxon either. 

Consequentially, if one can use photographs, illustrations and descriptions to compare a taxon to a lost specimen, we do not see any convincing reason why one can not also use photographs, illustrations and descriptions to diagnose a new taxon, as it fundamentally requires the same comparative actions. 

We hope that we now make a strong enough case for the erecting of a new taxon.

---

## [Decision Letter · Decision Letter 2]

3 Sep 2024

PONE-D-24-16876R2Re-evaluation of the Bahariya Formation carcharodontosaurid (Dinosauria: Theropoda) and its implications for allosauroid phylogenyPLOS ONE

Dear Dr. Kellermann,

Thank you for submitting your manuscript to PLOS ONE. After careful consideration, we feel that it has merit but does not fully meet PLOS ONE’s publication criteria as it currently stands. Therefore, we invite you to submit a revised version of the manuscript that addresses the points raised during the review process.

We look forward to receiving your revised manuscript.

Kind regards,

Olga Spekker, Ph.D.

Academic Editor

PLOS ONE

Journal Requirements:

Additional Editor Comments:

Dear Dr Kellermann,

We appreciate you submitting your manuscript to PLOS ONE and thank you for giving us the opportunity to consider your work.

I have completed my evaluation of your manuscript, which has been reviewed by three highly qualified reviewers during the different stages of the review process – all of the reviewers agree it is worth to be published in PLOS ONE. However, as I already highlighted in the last review round, I agree with one of the original reviewers that the taxonomic part erecting a new species is problematic. Therefore, I invited another reviewer in this review round, who also agrees with us that “introducing a new taxon in this circumstances is highly problematic and not warranted”, Based on the above, I kindly ask you to review that part and resubmit your manuscript after addressing the reviewer’s comments below. Unfortunately, without that, your manuscript cannot be accepted. Please, note that your revised submission may need to be re-reviewed.

PLOS ONE values your contribution and I look forward to receiving your revised manuscript.

Yours sincerely,

Dr Olga Spekker

Reviewers' comments:

Reviewer's Responses to Questions

**Comments to the Author**

1. If the authors have adequately addressed your comments raised in a previous round of review and you feel that this manuscript is now acceptable for publication, you may indicate that here to bypass the “Comments to the Author” section, enter your conflict of interest statement in the “Confidential to Editor” section, and submit your "Accept" recommendation.

Reviewer #3: (No Response)

2. Is the manuscript technically sound, and do the data support the conclusions?

Reviewer #3: Partly

3. Has the statistical analysis been performed appropriately and rigorously? 

Reviewer #3: Yes

4. Have the authors made all data underlying the findings in their manuscript fully available?

Reviewer #3: Yes

5. Is the manuscript presented in an intelligible fashion and written in standard English?

Reviewer #3: Yes

6. Review Comments to the Author

Reviewer #3: I congratulate the authors on their detective work and agree with them in many regards (e.g., authors need to be careful when comparing seemingly coeval assemblages from different parts of North Africa). The descriptions and observations are good for the most part and overall the MS is well-written.

Having said that, I concur with previous reviewers that it is highly problematic to publish a new genus and species name for a set of fossils that were destroyed many decades ago, based on drawings and photographs (I should add that I examined one of the photographs first hand). Renaming old - and most importantly lost - material could set a very problematic precedent indeed. It would be an entirely different situation if relevant new material had been found in the Bahariya Fm, but that is not the case here. The two other reviewers have already made some valid points, which I think have not been fully addressed. I am adding a few more here.

(1) The difficulty of reconciling Stromer's observations with the (sometimes difficult to interpret) photographs and drawings (which do not always match the photographs) in absence of an actual, physical set of fossils is evident in many passages, e,g,.:

"An expansion of the narial fossa to the nasal in the form of a supranarial fossa, as it is present in Acrocanthosaurus, Concavenator, Carcharodontosaurus and Meraxes [11,18,43,44] is not clearly visible in the photo of SNSB-BSPG 1922 X 46, but Stromer [1] described the area of the anterior end of the nasal lateral to the median premaxillary process as slightly concave. However, a clearly defined rim of such a fossa, as it is present in Carcharodontosaurus ([18]:Fig 134A) is not at all indicated in Stromers figures, nor visible on the photograph."

What are we to make of ambiguous aspects of the description, which cannot be resolved? As interesting as the new photograph is, the case made here to erect a new taxon is on very shaky ground, as many questions remain and the photograph, while of huge interest historically, does not offer a huge amount of new insights anatomically. This is not meant to downplay the scientific information the authors have carefully extracted from the image (well done), but simply reflects the limitations of an old photograph.

(2) Even if all of these issues are set aside, I don't think a highly convincing case can be made at this stage that this is a different genus (as opposed to, for example, a different species, as has been done with Carcharodontosaurus iguidensis, and even that would need carefully balanced arguments, see below). These decisions are obviously notoriously difficult to comment on when dealing with extinct taxa, as they are often somewhat arbitrary and based on different approaches to naming taxa (splitters vs lumpers etc). Either way, it would have been good and very important to include detailed comparisons to other theropods, known from large sample sizes (e.g. Allosaurus, Coelophysis), as these often show a rather remarkable degree of variation (reflecting everything from ontogeny to individual variation etc), especially in their skulls.

Several features highlighted by the authors may not be as unique as claimed here, including but not limited to:

- The horn-like rugosity may reflect sexual dimorphism.

- Speaking of rugosity, isolated specimens from the Kem Kem (some not accessioned in public collections) show a wide range of different degrees of rugosity (which is at odds with the author's comment that the Egyptian specimen shows ridges, furrows and rugosity that are far more pronounced than in the Moroccan Carcharodontosaurus). To be fair, as much of this material is not publicly available, one cannot blame the authors, but it might be of interest to them.

- Comparisons between the braincases are interesting but should come with lots of caveats: the Berlin specimen (endocast) is not very detailed (i.e. it is not necessarily the most faithful and undistorted endocast) and meaningful comparisons are consequently difficult.

- Note that teeth matching the shape of Stromer's specimen have been found in Morocco (the authors only mention one such tooth, figure 8E, but others have been collected), so in this regard, the Egyptian specimen is not necessarily unique. Note also that carcharodontosaurid teeth found in the Kem Kem show quite a bit of variation, likely reflecting different positions in the jaws.

Some other aspects may well represent meaningful differences, but for reasons outlined above (and below) I maintain that introducing a new taxon in these circumstances is highly problematic and not warranted.

(3) Geography. " Since Stromer’s referral of the Egyptian specimen to the Algerian M. saharicus is questionable and the Moroccan material was found in closer proximity to it, we accept the designation of the latter as neotype of C.

saharicus" - - - This is not convincing. If one looks at the geographical range of extant animals like Nile crocodiles, lions, hippos etc it is evident that they range across huge parts of the African continent (certainly historically). There is no particularly convincing reason to believe that theropod dinosaurs were not ranging across vast areas of Africa, so we need to be careful when arguing for taxonomic assignments based on a distance in km from one site to another. These arguments should be based on unambiguous anatomical observations etc, and not on the distance from Morocco to Algeria vs the distance from Algeria to Egypt...

(4) The authors argue for an exceptional case (avoiding the strong recommendation that scientists not designate as type material specimens the author only knows from descriptions and illustrations, but has not studied first hand) by stating that "1) The material described by Stromer [1] has played a pivotal role for our understanding of carcharodontosaurid anatomy, phylogeny and evolution". This is a problematic statement in the sense that, as this paper amply demonstrates, very little is actually known about the Egyptian material. It certainly played an important historical role, but to claim this long lost material played a pivotal role in carcharodontosaurid phylogeny and evolution is somewhat at odds with one of the key messages of this paper: that we know so little about this Egyptian material that much of what people have suggested about this material in the past - including its name and the referral of other material to it - is now in doubt according to the authors. If we contrast this with, say, Acrocanthosaurus, which has been described in great detail, the difference becomes quite clear.

In summary, I don't think it is wise or warranted to erect a new taxon in these unusual circumstances. Note also that, contrary to one claim made here in response to reviewer comments, taxa do not have to be named to be noticed in other papers. Indeed, the unnamed Kem Kem abelisaur, or the (previously unnamed) Kem Kem tapejarid were not neglected at all and found their way into numerous papers. The authors need not worry, I am sure that this paper is going to elicit a great deal of interest and it will be an essential reference for many future papers on Cretaceous North African dinosaurs - going down a slippery slope and naming a long-destroyed set of bones is not necessary to achieve this.

A few other comments:

- "S1. Select measurements of Alllissaurus markgrafi in mm." I assume this is from a previous version of the MS?

- The age of the Bahariya Fm (and the Kem Kem Group, and possibly roughly coeval strata in Algeria) is highly uncertain, and the Cenomanian age for many of these North African assemblages published by many authors in the past is based on very limited data (and a decent dose of circular reasoning).

- "Although we do agree that differences between Sauroniops and Carcharodontosaurus saharicus pointed out by Cau et al. [186–188] are intriguing, we agree with Cau et al [186] that this single frontal bone does warrant the erection of a new genus or species and instead consider Sauroniops a nomen dubium, pending the discovery of new

material." - - - - I think the authors meant to say "we disagree with Cau et al etc.

- When discussing faunal overlap, it is important to be clear about what the comparisons are based on. It is worth noting that only one sauropod genus is known from the Kem Kem Group and only two have been described from the Bahariya Fm. There may well be overlap, including at the genus level, and the discovery of isolated titanosaur remains in Morocco (which may turn out to be very similar to Paralititan, for example) certainly leaves some room for potential overlap. Outcrops in the Kem Kem are far more extensive, and A LOT more material has been found, therefore we need to be careful when it comes to how much importance we give to the presence/absence of some of the crocodyliforms etc. We are really not comparing like with like at all, so we need to be careful when deciding how much importance we give to the presence of seemingly identical or very similar taxa based on isolated remains (as the authors correctly point out), but also when we give too much importance to the absence of shared taxa (something the authors are not doing here) because, again, we are comparing one locality that has barely been explored in the last century and another one that has been explored - exploited one might say - by large numbers of commercial collectors (many Kem Kem specimens, including most of the crocodyliforms the authors refer to here as examples of taxa unique to Morocco, were commercially acquired and often illegaly exported) in addition to a handful of scientific expeditions.

- and it seems that the anteromedial corners of the fossae were overhang by a lamina, as it is the case in other carcharodontosaurids [52]. - - - This should read "as is the case".

7. PLOS authors have the option to publish the peer review history of their article (what does this mean?). If published, this will include your full peer review and any attached files.

Reviewer #3: No

---

## [Author Response · Author response to Decision Letter 2]

11 Sep 2024

Response to Editors: 

Dear Dr. Spekker,

we would like to thank you for inviting a third reviewer and appreciate your comments on our Manuscript. We have now hopefully adressed all the major points of critique within our Manuscript.

However, after careful consideration we remain with our initial decision to erect a new genus and species in this case. We will briefly summarize our points as to why we still remain with our current position.

1. The taxon is distinct and diagnostic (something that the initial two Reviewers seem to have accepted, or at least do not have any objections to)

2. The taxon is scientifically significant

3. Following ICZN Declration 45 73G-J it is possible to name a taxon based on lost material in exceptional cases. All of the requirements for this are met here

4. Naming this taxon will not set a problematic precedent, as this has been done before and there are recommendations set in place by the ICZN to do just that.

5. It is a matter of time before this taxon is named, if not by us then by somebody else. 

Naming a new genus is a pivotal part of the Manuscript and the logical conclusion fo the data we present. Which is why we will consider sending the Manuscript to a different journal, if it cannot be accepted. We hope we could now make a convincing point for our case and would like to invite all current reviewers and editors to reconsider after the numerous adjustments we made to our Manuscript.

Response to Reviwers: 

First of all, we would like to thank Reviewer 3 for their helpful commentaries, which in some cases have led to improvements in the reasonings for our arguments given in the manuscript. However, it appears to us, that there is a different approach to how we addressed some of the issues present in our paper.

We would first like to respond to the individual points brought forward by Reviewer 3 and end with a general conclusion.

Response to individual points

Having said that, I concur with previous reviewers that it is highly problematic to publish a new genus and species name for a set of fossils that were destroyed many decades ago, based on drawings and photographs (I should add that I examined one of the photographs first hand). Renaming old - and most importantly lost - material could set a very problematic precedent indeed. 

It would not set a precedent, as this has been done before (see Apesteguia 2007; Carpenter 2018), even on much more incomplete and more poorly described and illustrated material. Furthermore, a number of taxa have been described on material that had been destroyed by the time of publication and only photos or drawings, or, in the best case, a cast remains, even if the authors might still have seen it in person before destruction (Friedman & Daeschler 2006; Brum et al. 2021), and in some cases (e.g. ichnotaxonomy) this is even common practice. In these cases we have to trust that the observations of the authors, which cannot be verified from the photos directly are correct – why should we not trust an author who described fossils a hundred years ago? Thus, we do not see why our work would set a “problematic precedent”.

 (1) The difficulty of reconciling Stromer's observations with the (sometimes difficult to interpret) photographs and drawings (which do not always match the photographs) in absence of an actual, physical set of fossils is evident in many passages, e,g,.: "An expansion of the narial fossa to the nasal in the form of a supranarial fossa, as it is present in Acrocanthosaurus, Concavenator, Carcharodontosaurus and Meraxes [11,18,43,44] is not clearly visible in the photo of SNSB-BSPG 1922 X 46, but Stromer [1] described the area of the anterior end of the nasal lateral to the median premaxillary process as slightly concave. However, a clearly defined rim of such a fossa, as it is present in Carcharodontosaurus ([18]:Fig 134A) is not at all indicated in Stromers figures, nor visible on the photograph."

As we pointed out before, the drawings, photographs, and Stromers descriptions generally do agree. There is basically a direct overlap between photographs and scientific drawings of the specimens, showcasing the great accuracy of these illustrations. We compiled some quick examples of this (Fig. 1). Note that these are not to scale for time reasons. One could make further comparisons, see for example the photographs of the two mounted Bahariya theropods (our MS and Smith et al. 2006) in comparison to the illustrations given in Stromer 1931 and Stromer 1915. Stromers descriptions can therefore be crossreferenced with the illustrations and with the still existing specimens (like Lybicosuchus) and they are shown to be very accurate. 

In our case there are some details described by Stromer that cannot be seen in the illustrations or photographs, as the specimen was not depicted from the right angle, or the photograph lacks resolution. Where this was the case, we have always indicated this. 

The given example of the “narial fossa to the nasal in the form of a supranarial fossa” is actually a good case for this. We simply state that “An expansion of the narial fossa […] is not clearly visible in the photo” We do not make any statement on if or not this was present, but do add that “Stromer [1] described the area of the anterior end of the nasal lateral to the median premaxillary process as slightly concave“. This description would fit with the concavities (or slight depression) commonly found in theropods (such as Allosaurus) that are also often not clearly visible in photographs. However, in some taxa, such as Acrocanthosaurus, Sinraptor, or Asfaltovenator, this supranarial fossa has a marked, sharply defined rim, which is clearly visible in photos or drawings of the specimens (see Eddy & Clarke 2011: fig. 3; Currie & Zhao 1993: figs. 3, 5; Rauhut & Pol 2019: fig. 1A, B). This is also the case for the Moroccan specimen of Carcharodontosaurus (Ibrahim et al. 2020: fig. 134A). Such a marked rim should, if it was present, be visible in the photograph of the Egyptian specimen, and it would be very odd, if the otherwise quite detailed drawing of the nasal (and even the rather detailed descriptions of Stromer) would have omitted such a clear structure. Thus, in our statement we do not make any inferences about the exact shape of this concavity yet simply state that “a clearly defined rim of such a fossa” as is the case in some of the aforementioned taxa was not present. In cases such as these we were also conservative in our codings of the specimen for the dataset.

This is also the reason why we now restricted the diagnosis to characters that can be verified from at least two sources.

Fig. 1.: Examples for the accuracy of scientific illustrations in Stromers publications

What are we to make of ambiguous aspects of the description, which cannot be resolved? 

Where this was the case, we treated ambiguous aspects of the descriptions as such. We address this point further in our conclusion.

 (2) Even if all of these issues are set aside, I don't think a highly convincing case can be made at this stage that this is a different genus (as opposed to, for example, a different species, as has been done with Carcharodontosaurus iguidensis, and even that would need carefully balanced arguments, see below). These decisions are obviously notoriously difficult to comment on when dealing with extinct taxa, as they are often somewhat arbitrary and based on different approaches to naming taxa (splitters vs lumpers etc). 

We disagree here! If a specimen is referred to a certain taxon, this should be made on the basis of discrete apomorphies or a unique character combination shared with another specimen of the same taxon, not on the basis of general similarity or perceived affiliation with a higher clade (Carcharodontosauridae in the current case). Among the apomorphies listed by Brusatte & Sereno 2007, only the “pronounced grooved sculpturing of nearly the entire lateral surface of the maxilla” is actually shared between the Moroccan and Egyptian specimens. However, the rugositiy in the maxillae is a trait also present in other carcharodontosaurids like Meraxes or Giganotosaurus and thus is not diagnostic on a genus level. Other traits cannot be verified due to missing overlap, or are based on the incorrectly referred cervical vertebrae see [32, 33]. 

Our null hypothesis should not be that all specimens found that are generally similar do represent the same taxon (in this case we could go back to name every theropod Megalosaurus…). We make the argument that the Egyptian specimen is as distinct from the Moroccan neotype skull as are other, generally accepted carcharodontosaurid taxa, and this is supported by the phylogenetic hypothesis presented. If you want to keep the Egyptian specimen in the genus Carcharodontosaurus under this hypothesis, then at least the South American carcharodontosaurids Tyrannotitan, Meraxes, Giganotosaurus and Mapusaurus should also be subsumed under Carcharodontosaurus (Please note that we also question if Carcharodontosaurus iguidensis actually represents the genus Carcharodontosaurus (based on the phylogenetic results), but this question is beyond the scope of our work). If the reviewer (or any other researcher interested in this matter) finds discrete characters that conclusively show that the Egyptian specimen can indeed be referred to the genus Carcharodontosaurus to the exclusion of other carcharodontosaurid taxa, we would like to hear about that, but a general similarity does not suffice.

Either way, it would have been good and very important to include detailed comparisons to other theropods, known from large sample sizes (e.g. Allosaurus, Coelophysis), as these often show a rather remarkable degree of variation (reflecting everything from ontogeny to individual variation etc), especially in their skulls.

This statement is problematic for several reasons. First, the said variation in the named taxa has so far nowhere been documented, and personally studying all available specimens of Allosaurus and Coelophysis is certainly beyond the scope of this work. Second, as the reviewer might be aware, there are controversies about the taxonomic identities of some of this material, with at least three different species being currently recognized in Allosaurus (fragilis, europaeus and jimmadseni), and some authors either synonymizing or separating some specimens on species or even generic level (e.g. Saurophaganax, Epanterias). The entire sample of Coelophysis from Ghost Ranch needs to be revised, but the identification of other theropod taxa (Tawa) and theropod-like pseudosuchians (Effigia) from the same locality cast doubt on the previous assumption that all specimens of this sample that have originally been identified as Coelophysis represent the same taxon; again, such a revision of all material from Ghost Ranch has not been carried out and is clearly beyond the scope of our work. Furthermore, as noted above, the differences we mark between the Egyptian specimen and the Moroccan neotype skull are as considerable or even more so than between other carcharodontosaurid taxa. If we disregard these differences as individual or ontogenetic variation, we would also have to consider the widely accepted South American genera, Mapusaurus, Gigantotosaurus, Meraxes and Tyrannotitan to be mere junior synonyms of Carcharodontosaurus.

At the end of the day, any distinction between taxa, and especially genera on the base of morphology alone is somewhat arbitrary and meant more for communication rather than being an actual existing structure in nature. This is especially the case in fossil taxa, regardless whether or the material is still present, but one can even make the same arguments for modern genera. Thus, this point raised by the reviewer is a general concern in taxonomy and does not concern the question whether taxa might be erected on no longer existing material in general.

If we follow the scientific consensus and agree that the differences between the Moroccan specimen and either of the South American taxa warrants a distinction at the generic level, then this also applies for the Moroccan and Egyptian specimens.

Several features highlighted by the authors may not be as unique as claimed here, including but not limited to:

- The horn-like rugosity may reflect sexual dimorphism.

This is certainly true, but conjectural. Such a sexual dimorphism concerning the absence or presence of horns has so far not been demonstrated in any nonavian dinosaur, although this might simply be due to low sample size for many of the taxa concerned. However, in taxa where numerous specimens of horn (or similar ornamentation) -bearing taxa are known (e.g. Ceratosaurus, Allosaurus, Majungasaurus), this structure is always present, even though the exact development may vary. 

Quite recently, this exact question was also brought forward by Pol et al. 2024 – in their paper describing the South American abelisaurid Koleken. We will cite their thoughts on the question directly here, as it is applicable for our situation: “To date, presence or absence of horns have not been documented as a case of sexual dimorphism in archosaurs, either

extinct or extant (Padian and Horner, 2011; Hone et al., 2012), and have only been reported in a Triassic

archosauromorph (Sengupta et al., 2017). More specifically, no dinosaur has been determined to exhibit sexual dimorphism under rigorous analysis (Mallon, 2017). Body size has been reported as a feature

showing sexual dimorphism, but Hone and Mallon (2017) suggested that a minimum of 60 animals

were needed to detect this difference in alligators. Thus, data from other dinosaurs (and more generally,

archosaurs) do not suggest sexual dimorphism as a likely scenario, although with only two specimens at

hand we are unable to rigorously test this hypothesis. Moreover, the differences found between the type of Koleken and Carnotaurus are not restricted to the frontal horns but extend to multiple other bones, including the postorbital, nasal, dorsal vertebrae, and femur. Therefore, at the moment we consider it more conservative to recognize these as taxonomic differences between two closely related abelisaurid species that lived during the Maastrichtian in central Patagonia” 

So while we cannot 100% exclude the possibility that some of these features might be the result of sexual dimorphism, we come to the same conclusion as Pol et al. 2024 and consider it more conservative to regard these differences as taxonomic differences between related taxa, especially given that there are other differences that cannot be directly linked to possible display structures (See here our section “The taxonomic distinction of SNSB-BSPG 1922 X 46 from Carcharodontosaurus and its implications”). However, we added a cautionary note for the nasal horn, acknowledging that sexual dimorphism cannot be excluded.

- Speaking of rugosity, isolated specimens from the Kem Kem (some not accessioned in public collections) show a wide range of different degrees of rugosity (which is at odds with the author's comment that the Egyptian specimen shows ridges, furrows and rugosity that are far more pronounced than in the Moroccan Carcharodontosaurus). To be fair, as much of this material is not publicly available, one cannot blame the authors, but it might be of interest to them.

This is indeed of interest, but the implications of this are uncertain. There at least some indications that more than one carcharodontosaurid is present also in the Kem Kem assemblage (see for this Cau et al. 2012, 2013 and Paterna & Cau 2023) – thus, we would first need to established that these isolated (and so far undescribed) specimens indeed all belong to Carcharodontosaurus. For now, we can only compare the Egyptian specimen with the neotype skull and the differences here are clear.

- Comparisons between the braincases are interesting but should come with lots of caveats: the Berlin specimen (endocast) is not very detailed (i.e. it is not necessarily the most faithful and undistorted endocast) and meaningful comparisons are consequently difficult.

We agree with this statement, and added a cautionary note. However, it is the only physical specimen available, and differences ar

---

## [Editor Report · Decision Letter 3]

13 Sep 2024

Re-evaluation of the Bahariya Formation carcharodontosaurid (Dinosauria: Theropoda) and its implications for allosauroid phylogeny

PONE-D-24-16876R3

Dear Dr. Kellermann,

We’re pleased to inform you that your manuscript has been judged scientifically suitable for publication and will be formally accepted for publication once it meets all outstanding technical requirements.

Kind regards,

Olga Spekker, Ph.D.

Academic Editor

PLOS ONE
---

## [Editor Report · Acceptance letter]

7 Oct 2024

PONE-D-24-16876R3 

PLOS ONE

Dear Dr. Kellermann, 

I'm pleased to inform you that your manuscript has been deemed suitable for publication in PLOS ONE. Congratulations! Your manuscript is now being handed over to our production team.

Kind regards, 

on behalf of

Dr. Olga Spekker 

Academic Editor

PLOS ONE